# Attention with Trained Embeddings
# Provably Selects Important Tokens

**Diyuan Wu**[1,*]  **Aleksandr Shevchenko**[2,*]  **Samet Oymak**[3]  **Marco Mondelli**[1]

ISTA[1]   ETH Zürich[2]   University of Michigan[3]

## Abstract

Token embeddings play a crucial role in language modeling but, despite this practical relevance, their theoretical understanding remains limited. Our paper addresses the gap by characterizing the structure of embeddings obtained via gradient descent. Specifically, we consider a one-layer softmax attention model with a linear head for binary classification, i.e., $\mathtt{Softmax}(p^\top \boldsymbol{E}_X^\top)\boldsymbol{E}_X v = \frac{\sum_{i=1}^T \exp\left(p^\top E_{x_i}\right)E_{x_i}^\top v}{\sum_{j=1}^T \exp\left(p^\top E_{x_j}\right)}$, where $\boldsymbol{E}_X = [E_{x_1}, \ldots, E_{x_T}]^\top$ contains the embeddings of the input sequence, $p$ is the embedding of the $\langle\mathrm{cls}\rangle$ token and $v$ the output vector. First, we show that, already after a single step of gradient training with the logistic loss, the embeddings $\boldsymbol{E}_X$ capture the importance of tokens in the dataset by aligning with the output vector $v$ proportionally to the frequency with which the corresponding tokens appear in the dataset. Then, after training $p$ via gradient flow until convergence, the softmax selects the important tokens in the sentence (i.e., those that are predictive of the label), and the resulting $\langle\mathrm{cls}\rangle$ embedding maximizes the margin for such a selection. Experiments on real-world datasets (IMDB, Yelp) exhibit a phenomenology close to that unveiled by our theory.

## 1  Introduction

The introduction of the attention mechanism [5, 41] marked a paradigm shift in the design of frontier machine learning models, leading to significant advances such as ChatGPT [2], Claude [3], AlphaFold [18], CLIP [30] and Dall-E [31]. This success prompted a surge of interest in understanding the structure and function of attention layers, with their optimization dynamics and inductive biases being object of extensive theoretical research [1, 7, 10, 23, 37, 42] (see also Section 2). Embeddings are a crucial component of the attention mechanism [45], especially for downstream adaptation [13, 16, 19] with some works [20, 45] specifically highlighting their importance. However, despite the importance of learning embeddings, the existing analyses of transformer-like architectures either ignore the properties of embeddings by resorting to orthogonal structures [44], or omit embeddings completely by considering unprocessed inputs [39].

Our paper fills this gap by studying directly the embedding training dynamics. Specifically, we aim to provide theoretical insight to the following questions:

> *What is the structure learnt by the embeddings during gradient descent training?*
> *How is this structure related to the statistical properties of the data?*

In Figure 1, we investigate these questions by analyzing the embeddings of a two-layer transformer trained on a sentiment analysis task on IMDB and Yelp reviews. The plots reveal a remarkable simplicity in the structure of the learned embeddings, which capture the frequency of appearance of tokens in the dataset. Specifically, the predictive mechanism (overlap with the regression coefficient

---

*Equal contribution. Corresponding author: `diyuan.wu@ist.ac.at`.

39th Conference on Neural Information Processing Systems (NeurIPS 2025).

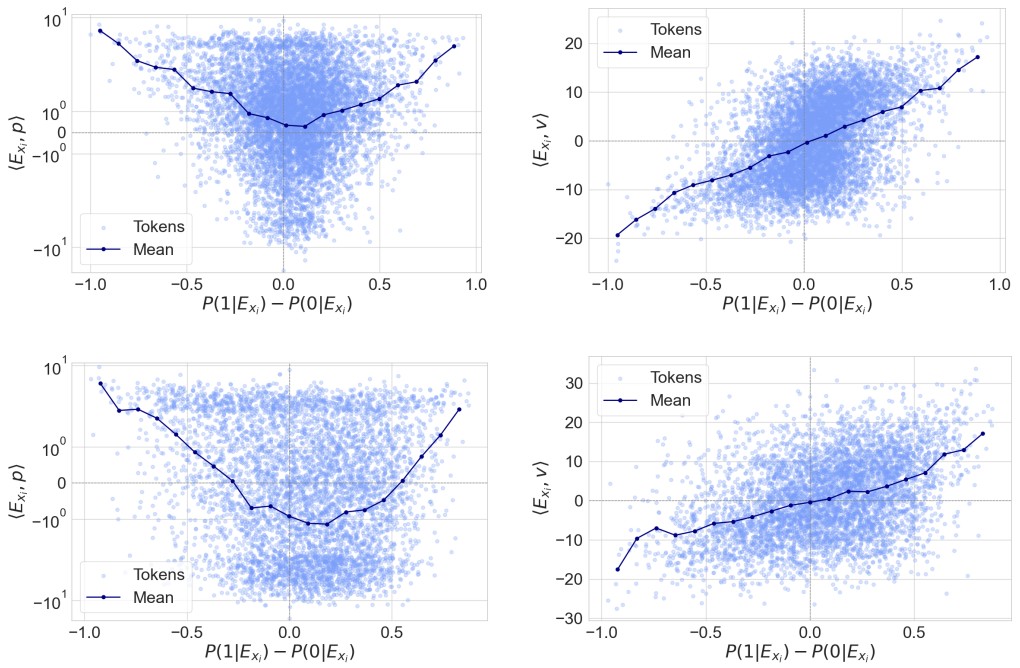

Figure 1: Dot-product of token embeddings with $\langle \text{cls} \rangle$ embedding $p$ (left) and regression coefficients $v$ (right), as a function of the token-wise difference in posterior probabilities, for IMDB (top row) and Yelp (bottom row) datasets. We consider the two-layer attention model in (18) with all parameters trained until convergence.

$v$) favors the tokens which appear more frequently in the corresponding positive/negative context. A similar pattern emerges at the selection stage of the attention mechanism (overlap with the $\langle \text{cls} \rangle$ embedding $p$), i.e., more frequent tokens have a higher attention score.

For the theoretical study of this emergent structure, we focus on a one-layer softmax attention model. Namely, for an input sequence $X = [x_1, \cdots, x_T]$, the output of the model is given by

$$f(X; p, \boldsymbol{E}) = \texttt{Softmax}(p^\top \boldsymbol{E}_X^\top) \boldsymbol{E}_X v = \frac{\sum_{i=1}^{T} \exp\left(p^\top E_{x_i}\right) E_{x_i}^\top v}{\sum_{j=1}^{T} \exp\left(p^\top E_{x_j}\right)}, \tag{1}$$

where $\boldsymbol{E}_X = [E_{x_1}, \ldots, E_{x_T}]^\top$ contains the embeddings of the input $X$, $p$ is the embedding of the $\langle \text{cls} \rangle$ token and $v$ is the final regression vector. Our main results are summarized below:

- We show that, already after a single step of gradient training with the standard logistic loss, the embeddings $\boldsymbol{E}_X$ capture the importance of tokens in the dataset by aligning with the output vector $v$ proportionally to the corresponding empirical frequencies (Lemma 4.1).

- In a setting where each sequence contains a single important token, the $\langle \text{cls} \rangle$ embedding obtained from gradient flow must select all important tokens. We further characterize all the possible directions that the $\langle \text{cls} \rangle$ embedding may converge to, which are the max-margin solutions associated to feasible token selections (Theorem 4.3).

- While in general the $\langle \text{cls} \rangle$ embedding may select irrelevant tokens, we identify sufficient conditions leading to the selection only of important tokens (Lemmas 4.4 and 4.5).

## 2 Related work

**Implicit bias, margin maximization, attention.** The implicit bias literature has been instrumental in understanding the behavior of neural networks or overparameterized models optimized by gradient methods [8, 4, 26]. A key phenomenon is that gradient descent on separable data with logistic loss directionally converges to the max-margin separator [35, 15]. More recently, a series of works [36, 22, 17, 40, 21, 34, 37, 32] has established an equivalence between the optimization geometry

of self-attention and a hard-margin SVM problem selecting a subset of tokens via linear constraints on the outer-products of token pairs. Compared to these works that mostly focus on the training of single-layer attention weights, we point out two differences. First, we study the role of embeddings and their joint training with the $\langle\mathrm{cls}\rangle$ token. Second, under our data model, we establish benign properties of the solution reached at convergence (which may not hold for arbitrary datasets [37]).

**Theory of attention.** A line of work [23, 24, 27] has explored whether attention-based architectures can extract causal structure from Markovian inputs. The mechanics of next-token prediction when training a single self-attention layer is characterized in [21]. Towards understanding how to utilize structural properties of the data, the behavior of transformers on sparse token selection tasks is considered in [33, 43]. The study [14] provides a theoretical justification to the tendency of modern language models to generate repetitive text by showing that the underlying self-attention mechanism collapses into sampling only a limited subset of tokens. This stands in contrast to the slightly different setup of [38] where the transformer model does not degrade to a "winner-takes-all" strategy. The works [10, 11, 9] take a mean-field view to analyze the clustering behavior in transformer representations that emerges after successive applications of the attention block. Under a random feature design, it is shown in [6] that softmax attention exhibits a sensitivity property which allows for a sharp change in attention scores given the perturbation of a single token. The role of the attention mechanism is also studied in [29] for prompt-tuning and in [12] for test-time-training.

## 3 Problem setup

**Data model.** We focus on binary text classification problems. We consider a (context) vocabulary set $\mathcal{S}$ with size $|\mathcal{S}|$, together with a $\langle\mathrm{cls}\rangle$ token for classification. Let $(X_i, y_i)_{i=1}^n$ be the dataset containing $n$ context sequences, where $y_i \in \{-1, 1\}$ and each context sequence $X \in \mathcal{X}_n := \{X_1, \ldots, X_n\}$ contains $T$ tokens, i.e., $X = [x_1, \ldots, x_T]$ with $x_i \in \mathcal{S}$. Without loss of generality, we let $\mathcal{S}$ be the set of tokens that appears in $\mathcal{X}_n$, as the embeddings of the remaining tokens are not trained and are not relevant for the problem at hand.

**Architecture.** We consider a one-layer softmax attention model with a linear head for classification. First, we append a $\langle\mathrm{cls}\rangle$ token at the end of the sequence $X$, and then we embed each token into a vector of dimension $d$. Namely, after the embedding layer, we have $\boldsymbol{E}_X = [E_{x_1}, \ldots, E_{x_T}]^\top \in \mathbb{R}^{T \times d}$, where $E_s \in \mathbb{R}^d$ denotes the embedding of the token $s$. We let $\boldsymbol{E} \in \mathbb{R}^{|\mathcal{S}| \times d}$ be the embedding matrix of all context tokens and $p \in \mathbb{R}^d$ the embedding of the $\langle\mathrm{cls}\rangle$ token.

We focus on the architecture defined in (1) where, given a vector $a \in \mathbb{R}^T$, $[\mathtt{Softmax}(a)]_i := \frac{\exp(a_i)}{\sum_{j=1}^T \exp(a_j)}$ for $i \in \{1, \ldots, T\}$. We remark that the same model is also studied in [37, 32]. In practice, it is common to include the $W_{KQ}$ matrix and consider a model with output $f(X; p, W_{KQ}, \boldsymbol{E}) = \mathtt{Softmax}(p^\top W_{KQ} \boldsymbol{E}_X^\top) \boldsymbol{E}_X v$. Since $p^\top W_{KQ}$ plays the same role as $p$ and one can easily reconstruct $W_{KQ}$ from $p$ in each gradient update as discussed in [37], we use the model in (1) for simplicity. We also note that our experiments on two-layer networks with a $W_{KQ}$ matrix reported in Appendix F.4 do not show any qualitatively different behavior, which further justifies our simplification.

**Optimization problem.** The output vector $v$ is fixed and all the embedding vectors $p, \boldsymbol{E}$ are trained with the standard logistic loss:

$$\mathcal{L}(\boldsymbol{E}, p) = \frac{1}{n} \sum_{k=1}^n \log(1 + \exp(-y_k f(X_k; \boldsymbol{E}, p))) = \widehat{\mathbb{E}}\left[\log(1 + \exp(-y f(X; \boldsymbol{E}, p)))\right], \quad (2)$$

where the notation $\widehat{\mathbb{E}}$ is a shorthand for the average over the dataset $\mathcal{D} = \{(X_k, y_k)\}_{k=1}^n$.

**Empirical statistics of each token in the dataset.** The goal of the paper is to characterize the structure of the embeddings $\boldsymbol{E}, p$ obtained by optimizing the objective (2) via gradient descent, and we show that such structure is related to the empirical statistics of the tokens in the dataset. Specifically, after training, the softmax attention learns to select tokens that are more correlated to the labels based on the dataset. To quantify the correlation between a token $s$ and the label $y$, we define the *average signed frequency* of a token as:

$$\alpha_s := \frac{1}{nT} \sum_{(X,y) \in \mathcal{D}} y \sum_{i=1}^T \mathbb{1}_{x_i = s} = \frac{1}{T} \widehat{\mathbb{E}}\left[y \sum_{i=1}^T \mathbb{1}_{x_i = s}\right]. \quad (3)$$

In words, $\alpha_s$ is obtained by taking the number of occurrences of $s$ in sequences with a positive label, subtracting the number of occurrences of $s$ in sequences with a negative label, and finally dividing by the total number of tokens $nT$. As such, it provides an average of the signed frequency of $s$, where the sign comes from the label of the sequences in which the token appears.

**Definition 3.1** (Positive, negative and irrelevant tokens)**.** We say that a token $s$ is *(i) positive* if $\alpha_s > 0$, *(ii) negative* if $\alpha_s < 0$, and *(iii) irrelevant* if $\alpha_s = 0$. Moreover, we say that a token $s$ is *completely positive* if it appears only in sequences with label $1$, and *completely negative* if it appears only in sequences with label $-1$.

In words, a token is positive (negative) when it is more frequently associated to the positive (negative) label; tokens that appear the same number of times associated to positive and negative labels are irrelevant. The quantity $\alpha_s$ quantifies how positive/negative a token is. Intuitively, if either $\alpha_s > \alpha_{s'} > 0$ or $\alpha_s < \alpha_{s'} < 0$ , then the token $s$ is more relevant than the token $s'$ for the classification task and, therefore, we expect that this will be reflected into the structure of the corresponding embeddings.

## 4   Main theoretical results

We show that the trained attention model (1) learns to select important tokens from the empirical statistics of the data. First, in Section 4.1 we prove that training the context embeddings with a single gradient step suffices to capture the empirical importance of tokens in the dataset; then, in Section 4.2 we characterize the implicit bias of training the $\langle \text{cls} \rangle$ embedding $p$ until convergence, having fixed the context embeddings after the first gradient step.

### 4.1   One step of gradient descent learns the importance of the tokens

We start by showing that the first step of gradient descent is already enough to give a correlation between embeddings and the output vector $v$. Furthermore, this correlation is proportional to the average signed frequency defined in (3). We initialize $v$ with any unit-norm vector and $E_s^0, p^0 \overset{\text{i.i.d.}}{\sim} \mathcal{N}(0, \frac{1}{d}I)$ for all $s \in \mathcal{S}$. Then, we perform one step of gradient descent with step size $\eta_0$ on all trainable embeddings:

$$p^1 = p^0 - \eta_0 \nabla_p \mathcal{L}(\boldsymbol{E}^0, p^0), \qquad E_s^1 = E_s^0 - \eta_0 \nabla_{E_s} \mathcal{L}(\boldsymbol{E}^0, p^0), \quad \text{for all } s \in \mathcal{S}. \tag{4}$$

**Lemma 4.1.** *For any $\delta > 0$, let*

$$d \geq \max\left\{ 256, \left( 2 \log \frac{|\mathcal{S}|^2}{\delta} \right)^2 \right\}. \tag{5}$$

*Then, after the first step of gradient descent in (4), we have that, for any $s \in \mathcal{S}$,*

$$E_s^1 = E_s^0 + \frac{\eta_0}{2} \alpha_s v + err_s, \qquad p^1 = p^0 + err_p, \tag{6}$$

*where the error terms $err_s, err_p$ are bounded with probability at least $1 - \delta$ as*

$$\max\{\max_{s \in \mathcal{S}} \|err_s\|_2, \|err_p\|_2\} \leq 11 \eta_0 d^{-\frac{1}{4}}. \tag{7}$$

Lemma 4.1 implies that after one step of training, the embedding vector $E_s$ of each token $s$ learns the empirical importance of the tokens by adding a vector in the direction of the output vector $v$ with magnitude proportional to $\alpha_s$.

The proof follows from the structure of the gradient update. In particular, it can be shown that

$$\nabla_{E_s} \mathcal{L}(\boldsymbol{E}^0, p^0) = -\widehat{\mathbb{E}}\left[ yg(X, y) \left( \sum_{i=1}^{T} (\sum_{j \neq i} (\mathbb{1}_{x_i = s} - \mathbb{1}_{x_j = s}) q_i q_j)(E_{x_i}^0)^\top v p^0 + \sum_{i=1}^{T} \mathbb{1}_{x_i = s} q_i v \right) \right], \tag{8}$$

with $q_i := \frac{\exp\left((p^0)^\top E_{x_i}^0\right)}{\sum_{j=1}^{T} \exp\left((p^0)^\top E_{x_j}^0\right)}$. Note that, for all $x_i$, $(p^0)^\top E_{x_i}^0$ is of order $1/\sqrt{d}$, due to the independent Gaussian initialization. Thus, the characterization in (8) implies that the gradient is roughly $\frac{\eta_0}{2} \alpha_s v$ plus a term that is vanishing in $d$. The full argument is deferred to Appendix C.1.

## 4.2 Gradient flow on $p$ performs max-margin token selection

Lemma 4.1 shows the informative overlap between the output vector $v$ and the context embedding vectors $E_s$ after the first step of gradient descent[2]. However, (6) also implies that the overlap between the $\langle \text{cls} \rangle$ embedding vector $p$ and $E_s$ does not improve after the first step. Thus, next, we study the training dynamics of $p$, characterizing its implicit bias. Specifically, we fix the context embedding matrix to $\boldsymbol{E}^1$ (obtained after the first gradient step) and train the $\langle \text{cls} \rangle$ embedding vector $p$ with gradient flow initialized at $p^1$ (obtained after the first gradient step):

$$\frac{\mathrm{d}}{\mathrm{d}t} p_t = -\nabla_p \mathcal{L}(\boldsymbol{E}^1, p_t). \tag{9}$$

We consider gradient flow for technical convenience, and all results in this section can be readily extended to gradient descent with small enough step size. For the rest of the section and in the related proofs appearing in the appendix, we will refer to the embeddings in $\boldsymbol{E}^1$ as $E_s$ and not $E_s^1$, omitting the superscript to favor readability.

**Max-margin token selection.** Given the $\langle \text{cls} \rangle$ embedding $p$ and a sequence $X$, we denote the set of tokens in $X$ selected by $p$ as

$$\mathcal{S}_X(p) = \{s : s = \arg\max_{s \in X} p^\top E_s\}, \tag{10}$$

and we define $\overline{\mathcal{S}_X(p)} = X \setminus \mathcal{S}_X(p)$. Intuitively, given a sequence $X$, the selected tokens in $X$ have the largest softmax weight (proportional to $\exp(p^\top E_{x_i})$). Note that, for $p' \neq p$, we may have that $\mathcal{S}_X(p') = \mathcal{S}_X(p)$ for all $X$. Thus, we define the equivalence relation

$$p \cong p' \iff \mathcal{S}_X(p) = \mathcal{S}_X(p'), \qquad \text{for all } X \in \mathcal{X}_n. \tag{11}$$

Intuitively, two vectors $p, p'$ are equivalent under the above relation if they select the same tokens for all the sequences. Given a vector $p_\circ$, we denote by $\mathcal{P}_{p_\circ}$ its equivalence class, and we define the set of max-margin directions among all vectors in $\mathcal{P}_{p_\circ}$ as

$$\mathcal{P}_*(p_\circ) = \left\{ \frac{\hat{p}}{\|\hat{p}\|_2} : \hat{p} = \arg\min_{p \in \mathcal{P}_{p_\circ}} \|p\|_2 \right.$$
$$\left. \text{s.t.} \quad p^\top (E_s - E_{s'}) \geq 1, \quad \forall s \in \mathcal{S}_X(\mathcal{P}_{p_\circ}), \; \forall s' \in \overline{\mathcal{S}_X(\mathcal{P}_{p_\circ})}, \; \forall X \in \mathcal{X}_n \right\}. \tag{12}$$

We first show in Lemma 4.2 (proved in Appendix C.2) that the max-margin problem in (12) always has a unique solution, i.e., $\mathcal{P}_*(p_\circ)$ is always a singleton. Thus, later on, we will use $\hat{p}(p_\circ)$ as the solution to (12), and $p_*(p_\circ) = \frac{\hat{p}(p_\circ)}{\|\hat{p}(p_\circ)\|_2}$. We drop the dependency on $p_\circ$ when there is no confusion.

**Lemma 4.2.** *For any $p_\circ \neq 0$, the max margin problem in (12) has a unique solution denoted as $\hat{p}$.*

**Implicit bias of gradient flow.** While Lemma 4.1 holds for any dataset, we need an extra assumption on the data to analyze the gradient flow, due to the complex loss landscape caused by softmax attention.

**Assumption 1.** Each sequence in $\mathcal{X}_n$ contains either a *single completely positive* token or a *single completely negative* token, and all remaining tokens are *irrelevant*.

Assumption 1 implies that all sequences in the dataset contain precisely one relevant token, and the relevant token also aligns with the label. We remark that datasets containing only one relevant token have been also considered in prior work, see [36, Theorem 1] and [25]. We further denote by $\mathcal{S}_c$ the set containing all completely positive and all completely negative tokens.

**Theorem 4.3.** *Under Assumption 1, for any $\delta > 0$, let*

$$\eta_0 \geq 4n^2 T^2, \quad d \geq \max\left\{ 256, \left(2 \log \frac{|\mathcal{S}|^2}{\delta}\right)^2, (88\eta_0^2 + 111\eta_0 + 2)^8, |\mathcal{S}| + 3 \right\}. \tag{13}$$

*Let $p_t$ be the solution of the gradient flow (9). Then, with probability at least $1 - \delta$, we have that $\|p_t\|_2 \to \infty$. Furthermore, assuming that $p_\infty := \lim_{t \to +\infty} \frac{p_t}{\|p_t\|_2}$ exists, the limiting direction $p_\infty$ satisfies the following properties with probability at least $1 - \delta$:*

---

[2]In Appendix F.3, we provide additional numerical evidence that learning *all* embeddings is essential. This is corroborated by the fact that the more restrictive one-layer model (1), with untrained orthogonal token embeddings, fails to catch up with its more flexible counterpart.

1. $p_\infty$ selects all completely positive and completely negative tokens, i.e., $\mathcal{S}_c \subseteq \bigcup_X \mathcal{S}_X(p_\infty)$.

2. $p_\infty$ is the max-margin direction for such a selection, i.e., $p_\infty = p_*(p_\infty)$.

Theorem 4.3 shows that, if $p_t$ converges in direction, it must converge to the max-margin direction that selects all the completely positive/negative token. A sketch of the argument is given below and the complete proof is in Appendix C.3.

*Proof sketch.* We prove the three statements separately. First, we show that $\|p_t\|_2 \to \infty$ (Lemma C.1). To do so, we explicitly construct a vector $\hat{p}$ such that $\hat{p} \nabla_p \mathcal{L}(\boldsymbol{E}^1, p) < 0$ for all $p$. This means that there is no stationary point with finite norm, which implies that the norm of the vector obtained via gradient flow diverges.

Next, we show that, if the directional limit $p_\infty$ exists, then it must select all the important tokens (Lemma C.2). To do so, we note that, after one step, the model approximately selects the important tokens. This implies that $p_t$ selects important tokens for all $t$, as gradient flow cannot increase the loss.

Finally, we show that $p_\infty$ is the max-margin solution of a feasible selection (Lemma C.6). To do so, we assume by contradiction that the directional limit is any vector $p'$ that is not the max-margin solution of a feasible selection. Then under this assumption, we prove that $\lim_{t \to \infty} \frac{p_t}{\|p_t\|_2} = p_*(p')$, which is in fact the solution of (12), thus giving a contradiction. $\qquad \square$

Before proceeding with the characterization of the max-margin solution, we highlight some differences with respect to the related works [37] and [40] whose setup is similar to ours. Theorem 3 in [37] shows that gradient descent on $p$ converges to a locally optimal max-margin solution when initialized in a regime close enough to such solution, and Theorem 4 in [37] shows that the regularization path can only converge to locally max-margin solutions. However, these results do not exclude the possibility of the gradient flow converging to directions that are *not* locally optimal and *not* the max-margin direction. In contrast, we characterize all possible directions the gradient flow converges to, showing that these are max-margin directions that select all completely positive/negative tokens. Furthermore, we do so neither starting from an initialization that is close enough to such solution, nor with only one sequence in the dataset. This requires a different proof strategy as compared to [37]. Finally, [37] does not consider any connections between the directional convergence and the importance of the tokens, which, in contrast, is our main focus.

Let us now comment on the comparison with [40]. There are two key assumptions crucially needed in [40] that do not apply to our case. More precisely, [40, Assumption 1] requires that all the tokens in non-relevant positions are nearly orthogonal to each other, which is critical in their proof. This is not true in our case, since all the irrelevant tokens appear at least in two sequences (otherwise, the token would have non-zero average signed frequency). Thus, the global convergence results in [40] cannot be applied. Furthermore, [40, Assumption 2] requires all the irrelevant tokens in the sequence to have exactly the same score $\gamma_s = y E_s^\top v$. This assumption leads to the fact that for each sequence, only the term that involves the relevant tokens contributes to the directional gradient. In our setting, even though the score differences between different irrelevant tokens are of order $1/\sqrt{d}$, these errors matter when the gradient norm is diverging to infinity, and the analysis in [40] cannot be applied.

**Characterization of the max-margin solution.** Theorem 4.3 gives that, if gradient flow converges in direction, the limiting direction is the max-margin one that selects all important tokens in each sentence. However, this does not exclude a-priori the possibility that gradient flow also selects some irrelevant tokens. To address this point, we now establish if and how many irrelevant tokens can be selected: in general, it is not possible to select all irrelevant tokens (Lemma 4.4) and, under an additional assumption, no irrelevant token is selected (Lemma 4.5).

**Lemma 4.4.** *Suppose $\hat{p}$ selects all the tokens, i.e., $\mathcal{S}_X(\hat{p}) = \mathcal{S}$. Then, $p_\infty \neq \hat{p}$.*

*Proof.* If $\hat{p}$ selects all the tokens, then $\hat{p}^\top E_{x_i} = \hat{p}^\top E_{x_j}$ for all $x_i, x_j \in X$ and for all $X \in \mathcal{X}_n$. Thus, by Lemma A.2, $\hat{p}^\top \nabla_p \mathcal{L}(\boldsymbol{E}, p) = 0$ for any $p$, which gives the desired result. $\qquad \square$

Lemma 4.4 shows that the directional limit $p_\infty$ (when it exists) cannot select all tokens and, as it selects all important ones, it must be biased towards them. As an application, consider the case where

there is only one irrelevant token in the vocabulary. Then, the combination of Theorem 4.3 and Lemma 4.4 gives that only the completely positive/negative tokens are selected by gradient flow.

Going beyond the case where there is a single irrelevant token, the result below provides a sufficient condition for gradient flow to select only important tokens.

**Lemma 4.5.** *Under Assumption 1, for any $\delta > 0$, assume that* (13) *holds. Let $\hat{p}$ be the solution of the max-margin problem* (12) *that only selects the completely positive/negative tokens, i.e.,*

$$\hat{p} = \arg\min_{p} \|p\|_2, \qquad s.t. \quad p^\top (E_{s_*^X} - E_s) \geq 1, \qquad \forall s \in X \setminus \{s_*^X\}, \ \forall X \in \mathcal{X}_n,$$

*where $s_*^X$ denotes the unique completely positive/negative token in the sequence $X$. Assume that $p_\infty := \lim_{t\to\infty} \frac{p_t}{\|p_t\|_2}$ exists and that, for any $\hat{p}'$ solving* (12) *with a different selection, $\|\hat{p}\|_2 < (1-\mu)\|\hat{p}'\|_2$ for some constant $\mu$ that does not depend $d$. Then, by further taking*

$$d \geq \left\{ \left( \frac{2816 T n^2 (1 + 2\eta_0)}{\mu} \right)^4, \left( 2\eta_0 \left( n\sqrt{2\log \frac{|\mathcal{S}|^2}{\delta}} + 44n \right) \right)^4 \right\},$$

*we have that $p_\infty = \frac{\hat{p}}{\|\hat{p}\|_2}$ with probability at least $1 - \delta$.*

The proof of Lemma 4.5 is deferred to Appendix C.4. The sufficient condition of the result above requires the max-margin direction that does not select irrelevant tokens to have a larger margin than any other max-margin solution associated to a different token selection. We expect this to be the case e.g. for datasets where all the completely positive/negative tokens have the same $\alpha_s$. In fact, given the structure of the context embeddings in (6), the max-margin solution $\hat{p}$ is expected to satisfy $\hat{p}^\top v \approx 0, \hat{p}^\top E_s \approx 1, \hat{p}^\top E_{s'} \approx 0$ for all $s \in \mathcal{S}_X(\hat{p})$ and $s' \in \overline{\mathcal{S}_X(\hat{p})}$. Since the token embeddings at initialization are approximately orthogonal to each other, $\hat{p} \approx \sum_{s \in \mathcal{S}_X(p)} E_s^0$, meaning that $\|\hat{p}\|_2 \approx \sqrt{|\mathcal{S}_X(p)|}$, which implies that the sufficient condition holds.

### 4.3 Additional generalizations of the theoretical results

**Multiclass classification.** Consider $K$-class classification, let $v_k$ be the $k$-th classifier and perform one step of gradient descent on the token embeddings. Then, after some manipulations deferred to Appendix E, we have

$$E_s^\top v_k \approx \frac{\eta_0}{T} \widehat{\mathbb{E}} \left[ \sum_{i=1}^T \frac{K-1}{K} \mathbb{1}_{y = e_k, x_i = s} - \frac{1}{K} \mathbb{1}_{y \neq e_k, x_i = s} \right], \tag{14}$$

where by using $\approx$, we hide quantities that have a vanishing order in $d$. The RHS of (14) is a generalization of the average signed frequency that indicates the importance of a token with respect to class $k$. In particular, if we divide the RHS by $(K-1)$, the first term is the co-occurrence of token $s$ with label $k$ and the second term is the average co-occurrence of token $s$ with other labels. Thus, intuitively, if a token occurs much more frequently with label $k$ than other labels, this token is more important w.r.t. label $k$ and the correlation with $v_k$ is larger. We also conduct experiments on the Yelp datasets with 4 classes (see Appendix F.5), and we find that $E_s^\top k$ is still increasing with respect to the importance of the token for each class, as for binary classification.

**Training $v$ and $E_X$.** If we train $v$ but fix the context embeddings, the implicit bias of $p$ is not affected in the limit. To see this, we compute

$$-\nabla_v \mathcal{L}(p, v) = \widehat{\mathbb{E}} \left[ yg(X, y) \sum_{i=1}^T q_i(X) E_{x_i} \right] \approx \widehat{\mathbb{E}} \left[ yg(X, y) \sum_{x_i \in \mathcal{S}_X(p)} q_i(X) E_{x_i} \right].$$

Note that, if $p$ converges in direction, it must select all relevant tokens. This means that $yE_s^\top v$ will be increasing if $s$ is relevant, and will not move a lot if $s$ is irrelevant, since it appears exactly the same number of times in positive and negative sequences. While we do not have a theoretical characterization for jointly training $E, p, v$, in our experiments the token embeddings $E$ are trained simultaneously with $p$. Thus, we expect the token selection to be the same as if we only trained $p$.

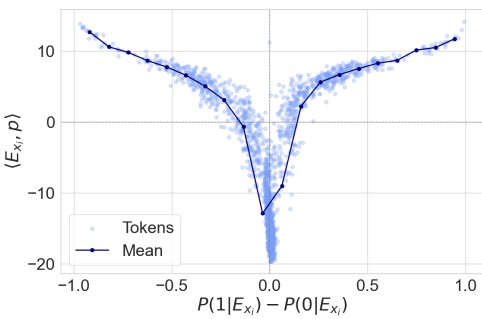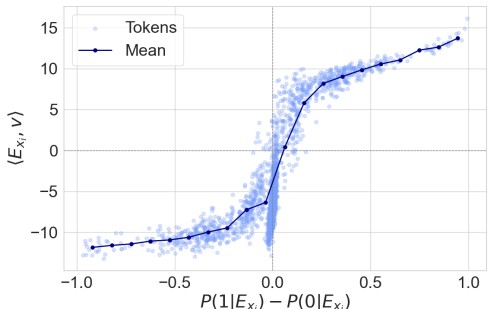

Figure 2: Dot-product of token embeddings with $\langle \text{cls} \rangle$ embedding $p$ (left) and regression coefficients $v$ (right), as a function of the token-wise difference in posterior probabilities for synthetic data sampled according to (16). We consider the one-layer attention model in (1) with all parameters trained until convergence. The point cloud around zero corresponds to the tokens in the irrelevant set.

## 5 Numerical experiments

To support our theoretical findings, we showcase the correlation of the embeddings with the $\langle \text{cls} \rangle$ embedding $p$ and the output vector $v$, having trained *all* the parameters with gradient descent until convergence. We consider different datasets (synthetic data in Figure 2; IMDB/Yelp datasets in Figures 1 and 3) and different architectures (one-layer model (1) in Figures 2 and 3; two-layer model (18) in Figure 1). Taken together, the experiments display an excellent agreement with our theory going beyond the one-layer architecture (1) and also beyond the requirements on the data-generating process. Specifically, the trained embeddings capture the importance of the corresponding tokens: the dot-product with $v$ is proportional to how frequently the token appears in positive sequences rather than in negative ones, and the dot-product with $p$ is proportional to the modulus of such frequency. We detail below the experimental design.

**Synthetic data.** Let us define the data-generating process for the synthetic experiments in Figure 2. The data is generated according to a $K$-level model. Namely, the vocabulary set $\mathcal{S}$ is partitioned as

$$\mathcal{S} = \tilde{\mathcal{S}} \cup \left\{ \mathcal{S}_k^{-1} \right\}_{k=1}^K \cup \left\{ \mathcal{S}_k^{+1} \right\}_{k=1}^K . \tag{15}$$

Here, $\tilde{\mathcal{S}}$ contains *irrelevant* tokens appearing in both positive and negative contexts with equal probability, while $\mathcal{S}_k^{+1}$ and $\mathcal{S}_k^{-1}$ (for $k \in \{1, \dots, K\}$) contain tokens appearing *mostly* in positive and negative contexts, respectively. Formally, define the importance levels $\tilde{\delta}, \delta_1, \dots, \delta_K > 0$. Then, given the sequence label $y \in \{-1, +1\}$ and $s \in \mathcal{S}$, we sample the tokens from the vocabulary as

$$p(s|y) = \begin{cases} \frac{1-\tilde{\delta}}{|\tilde{\mathcal{S}}|}, & s \in \tilde{\mathcal{S}}, \\ \frac{\tilde{\delta}(1-\delta_k)}{\sum_{k=1}^K |\mathcal{S}_k^y|}, & s \in \mathcal{S}_k^y, \\ \frac{\tilde{\delta}\delta_k}{\sum_{k=1}^K |\mathcal{S}_k^{\neg y}|}, & s \in \mathcal{S}_k^{\neg y}, \end{cases} \tag{16}$$

where $\neg$ denotes the binary inversion, i.e., $\neg(+1) = -1$ and $\neg(-1) = +1$. The law (16) implies the following posterior distribution:

$$p(y|s) = \begin{cases} 1/2, & s \in \tilde{\mathcal{S}}, \\ 1 - \delta_k, & s \in \mathcal{S}_k^y, \\ \delta_k, & s \in \mathcal{S}_k^{\neg y}. \end{cases} \tag{17}$$

From (17), it is clear that *(i)* $\tilde{\mathcal{S}}$ contains *irrelevant* tokens as the posterior is uniform, and *(ii)* $\delta_k$ quantifies the importance of the tokens in $\mathcal{S}_k^{\pm 1}$ by skewing the posterior to be $(\delta_k, 1 - \delta_k)$. For the experiments in Figure 2, we select the following hyper-parameters: $|\mathcal{S}| = 2048$, $K = 8$ and sequence length $T = 256$; $|\mathcal{S}_k^{+1}| = |\mathcal{S}_k^{-1}|$ with $|\mathcal{S}_k^{+1}| = 4 + 2^{k-1}$, and $|\tilde{\mathcal{S}}| = 964$; $\tilde{\delta} = 0.05$ and $\{\delta_k\}_{k=1}^K = \{0.45, 0.35, 0.3, 0.25, 0.2, 0.1, 0.05, 0.02\}$.

Figure 2 shows a clear separation between positive and negative tokens (right plot with the dot-product $\langle E_{x_i}, v \rangle$), and the selection mechanism ($\langle \text{cls} \rangle$ token) assigns high weights to tokens that have larger importance $\delta_k$ (left plot with the dot-product $\langle E_{x_i}, p \rangle$).

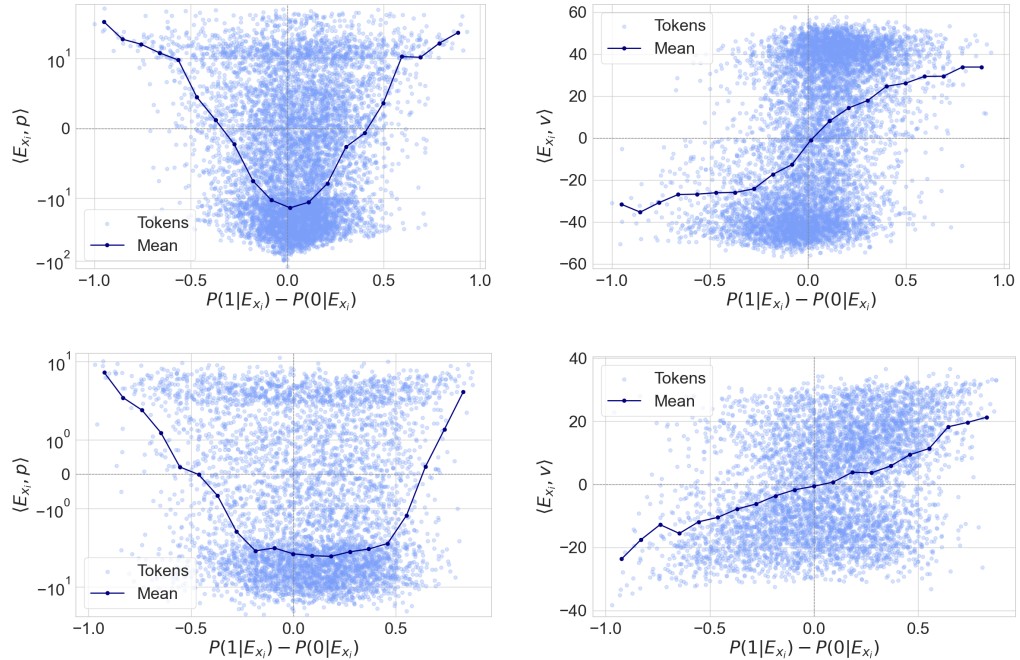

Figure 3: Dot-product of token embeddings with $\langle\text{cls}\rangle$ embedding $p$ and regression coefficients $v$, as a function of the token-wise difference in posterior for IMDB dataset (top row) and Yelp dataset (bottom row). We consider the one-layer attention model in (1) with all parameters trained until convergence.

**IMDB and Yelp datasets.** The IMDB dataset[3] consists of $50000$ reviews of average length $239$ words per review, associated to either a positive or a negative sentiment. Yelp reviews[4] provide a much larger selection. To align the data size and sequence length with the IMDB dataset, we randomly subsample a portion of the Yelp dataset constrained on the sequence length, i.e., we select reviews which have at least $1000$ and not more than $1500$ characters. In addition, Yelp reviews provide a five-star ranking, which we convert to the binary sentiment based on the following rule: $1/2$ stars reviews are assigned label $-1$; $4/5$ star reviews are assigned label $+1$; neutral reviews, i.e., $3$-star score, are removed. We adhere to a typical preprocessing pipeline for both datasets: we start by cleaning the data from punctuation symbols and omitting the stop-words, followed by an application of stemming; and we use the Bert tokenizer from Hugging Face[5] to tokenize sequences. Tokens that appear less than $50$ times are purged.

The numerical simulations for both datasets are reported in Figure 3, which displays a phenomenology similar to that obtained for synthetic data in Figure 2. We have also performed an additional experiment providing evidence that the trained embeddings indeed improve performance on the classification tasks. In particular, we have fixed context embeddings (randomly initialized) and we have only trained the $\langle\text{cls}\rangle$ token embedding and the classifier weights. The resulting test accuracy is around $72\%$ with test error around $0.6$, while for the original baseline the test accuracy is around $95\%$ with test error around $0.1$. This clearly indicates the importance of training the embeddings.

**Two-layer model.** We also consider the following two-layer model:

$$\boldsymbol{E}'_X = \text{LayerNorm}(\texttt{Softmax}(\boldsymbol{E}_X\boldsymbol{E}_X^\top)\boldsymbol{E}_X + \boldsymbol{E}_X), \quad f(X;p,\boldsymbol{E}) = \texttt{Softmax}(p^\top(\boldsymbol{E}'_X)^\top)\boldsymbol{E}'_X v, \tag{18}$$

which includes both a skip connection and the layer-norm. We note that, for both IMDB and Yelp data, the model in (18) achieves significantly smaller loss values at convergence (of the order of $10^{-5}$, in contrast to the order of $10^{-1}$ achieved by the model in (1)). However, even if this model is more

---

[3]`https://www.kaggle.com/datasets/lakshmi25npathi/imdb-dataset-of-50k-movie-reviews`
[4]`https://www.kaggle.com/datasets/yelp-dataset/yelp-dataset`
[5]`https://huggingface.co/google-bert/bert-base-uncased`

complex than the one analyzed in Section 4, the results in Figure 1 are still remarkably similar to those in Figures 2 and 3.

Finally, we note that all plots consider on the $x$-axis the difference in posterior probabilities

$$p(1|\boldsymbol{E}_{x_i}) - p(0|\boldsymbol{E}_{x_i}) = \frac{\sum_{(X,y)\in\mathcal{D}} y \sum_{i=1}^{T} \mathbb{1}_{x_i=s}}{\sum_{(X,y)\in\mathcal{D}} \sum_{i=1}^{T} \mathbb{1}_{x_i=s}} \tag{19}$$

in place of the quantity $\alpha_s$ defined in (3). In fact, while the quantity in (3) appears naturally from the analysis of gradient descent, the difference in posterior probabilities provides better visuals for real data (IMDB and Yelp). The difference between (3) and (19) lies in the normalization used: the posterior difference in (19) is the discrepancy between counts of the token $x_i$ in positive and negative sentences normalized by the total number of occurrences of $x_i$, while the quantity in (19) normalizes the discrepancy by the total number of tokens $nT$ in the datasets. For synthetic data sampled according to (16), due to the uniform nature of the sampling procedure, all tokens appear the same number of times. Thus, both quantities are the same up to a fixed scaling and, thus, they are equivalent. Additional details on the hyperparameter settings for all the experimental setups are contained in Appendix D.

**Additional numerical results.** In Appendix F.1, we consider IMDB and Yelp datasets, train one full-gradient step on the embeddings (as in the theory) and then fix the context embeddings only training the $\langle\text{cls}\rangle$ embedding as well as the output vector. As a result, the average standard deviation over the bins is reduced by roughly 8 times for the inner product with $v$ and by roughly 6 times for the inner product with $p$, compared to training all context embeddings until convergence. This gives evidence that the first step of gradient descent aligns the model with the first-order statistics of the tokens. With more training, the alignment weakens, i.e., the standard deviation grows. This is due to the fact that the dataset also contains higher-order statistics that an optimal classifier must capture; these emerge in the parameters only after additional training.

We have then performed several additional experiments to establish whether the conclusions from our theoretical analysis continue to hold in more elaborate predictive models. In Appendix F.2, we show that the presence of the MLP layer does not qualitatively change the token selection mechanism (overlap with $p$). Nevertheless, the monotonic behavior of $\langle E_{x_i}, v \rangle$ becomes less evident. In Appendix F.4, we inspect two-layer model (18) with key-query matrices $W_{KQ}^{(1)}$ and $W_{KQ}^{(2)}$ in place for both layers. We identify similar trends in overlaps with the regression vector $v$ and the $\langle\text{cls}\rangle$ token. However, the latter takes a slightly modified form. In particular, instead of correlating with $\langle E_{x_i}, p \rangle$, the correct quantity to access in this case is $p^\top W_{KQ}^{(2)} E_{x_i}$. In Appendix F.5, we observe a similar phenomenology in the multi-class case for overlaps with each class regression vector $v_k$ and a certain token-wise "one-versus-all" statistic.

## 6   Conclusions and limitations

In this paper, we study how the embedding vectors trained via gradient methods capture the importance of different tokens in the dataset. We theoretically characterize *(i)* the context embedding $E_s$ after one gradient step, and *(ii)* the implicit bias of the $\langle\text{cls}\rangle$ embedding $p$ after training with gradient flow until convergence. We conduct experiments on synthetic and realistic datasets which demonstrate the generality of our findings.

A limitation of our work is that the characterization we put forward is only in terms of the first-order statistics of the tokens (i.e., the frequencies with which they occur in the dataset), and it does not describe how the model learns the causal structure between tokens. In practice, both first-order statistics and causal structure are expected to be crucial for the model to "understand" a text. While our theory assumes a one-layer attention model, the numerical results of Figure 1 suggests that a similar qualitative picture holds more generally. This prompts us to conjecture that in deeper attention models with multiple heads, the earlier layers form induction heads [28] which learn the causal structure between tokens, and later layers perform classification based on the empirical statistics of the resulting $k$-tuples. We regard this investigation as an exciting future direction.

## Acknowledgements

DW and MM were funded in part by the Austrian Science Fund (FWF) 10.55776/COE12. For the purpose of open access, the authors have applied a CC BY public copyright license to any Author Accepted Manuscript version arising from this submission. AS and MM were partially funded by the European Union (ERC, INF2, project number 101161364). Views and opinions expressed are however those of the author(s) only and do not necessarily reflect those of the European Union or the European Research Council Executive Agency. Neither the European Union nor the granting authority can be held responsible for them. AS was also supported by the Swiss National Science Foundation under grant 204439. SO was supported by the Office of Naval Research grant N000142412289.

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

## NeurIPS Paper Checklist

1. **Claims**

   Question: Do the main claims made in the abstract and introduction accurately reflect the paper's contributions and scope?

   Answer: [Yes]

   Justification: We state the scope of the results in the abstract and introduction part.

   Guidelines:

   - The answer NA means that the abstract and introduction do not include the claims made in the paper.
   - The abstract and/or introduction should clearly state the claims made, including the contributions made in the paper and important assumptions and limitations. A No or NA answer to this question will not be perceived well by the reviewers.
   - The claims made should match theoretical and experimental results, and reflect how much the results can be expected to generalize to other settings.
   - It is fine to include aspirational goals as motivation as long as it is clear that these goals are not attained by the paper.

2. **Limitations**

   Question: Does the paper discuss the limitations of the work performed by the authors?

   Answer: [Yes]

   Justification: We discuss limitations in Section 6

   Guidelines:

   - The answer NA means that the paper has no limitation while the answer No means that the paper has limitations, but those are not discussed in the paper.
   - The authors are encouraged to create a separate "Limitations" section in their paper.
   - The paper should point out any strong assumptions and how robust the results are to violations of these assumptions (e.g., independence assumptions, noiseless settings, model well-specification, asymptotic approximations only holding locally). The authors should reflect on how these assumptions might be violated in practice and what the implications would be.
   - The authors should reflect on the scope of the claims made, e.g., if the approach was only tested on a few datasets or with a few runs. In general, empirical results often depend on implicit assumptions, which should be articulated.
   - The authors should reflect on the factors that influence the performance of the approach. For example, a facial recognition algorithm may perform poorly when image resolution is low or images are taken in low lighting. Or a speech-to-text system might not be used reliably to provide closed captions for online lectures because it fails to handle technical jargon.
   - The authors should discuss the computational efficiency of the proposed algorithms and how they scale with dataset size.
   - If applicable, the authors should discuss possible limitations of their approach to address problems of privacy and fairness.
   - While the authors might fear that complete honesty about limitations might be used by reviewers as grounds for rejection, a worse outcome might be that reviewers discover limitations that aren't acknowledged in the paper. The authors should use their best judgment and recognize that individual actions in favor of transparency play an important role in developing norms that preserve the integrity of the community. Reviewers will be specifically instructed to not penalize honesty concerning limitations.

3. **Theory assumptions and proofs**

   Question: For each theoretical result, does the paper provide the full set of assumptions and a complete (and correct) proof?

   Answer: [Yes]

   Justification: We state the assumptions in Assumption 1, and for each Theorem and Lemma, we state the assumptions under which it holds.

   Guidelines:

   - The answer NA means that the paper does not include theoretical results.
   - All the theorems, formulas, and proofs in the paper should be numbered and cross-referenced.
   - All assumptions should be clearly stated or referenced in the statement of any theorems.
   - The proofs can either appear in the main paper or the supplemental material, but if they appear in the supplemental material, the authors are encouraged to provide a short proof sketch to provide intuition.
   - Inversely, any informal proof provided in the core of the paper should be complemented by formal proofs provided in appendix or supplemental material.
   - Theorems and Lemmas that the proof relies upon should be properly referenced.

4. **Experimental result reproducibility**

   Question: Does the paper fully disclose all the information needed to reproduce the main experimental results of the paper to the extent that it affects the main claims and/or conclusions of the paper (regardless of whether the code and data are provided or not)?

Answer: [Yes]

Justification: We state all the training hyperparameters in Appendix D.

Guidelines:

- The answer NA means that the paper does not include experiments.
- If the paper includes experiments, a No answer to this question will not be perceived well by the reviewers: Making the paper reproducible is important, regardless of whether the code and data are provided or not.
- If the contribution is a dataset and/or model, the authors should describe the steps taken to make their results reproducible or verifiable.
- Depending on the contribution, reproducibility can be accomplished in various ways. For example, if the contribution is a novel architecture, describing the architecture fully might suffice, or if the contribution is a specific model and empirical evaluation, it may be necessary to either make it possible for others to replicate the model with the same dataset, or provide access to the model. In general. releasing code and data is often one good way to accomplish this, but reproducibility can also be provided via detailed instructions for how to replicate the results, access to a hosted model (e.g., in the case of a large language model), releasing of a model checkpoint, or other means that are appropriate to the research performed.
- While NeurIPS does not require releasing code, the conference does require all submissions to provide some reasonable avenue for reproducibility, which may depend on the nature of the contribution. For example
  (a) If the contribution is primarily a new algorithm, the paper should make it clear how to reproduce that algorithm.
  (b) If the contribution is primarily a new model architecture, the paper should describe the architecture clearly and fully.
  (c) If the contribution is a new model (e.g., a large language model), then there should either be a way to access this model for reproducing the results or a way to reproduce the model (e.g., with an open-source dataset or instructions for how to construct the dataset).
  (d) We recognize that reproducibility may be tricky in some cases, in which case authors are welcome to describe the particular way they provide for reproducibility. In the case of closed-source models, it may be that access to the model is limited in some way (e.g., to registered users), but it should be possible for other researchers to have some path to reproducing or verifying the results.

5. **Open access to data and code**

Question: Does the paper provide open access to the data and code, with sufficient instructions to faithfully reproduce the main experimental results, as described in supplemental material?

Answer: [No]

Justification: We do not find it necessary to release the code. Our experiments concern the training of rather standard architectures and they can be readily reproduced without needing to upload the code.

Guidelines:

- The answer NA means that paper does not include experiments requiring code.
- Please see the NeurIPS code and data submission guidelines (`https://nips.cc/public/guides/CodeSubmissionPolicy`) for more details.
- While we encourage the release of code and data, we understand that this might not be possible, so "No" is an acceptable answer. Papers cannot be rejected simply for not including code, unless this is central to the contribution (e.g., for a new open-source benchmark).
- The instructions should contain the exact command and environment needed to run to reproduce the results. See the NeurIPS code and data submission guidelines (`https://nips.cc/public/guides/CodeSubmissionPolicy`) for more details.
- The authors should provide instructions on data access and preparation, including how to access the raw data, preprocessed data, intermediate data, and generated data, etc.

- The authors should provide scripts to reproduce all experimental results for the new proposed method and baselines. If only a subset of experiments are reproducible, they should state which ones are omitted from the script and why.
- At submission time, to preserve anonymity, the authors should release anonymized versions (if applicable).
- Providing as much information as possible in supplemental material (appended to the paper) is recommended, but including URLs to data and code is permitted.

6. **Experimental setting/details**

Question: Does the paper specify all the training and test details (e.g., data splits, hyperparameters, how they were chosen, type of optimizer, etc.) necessary to understand the results?

Answer: [Yes]

Justification: We provide all hyperparameters for the experiments in D.

Guidelines:

- The answer NA means that the paper does not include experiments.
- The experimental setting should be presented in the core of the paper to a level of detail that is necessary to appreciate the results and make sense of them.
- The full details can be provided either with the code, in appendix, or as supplemental material.

7. **Experiment statistical significance**

Question: Does the paper report error bars suitably and correctly defined or other appropriate information about the statistical significance of the experiments?

Answer: [Yes]

Justification: We give the scatterplots corresponding to all embeddings (together with the average correlations).

Guidelines:

- The answer NA means that the paper does not include experiments.
- The authors should answer "Yes" if the results are accompanied by error bars, confidence intervals, or statistical significance tests, at least for the experiments that support the main claims of the paper.
- The factors of variability that the error bars are capturing should be clearly stated (for example, train/test split, initialization, random drawing of some parameter, or overall run with given experimental conditions).
- The method for calculating the error bars should be explained (closed form formula, call to a library function, bootstrap, etc.)
- The assumptions made should be given (e.g., Normally distributed errors).
- It should be clear whether the error bar is the standard deviation or the standard error of the mean.
- It is OK to report 1-sigma error bars, but one should state it. The authors should preferably report a 2-sigma error bar than state that they have a 96% CI, if the hypothesis of Normality of errors is not verified.
- For asymmetric distributions, the authors should be careful not to show in tables or figures symmetric error bars that would yield results that are out of range (e.g. negative error rates).
- If error bars are reported in tables or plots, The authors should explain in the text how they were calculated and reference the corresponding figures or tables in the text.

8. **Experiments compute resources**

Question: For each experiment, does the paper provide sufficient information on the computer resources (type of compute workers, memory, time of execution) needed to reproduce the experiments?

Answer: [No]

Justification: Our experiments are on shallow attention layers, and they do not require a large amount of computational resources.

Guidelines:

- The answer NA means that the paper does not include experiments.
- The paper should indicate the type of compute workers CPU or GPU, internal cluster, or cloud provider, including relevant memory and storage.
- The paper should provide the amount of compute required for each of the individual experimental runs as well as estimate the total compute.
- The paper should disclose whether the full research project required more compute than the experiments reported in the paper (e.g., preliminary or failed experiments that didn't make it into the paper).

9. **Code of ethics**

Question: Does the research conducted in the paper conform, in every respect, with the NeurIPS Code of Ethics https://neurips.cc/public/EthicsGuidelines?

Answer: [Yes]

Justification: We have read and conform to the NeurIPS Code of Ethics.

Guidelines:

- The answer NA means that the authors have not reviewed the NeurIPS Code of Ethics.
- If the authors answer No, they should explain the special circumstances that require a deviation from the Code of Ethics.
- The authors should make sure to preserve anonymity (e.g., if there is a special consideration due to laws or regulations in their jurisdiction).

10. **Broader impacts**

Question: Does the paper discuss both potential positive societal impacts and negative societal impacts of the work performed?

Answer: [NA]

Justification: This paper presents work whose goal is to advance the field of Machine Learning. There are many potential societal consequences of our work, none which we feel must be specifically highlighted here.

Guidelines:

- The answer NA means that there is no societal impact of the work performed.
- If the authors answer NA or No, they should explain why their work has no societal impact or why the paper does not address societal impact.
- Examples of negative societal impacts include potential malicious or unintended uses (e.g., disinformation, generating fake profiles, surveillance), fairness considerations (e.g., deployment of technologies that could make decisions that unfairly impact specific groups), privacy considerations, and security considerations.
- The conference expects that many papers will be foundational research and not tied to particular applications, let alone deployments. However, if there is a direct path to any negative applications, the authors should point it out. For example, it is legitimate to point out that an improvement in the quality of generative models could be used to generate deepfakes for disinformation. On the other hand, it is not needed to point out that a generic algorithm for optimizing neural networks could enable people to train models that generate Deepfakes faster.
- The authors should consider possible harms that could arise when the technology is being used as intended and functioning correctly, harms that could arise when the technology is being used as intended but gives incorrect results, and harms following from (intentional or unintentional) misuse of the technology.
- If there are negative societal impacts, the authors could also discuss possible mitigation strategies (e.g., gated release of models, providing defenses in addition to attacks, mechanisms for monitoring misuse, mechanisms to monitor how a system learns from feedback over time, improving the efficiency and accessibility of ML).

11. **Safeguards**

    Question: Does the paper describe safeguards that have been put in place for responsible release of data or models that have a high risk for misuse (e.g., pretrained language models, image generators, or scraped datasets)?

    Answer: [NA]

    Justification: Our paper does not contain safety issues.

    Guidelines:

    - The answer NA means that the paper poses no such risks.
    - Released models that have a high risk for misuse or dual-use should be released with necessary safeguards to allow for controlled use of the model, for example by requiring that users adhere to usage guidelines or restrictions to access the model or implementing safety filters.
    - Datasets that have been scraped from the Internet could pose safety risks. The authors should describe how they avoided releasing unsafe images.
    - We recognize that providing effective safeguards is challenging, and many papers do not require this, but we encourage authors to take this into account and make a best faith effort.

12. **Licenses for existing assets**

    Question: Are the creators or original owners of assets (e.g., code, data, models), used in the paper, properly credited and are the license and terms of use explicitly mentioned and properly respected?

    Answer: [Yes]

    Justification: We provide the link of the datasets we use.

    Guidelines:

    - The answer NA means that the paper does not use existing assets.
    - The authors should cite the original paper that produced the code package or dataset.
    - The authors should state which version of the asset is used and, if possible, include a URL.
    - The name of the license (e.g., CC-BY 4.0) should be included for each asset.
    - For scraped data from a particular source (e.g., website), the copyright and terms of service of that source should be provided.
    - If assets are released, the license, copyright information, and terms of use in the package should be provided. For popular datasets, `paperswithcode.com/datasets` has curated licenses for some datasets. Their licensing guide can help determine the license of a dataset.
    - For existing datasets that are re-packaged, both the original license and the license of the derived asset (if it has changed) should be provided.
    - If this information is not available online, the authors are encouraged to reach out to the asset's creators.

13. **New assets**

    Question: Are new assets introduced in the paper well documented and is the documentation provided alongside the assets?

    Answer: [NA]

    Justification: We do not introduce any new assets.

    Guidelines:

    - The answer NA means that the paper does not release new assets.
    - Researchers should communicate the details of the dataset/code/model as part of their submissions via structured templates. This includes details about training, license, limitations, etc.
    - The paper should discuss whether and how consent was obtained from people whose asset is used.

- At submission time, remember to anonymize your assets (if applicable). You can either create an anonymized URL or include an anonymized zip file.

14. **Crowdsourcing and research with human subjects**

    Question: For crowdsourcing experiments and research with human subjects, does the paper include the full text of instructions given to participants and screenshots, if applicable, as well as details about compensation (if any)?

    Answer: [NA]

    Justification: This paper does not involve crowdsourcing nor research with human subjects.

    Guidelines:

    - The answer NA means that the paper does not involve crowdsourcing nor research with human subjects.
    - Including this information in the supplemental material is fine, but if the main contribution of the paper involves human subjects, then as much detail as possible should be included in the main paper.
    - According to the NeurIPS Code of Ethics, workers involved in data collection, curation, or other labor should be paid at least the minimum wage in the country of the data collector.

15. **Institutional review board (IRB) approvals or equivalent for research with human subjects**

    Question: Does the paper describe potential risks incurred by study participants, whether such risks were disclosed to the subjects, and whether Institutional Review Board (IRB) approvals (or an equivalent approval/review based on the requirements of your country or institution) were obtained?

    Answer: [NA]

    Justification: This paper does not involve crowdsourcing nor research with human subjects.

    Guidelines:

    - The answer NA means that the paper does not involve crowdsourcing nor research with human subjects.
    - Depending on the country in which research is conducted, IRB approval (or equivalent) may be required for any human subjects research. If you obtained IRB approval, you should clearly state this in the paper.
    - We recognize that the procedures for this may vary significantly between institutions and locations, and we expect authors to adhere to the NeurIPS Code of Ethics and the guidelines for their institution.
    - For initial submissions, do not include any information that would break anonymity (if applicable), such as the institution conducting the review.

16. **Declaration of LLM usage**

    Question: Does the paper describe the usage of LLMs if it is an important, original, or non-standard component of the core methods in this research? Note that if the LLM is used only for writing, editing, or formatting purposes and does not impact the core methodology, scientific rigorousness, or originality of the research, declaration is not required.

    Answer: [NA]

    Justification: The core results in this paper does not involve LLMs as any important, original, or non-standard components.

    Guidelines:

    - The answer NA means that the core method development in this research does not involve LLMs as any important, original, or non-standard components.
    - Please refer to our LLM policy (https://neurips.cc/Conferences/2025/LLM) for what should or should not be described.

**Additional notation.** Throughout the appendices, to simplify the notation, we write

$$a_i(X) := p^\top E_{x_i}, \qquad q_i(X) := \frac{\exp(a_i(X))}{\sum_{j=1}^T \exp(a_j(X))}, \tag{20}$$

so that $f(X; p, \boldsymbol{E}) = \sum_{i=1}^T q_i(X) E_{x_i}^\top v$. We will drop the dependence on $X$ in $a_i(X), q_i(X)$ when there is no confusion. We also denote

$$\gamma_i(X, y) := y E_{x_i}^\top v, \tag{21}$$

dropping again the dependency on $X, y$ when there is no confusion. Finally, we define

$$g(X, y) := \frac{1}{1 + \exp(y f(X; p, \boldsymbol{E}))}. \tag{22}$$

**Properties of initialization.** By standard concentration inequalities, with probability at least $1 - \delta$, at initialization we have

$$\max\left\{ \max_{s \neq s' \in \mathcal{S}} |E_s^\top E_{s'}|, \max_{s \in \mathcal{S}} |E_s^\top v|, \max_{s \in \mathcal{S}} |E_s^\top p|, |p^\top v| \right\} \leq \frac{1}{\sqrt{d}} \sqrt{2 \log \frac{|\mathcal{S}|^2}{\delta}},$$

$$\max\left\{ \max_{s \in \mathcal{S}} \|E_s\|_2, \|p\|_2 \right\} \leq 2, \qquad \min_{s \in \mathcal{S}} \|E_s\|_2 \geq \frac{1}{2}. \tag{23}$$

For all results of the paper holding with probability at least $1 - \delta$, we will be implicitly conditioning on (23).

## A Technical lemmas

**Lemma A.1.** *The gradients of the empirical loss are given by*

$$\nabla_{E_s} \mathcal{L}(\boldsymbol{E}, p) = -\widehat{\mathbb{E}} \left[ y g(X, y) \left( \sum_{i=1}^T \left( \sum_{j \neq i} (\mathbb{1}_{x_i = s} - \mathbb{1}_{x_j = s}) q_i(X) q_j(X) \right) E_{x_i}^\top v p + \sum_{i=1}^T \mathbb{1}_{x_i = s} q_i v \right) \right],$$

$$\nabla_p \mathcal{L}(\boldsymbol{E}, p) = -\widehat{\mathbb{E}} \left[ y g(X, y) \left( \sum_{i=1}^T \left( \sum_{j \neq i} q_i(X) q_j(X) (E_{x_i} - E_{x_j}) \right) E_{x_i}^\top v \right) \right],$$

*where we have defined $g(X, y) = \frac{1}{1 + \exp(y f(X))}$.*

*Proof.* We start by taking the gradient of $q_i$ as

$$\nabla_{E_s} q_i(X) = \frac{\mathbb{1}_{x_i = s} \exp\left( E_{x_i}^\top p \right) p \left( \sum_{j=1}^T \exp\left( E_{x_j}^\top p \right) \right) - \left( \sum_{j=1}^T \mathbb{1}_{x_j = s} \exp\left( E_{x_j}^\top p \right) p \right) \exp(E_{x_i}^\top p)}{\left( \sum_{j=1}^T \exp\left( E_{x_j}^\top p \right) \right)^2}$$

$$= \frac{p \sum_{j=1}^T (\mathbb{1}_{x_i = s} - \mathbb{1}_{x_j = s}) \exp\left( E_{x_j}^\top p \right) \exp\left( E_{x_i}^\top p \right)}{\left( \sum_{j=1}^T \exp\left( E_{x_j}^\top p \right) \right)^2}$$

$$= p \left( \sum_{j=1}^T (\mathbb{1}_{x_i = s} - \mathbb{1}_{x_j = s}) q_i q_j \right)$$

$$= p \left( \sum_{j \neq i} (\mathbb{1}_{x_i = s} - \mathbb{1}_{x_j = s}) q_i q_j \right),$$

$$\nabla_p q_i(X) = \frac{\left(\exp(E_{x_i}^\top p) E_{x_i}\right)\left(\sum_{j=1}^T \exp\left(E_{x_j}^\top p\right)\right) - \sum_{j=1}^T \exp\left(E_{x_j}^\top p\right) E_{x_j} \exp(E_{x_i}^\top p)}{\left(\sum_{j=1}^T \exp\left(E_{x_j}^\top p\right)\right)^2}$$

$$= \frac{\sum_{j=1}^T \exp\left(E_{x_j}^\top p\right) \exp(E_{x_i}^\top p)(E_{x_i} - E_{x_j})}{\left(\sum_{j=1}^T \exp\left(E_{x_j}^\top p\right)\right)^2}$$

$$= \sum_{j=1}^T q_i q_j (E_{x_i} - E_{x_j})$$

$$= \sum_{j \neq i} q_i q_j (E_{x_i} - E_{x_j}).$$

Next, we look at the gradient of $f(X; p, \boldsymbol{E})$:

$$\nabla_{E_s} f(X; p, \boldsymbol{E}) = \sum_{i=1}^T \left(\nabla_{E_s} q_i\right) E_{x_i}^\top v + \sum_{i=1}^T \mathbb{1}_{x_i = s} q_i v$$

$$= \sum_{i=1}^T \left(\sum_{j \neq i} (\mathbb{1}_{x_i = s} - \mathbb{1}_{x_j = s}) q_i q_j\right) E_{x_i}^\top v p + \sum_{i=1}^T \mathbb{1}_{x_i = s} q_i v,$$

$$\nabla_p f(X; p, \boldsymbol{E}) = \sum_{i=1}^T \left(\sum_{j \neq i} q_i q_j (E_{x_i} - E_{x_j})\right) E_{x_i}^\top v.$$

This allows us to conclude that

$$\nabla_{E_s} \mathcal{L}(\boldsymbol{E}, p) = \widehat{\mathbb{E}}\left[\frac{-y}{1 + \exp(y f(X; p, \boldsymbol{E}))} \nabla_{E_s} f(X; p, \boldsymbol{E})\right]$$

$$= \widehat{\mathbb{E}}\left[\frac{-y}{1 + \exp(y f(X; p, \boldsymbol{E}))} \left(\sum_{i=1}^T \left(\sum_{j \neq i} (\mathbb{1}_{x_i = s} - \mathbb{1}_{x_j = s}) q_i(X) q_j(X)\right) E_{x_i}^\top v p \right.\right.$$

$$\left.\left. + \sum_{i=1}^T \mathbb{1}_{x_i = s} q_i(X) v\right)\right],$$

$$\nabla_p \mathcal{L}(\boldsymbol{E}, p) = \widehat{\mathbb{E}}\left[\frac{-y}{1 + \exp(y f(X; p, \boldsymbol{E}))} \nabla_p f(X; p, \boldsymbol{E})\right]$$

$$= \widehat{\mathbb{E}}\left[\frac{-y}{1 + \exp(y f(X; p, \boldsymbol{E}))} \left(\sum_{i=1}^T \left(\sum_{j \neq i} q_i(X) q_j(X)(E_{x_i} - E_{x_j})\right) E_{x_i}^\top v\right)\right],$$

thus concluding the proof. □

**Lemma A.2.** *For any vector $\widehat{p}$, we have*

$$-\widehat{p}^\top \nabla_p \mathcal{L}(\boldsymbol{E}, p) = \widehat{\mathbb{E}}\left[g(X, y)\left(\sum_{i=1}^T \sum_{j > i} (\widehat{a}_i(X) - \widehat{a}_j(X)) q_i(X) q_j(X)(\gamma_i(X, y) - \gamma_j(X, y))\right)\right],$$

*where $\widehat{a}_i = \widehat{p}^\top E_{x_i}$ for all $i \in \{1, \ldots, T\}$.*

*Proof.* From Lemma A.1, we have

$$\nabla_p \mathcal{L}(\boldsymbol{E}, p) = -\widehat{\mathbb{E}}\left[y g(X, y)\left(\sum_{i=1}^T \left(\sum_{j \neq i} q_i(X) q_j(X)(E_{x_i} - E_{x_j})\right) E_{x_i}^\top v\right)\right]$$

$$= -\widehat{\mathbb{E}}\left[g(X, y)\left(\sum_{i=1}^T \left(\sum_{j \neq i} q_i(X) q_j(X)(E_{x_i} - E_{x_j})\right) \gamma_i(X, y)\right)\right]$$

$$= -\widehat{\mathbb{E}}\left[g(X, y) \boldsymbol{E}_X^\top \left(\text{Diag}(q_X) - q_X q_X^\top\right) \gamma(X, y)\right],$$

where $q_X = [q_1(X), \ldots, q_T(X)]^\top, \gamma(X, y) = [\gamma_1(X, y), \ldots \gamma_T(X, y)]^\top$ and $\mathrm{Diag}(q_X)$ denotes the diagonal matrix with $[\mathrm{Diag}(q_X)]_{i,i} = q_i(X)$.

Thus, letting $\widehat{a} = [\widehat{a}_1, \ldots, \widehat{a}_T] \in \mathbb{R}^T$ with $\widehat{a}_i = \widehat{p}^\top E_{x_i}$, we have

$$
\begin{aligned}
-\widehat{p}^\top \nabla_p \mathcal{L}(\boldsymbol{E}, p) &= \widehat{\mathbb{E}} \left[ g(X, y) \widehat{p}^\top \boldsymbol{E}_X^\top (\mathrm{Diag}(q_X) - q_X q_X^\top) \gamma(X, y) \right] \\
&= \widehat{\mathbb{E}} \left[ g(X, y) \widehat{a}^\top (\mathrm{Diag}(q_X) - q_X q_X^\top) \gamma(X, y) \right] \\
&= \widehat{\mathbb{E}} \left[ g(X, y) \left( \sum_{i=1}^T \widehat{a}_i q_i (1 - q_i) \gamma_i - \sum_{i=1}^T \sum_{j \neq i} \widehat{a}_i q_i q_j \gamma_j \right) \right] \\
&= \widehat{\mathbb{E}} \left[ g(X, y) \left( \sum_{i=1}^T \sum_{j \neq i} \widehat{a}_i q_i q_j (\gamma_i - \gamma_j) \right) \right] \quad (\text{use } 1 - q_i = \sum_{j \neq i} q_j) \\
&= \widehat{\mathbb{E}} \left[ g(X, y) \left( \frac{1}{2} \sum_{i=1}^T \sum_{j \neq i} \widehat{a}_i q_i q_j (\gamma_i - \gamma_j) + \frac{1}{2} \sum_{j=1}^T \sum_{i \neq j} \widehat{a}_j q_i q_j (\gamma_j - \gamma_i) \right) \right] \\
&= \widehat{\mathbb{E}} \left[ g(X, y) \left( \frac{1}{2} \sum_{i=1}^T \sum_{j \neq i} (\widehat{a}_i - \widehat{a}_j) q_i q_j (\gamma_i - \gamma_j) \right) \right] \\
&= \widehat{\mathbb{E}} \left[ g(X, y) \left( \sum_{i=1}^T \sum_{j > i} (\widehat{a}_i - \widehat{a}_j) q_i q_j (\gamma_i - \gamma_j) \right) \right].
\end{aligned}
$$

$\square$

**Lemma A.3** (Convergence lemma). *Let $\|p_t\|_2 \to \infty$ and suppose there exists $\widehat{p}$ such that, for any $\epsilon > 0$, there is a $\bar{t}(\epsilon)$ ensuring*

$$
-\frac{\widehat{p}^\top}{\|\widehat{p}\|_2} \nabla_p \mathcal{L}(\boldsymbol{E}, p_t) \geq -(1 - \epsilon) \frac{p_t^\top}{\|p_t\|_2} \nabla_p \mathcal{L}(\boldsymbol{E}, p_t), \qquad \text{for all } t \geq \bar{t}(\epsilon). \tag{24}
$$

*Then, if $\lim_{t \to \infty} \frac{p_t}{\|p_t\|_2}$ exists, we have*

$$
\lim_{t \to \infty} \frac{p_t}{\|p_t\|_2} = \frac{\widehat{p}}{\|\widehat{p}\|_2}.
$$

*Proof.* By the definition of the gradient flow, (24) is equivalent to

$$
\frac{\widehat{p}^\top}{\|\widehat{p}\|_2} \frac{\mathrm{d}p_t}{\mathrm{d}t} \geq (1 - \epsilon) \frac{p_t^\top}{\|p_t\|_2} \frac{\mathrm{d}p_t}{\mathrm{d}t}.
$$

We note that

$$
\frac{p_t^\top}{\|p_t\|_2} \frac{\mathrm{d}p_t}{\mathrm{d}t} = \frac{\mathrm{d}}{\mathrm{d}t} \|p_t\|_2.
$$

Thus, by integrating both sides from $[\bar{t}(\epsilon), t]$, we have:

$$
\frac{\widehat{p}^\top}{\|\widehat{p}\|_2} (p_t - p_{\bar{t}(\epsilon)}) \geq (1 - \epsilon)(\|p_t\|_2 - \|p_{\bar{t}(\epsilon)}\|_2),
$$

which gives

$$
\frac{\widehat{p}^\top p_t}{\|\widehat{p}\|_2 \|p_t\|_2} \geq (1 - \epsilon) - (1 - \epsilon) \frac{\|p_{\bar{t}(\epsilon)}\|_2}{\|p_t\|_2} + \frac{\widehat{p}^\top p_{\bar{t}(\epsilon)}}{\|\widehat{p}\|_2 \|p_t\|_2}.
$$

Since $p_{\bar{t}(\epsilon)}, \widehat{p}$ have finite norm for fixed $\epsilon$, by taking the limit on both sides, we have

$$
\liminf_{t \to \infty} \frac{\widehat{p}^\top p_t}{\|\widehat{p}\|_2 \|p_t\|_2} \geq 1 - \epsilon.
$$

As we assume that $\lim_{t \to \infty} \frac{p_t}{\|p_t\|_2}$ exist and the above argument holds for any $\epsilon$, we conclude

$$\lim_{t \to \infty} \frac{p_t}{\|p_t\|_2} = \frac{\hat{p}}{\|\hat{p}\|_2}.$$

$\square$

**Lemma A.4.** *Given a sequence $X$, model parameters $\mathbf{E}, p, v$, and indices $i_*, j$ s.t. $x_{i_*} \in \mathcal{S}_X(p), x_j \in X \setminus \mathcal{S}_X(p)$, the following results hold.*

1. *We have*
$$\frac{1}{T} \leq q_{i_*} \leq 1.$$

2. *If there exist $\tau > 0$ such that $p^\top(E_{x_{i_*}} - E_{x_j}) \geq \tau$ for all $x_{i_*} \in \mathcal{S}_X(p)$, then we have*
$$q_j \leq \frac{1}{1 + \exp(\tau)}.$$

3. *If there exist $\tau > 0$ such that $p^\top(E_{x_{i_*}} - E_{x_j}) \leq \tau$ for all $x_{i_*} \in \mathcal{S}_X(p)$, then we have*
$$q_j \geq \frac{1}{T \exp(\tau)}.$$

*Proof.* The upper bound on $q_{i_*}$ is trivial. For the lower bound:

$$\begin{aligned}
q_{i_*} &= \frac{\exp(p^\top E_{x_{i_*}})}{\exp(p^\top E_{x_{i_*}}) + \sum_{j \neq i_*} \exp(p^\top E_{x_j})} \\
&\geq \frac{\exp(p^\top E_{x_{i_*}})}{T \exp(p^\top E_{x_{i_*}})} = \frac{1}{T}.
\end{aligned}$$

If there exists $\tau > 0$ such that $p^\top(E_{x_{i_*}} - E_{x_j}) \geq \tau$ for all $x_i \in \mathcal{S}_X(p)$, then we have

$$\begin{aligned}
q_j &= \frac{1}{1 + \sum_{i \neq j} \exp(p^\top(E_{x_i} - E_{x_j}))} \\
&\leq \frac{1}{1 + \exp(p^\top(E_{x_{i_*}} - E_{x_j}))} \\
&\leq \frac{1}{1 + \exp(\tau)}.
\end{aligned}$$

If there exists $\tau > 0$ such that $p^\top(E_{x_{i_*}} - E_{x_j}) \leq \tau$ for all $x_{i_*} \in \mathcal{S}_X(p)$, then we have

$$\begin{aligned}
q_{i_*} &= \frac{1}{1 + \sum_{i \neq j} \exp(p^\top(E_{x_i} - E_{x_j}))} \\
&\geq \frac{1}{1 + (T-1)\exp(p^\top(E_{x_{i_*}} - E_{x_j}))} \quad \text{(by definition of } \mathcal{S}_X(p)\text{)} \\
&\geq \frac{1}{T \exp(\tau)}.
\end{aligned}$$

$\square$

# B  Properties after the first gradient step

**Lemma B.1** (Boundedness of the embeddings). *For any $\delta > 0$, let*

$$d \geq \max\left\{256, \left(2\log\frac{|\mathcal{S}|^2}{\delta}\right)^2\right\},$$

*then with probability at least $1 - \delta$,*

$$\max_{s \in \mathcal{S}} \|E_s^1\|_2 \leq 2(1 + 2\eta_0), \qquad \|p^1\|_2 \leq 2 + 11\eta_0 d^{-\frac{1}{4}}.$$

*Proof.* By using (23), we have that

$$\max_{s \in \mathcal{S}} \|E_s^1\|_2 \leq \max_s \left( \|E_s^0\|_2 + \frac{\eta_0}{2} \|v\|_2 + \|err_s\|_2 \right)$$

$$\leq \max_{s \in \mathcal{S}} \left( 2 + \frac{\eta_0}{2} + 11\eta_0 d^{-\frac{1}{4}} \right)$$

$$\leq 2 + 4\eta_0,$$

and that

$$\|p^1\|_2 \leq \|p^0\|_2 + \|err_p\|_2 \leq 2 + 11\eta_0 d^{-\frac{1}{4}}. \tag{25}$$

$\square$

**Lemma B.2** (Upper bound on the loss). *For any $\delta > 0$, let*

$$d \geq \max \left\{ 256, \left( 2 \log \frac{|\mathcal{S}|^2}{\delta} \right)^2, (88\eta_0^2 + 111\eta_0 + 2)^8 \right\},$$

*then with probability at least $1 - \delta$,*

$$\mathcal{L}(\boldsymbol{E}^1, p^1) \leq \widehat{\mathbb{E}} \left[ \log \left( 1 + \exp \left( -\frac{1}{T} \sum_{i=1}^T \frac{\eta_0}{2} y \alpha_{x_i} + \frac{1}{22\eta_0} \right) \right) \right].$$

*Proof.* We first lower bound $yf(X; p, \boldsymbol{E})$ for each pair $X, y$. After the first step, we have

$$\max_{s,s'} |(p^1)^\top (E_s^1 - E_{s'}^1)| = \max_{s,s'} \Big| (p^0)^\top (E_s^0 - E_{s'}^0) + \frac{\eta_0}{2} (\alpha_s - \alpha_{s'})(p^0)^\top v$$

$$+ err_p^\top (E_s^1 - E_{s'}^1) + (err_s - err_{s'})^\top p^1 \Big|.$$

We bound each term separately:

$$\max_{s,s'} |(p^0)^\top (E_s^0 - E_{s'}^0)| \leq 2 \max_s |(p^0)^\top E_s^0| \leq 2d^{-\frac{1}{4}},$$

$$\frac{\eta_0}{2} (\alpha_s - \alpha_{s'})|(p^0)^\top v| \leq \eta_0 |(p^0)^\top v| \leq \eta_0 d^{-\frac{1}{4}},$$

$$|err_p^\top (E_s^1 - E_{s'}^1)| \leq \|err_p^\top\|_2 \|E_s^1 - E_{s'}^1\|_2 \leq 44\eta_0 d^{-\frac{1}{4}} (1 + 2\eta_0),$$

$$|(err_s - err_{s'})^\top p^1| \leq 2\|p^1\|_2 \max_s \|err_s\|_2 \leq 22\eta_0 d^{-\frac{1}{4}} \left( 2 + 11\eta_0 d^{-\frac{1}{4}} \right),$$

where we have used (23). By picking $d \geq (88\eta_0^2 + 111\eta_0 + 2)^8$, we get $\max_{s,s'} |(p^1)^\top (E_s^1 - E_{s'}^1)| \leq d^{-\frac{1}{8}}$, which implies that, for any $X$ and any $i \in \{1, \ldots, T\}$,

$$\frac{1}{T} - \frac{2d^{-\frac{1}{8}}}{T} \leq q_i(X) \leq \frac{1}{T} + \frac{2d^{-\frac{1}{8}}}{T}.$$

Thus, we lower bound $yf(X; p, \boldsymbol{E})$ for each pair $(X, y)$ as

$$yf(X; p, \boldsymbol{E}) = \sum_{i=1}^T q_i(X) \gamma_i(X)$$

$$\geq \frac{1}{T} \sum_{i=1}^T \frac{\eta_0}{2} y \alpha_{x_i} - \sum_{i=1}^T \frac{2d^{-\frac{1}{8}}}{T} \frac{\eta_0}{2} \alpha_{x_i} + \sum_{i=1}^T y q_i(X) v^\top (E_{x_i}^0 + err_{x_i})$$

$$\geq \frac{1}{T} \sum_{i=1}^T \frac{\eta_0}{2} y \alpha_{x_i} - d^{-\frac{1}{8}} \eta_0 - (1 + 2d^{-\frac{1}{8}}) v^\top (E_{x_i}^0 + err_{x_i})$$

$$\geq \frac{1}{T} \sum_{i=1}^T \frac{\eta_0}{2} y \alpha_{x_i} - d^{-\frac{1}{8}} \eta_0 - 3(1 + 11\eta_0) d^{-\frac{1}{4}}$$

$$\geq \frac{1}{T} \sum_{i=1}^T \frac{\eta_0}{2} y \alpha_{x_i} - \frac{1}{22\eta_0},$$

which allows us to conclude that

$$
\begin{aligned}
\mathcal{L}(\boldsymbol{E}^1, p^1) &= \widehat{\mathbb{E}}\left[\log(1 + \exp(-yf(X; p, \boldsymbol{E})))\right] \\
&\leq \widehat{\mathbb{E}}\left[\log\left(1 + \exp\left(-\frac{1}{T}\sum_{i=1}^{T}\frac{\eta_0}{2}y\alpha_{x_i} + \frac{1}{22\eta_0}\right)\right)\right].
\end{aligned}
\tag{26}
$$

$\square$

# C   Proofs for Section 4

## C.1   Proof of Lemma 4.1

For simplicity, in the proof we drop the time dependency in all the variables. By picking

$$
d \geq \left(2\log\frac{|\mathcal{S}|^2}{\delta}\right)^2,
$$

from (23) we have

$$
\max\left\{\max_{s\in\mathcal{S}}|E_s^\top v|, \max_{s\in\mathcal{S}}|E_s^\top p|, |p^\top v|\right\} \leq d^{-\frac{1}{4}},
$$

$$
\max\left\{\max_{s\in\mathcal{S}}\|E_s\|_2, \|p\|_2\right\} \leq 2.
$$

Thus, at initialization, we have that, for all $s$,

$$
\exp\left(-d^{-\frac{1}{4}}\right) \leq \exp(p^\top E_s) \leq \exp\left(d^{-\frac{1}{4}}\right),
$$

which implies that, for any sequence $X$ and any position $i$,

$$
\frac{1}{T + 2T\left(d^{-\frac{1}{4}}\right)} \leq \frac{1}{1 + (T-1)\exp\left(2d^{-\frac{1}{4}}\right)} \leq q_i(X) \leq \frac{1}{1 + (T-1)\exp\left(-2d^{-\frac{1}{4}}\right)} \leq \frac{1}{T - 2T\left(d^{-\frac{1}{4}}\right)},
$$

where we use the fact that for $z \in [-1, 1], 1 - |z| \leq \exp(z) \leq 1 + |z|$.

Furthermore, for $d > 256$ and for any sequence $(X, y)$, we have

$$
\frac{1}{T} - \frac{4d^{-\frac{1}{4}}}{T} \leq q_i(X) \leq \frac{1}{T} + \frac{4d^{-\frac{1}{4}}}{T},
$$

and

$$
-2d^{-\frac{1}{4}} \leq \frac{-Td^{-\frac{1}{4}}}{T - 2Td^{-\frac{1}{4}}} \leq yf(X; p, \boldsymbol{E}) \leq \frac{Td^{-\frac{1}{4}}}{T - 2Td^{-\frac{1}{4}}} \leq 2d^{-\frac{1}{4}}.
$$

Then,

$$
g(X, y) \leq \frac{1}{1 + \exp\left(-2d^{-\frac{1}{4}}\right)} \leq \frac{1}{2 - 2d^{-\frac{1}{4}}} \leq \frac{1}{2} + d^{-\frac{1}{4}},
$$

and similarly

$$
g(X, y) \geq \frac{1}{2} - d^{-\frac{1}{4}}.
$$

Now we look at the gradient update of the first step. By Lemma A.1, we have

$$-\nabla_{E_s}\mathcal{L}(\boldsymbol{E},p) = \widehat{\mathbb{E}}\left[yg(X,y)\left(\sum_{i=1}^{T}\left(\sum_{j\neq i}(\mathbb{1}_{x_i=s}-\mathbb{1}_{x_j=s})q_iq_j\right)E_{x_i}^{\top}vp + \sum_{i=1}^{T}\mathbb{1}_{x_i=s}q_iv\right)\right]$$

$$= \frac{1}{2T}\widehat{\mathbb{E}}\left[y\sum_{i=1}^{T}\mathbb{1}_{x_i=s}\right]v$$

$$+ \frac{1}{2}\widehat{\mathbb{E}}\left[y\sum_{i=1}^{T}\mathbb{1}_{x_i=s}\left(q_i - \frac{1}{T}\right)\right]v$$

$$+ \widehat{\mathbb{E}}\left[yg(X,y)\left(\sum_{i=1}^{T}\left(\sum_{j\neq i}(\mathbb{1}_{x_i=s}-\mathbb{1}_{x_j=s})q_iq_j\right)E_{x_i}^{\top}vp\right)\right]$$

$$+ \widehat{\mathbb{E}}\left[y\left(g(X,y) - \frac{1}{2}\right)\sum_{i=1}^{T}\mathbb{1}_{x_i=s}q_iv\right],$$

$$-\nabla_p\mathcal{L}(\boldsymbol{E},p) = \widehat{\mathbb{E}}\left[yg(X,y)\left(\sum_{i=1}^{T}\left(\sum_{j\neq i}q_iq_j(E_{x_i} - E_{x_j})E_{x_i}^{\top}v\right)\right)\right].$$

We note that

$$\frac{1}{2T}\widehat{\mathbb{E}}\left[y\sum_{i=1}^{T}\mathbb{1}_{x_i=s}\right]v = \frac{1}{2}\alpha_s v,$$

and we bound the remaining error terms.

We have that

$$\left\|\frac{1}{2}\widehat{\mathbb{E}}\left[y\sum_{i=1}^{T}\mathbb{1}_{x_i=s}\left(q_i - \frac{1}{T}\right)\right]v\right\|_2 \leq d^{-\frac{1}{4}},$$

and

$$\left\|\widehat{\mathbb{E}}\left[yg(X,y)\left(\sum_{i=1}^{T}\left(\sum_{j\neq i}(\mathbb{1}_{x_i=s}-\mathbb{1}_{x_j=s})q_iq_j\right)E_{x_i}^{\top}vp\right)\right. \right.$$

$$\left.\left. + y\left(g(X,y) - \frac{1}{2}\right)\sum_{i=1}^{T}\mathbb{1}_{x_i=s}q_iv\right]\right\|_2 \leq 10d^{-\frac{1}{4}}.$$

Furthermore, we also have that

$$\|\nabla_p\mathcal{L}(\boldsymbol{E},p)\|_2 \leq 8d^{-\frac{1}{4}}.$$

Thus, the desired claim follows.

### C.2 Proof of Lemma 4.2

*Proof.* We first show that, if (12) is feasible, then the solution is unique. Indeed, assume by contradiction that $p_1, p_2$ are two different solutions of (12). Clearly, $p_1$ and $p_2$ have the same norm, so $\frac{p_1^{\top}p_2}{\|p_1\|_2\|p_2\|_2} \neq 1$. Then, any convex combination of $p_1, p_2$ gives a feasible solution with a strictly smaller norm, which is a contradiction.

Next, we show that (12) is always feasible. To see this, by definition, there exists some $\tau$ such that

$$p_{\circ}^{\top}(E_s - E_{s'}) \geq \tau, \qquad \forall s \in \mathcal{S}_X(p_{\circ}), \; \forall s' \in \overline{\mathcal{S}_X(p_{\circ})}, \; \forall X \in \mathcal{X}_n.$$

Then, $\frac{p_{\circ}}{\tau}$ is a feasible solution of (12) which concludes the proof. $\qquad\square$

## C.3 Proof of Theorem 4.3

We prove each part separately. We first show that $\lim_{t \to \infty} \|p_t\|_2 = \infty$.

**Lemma C.1.** *Under Assumption 1, for any $\delta > 0$, by picking*

$$d \geq \max \left\{ 256, \left( 2 \log \frac{|\mathcal{S}|^2}{\delta} \right)^2, |\mathcal{S}| + 3 \right\},$$

*with probability at least $1 - \delta$, we have $\lim_{t \to \infty} \|p_t\|_2 = \infty$.*

*Proof.* It is sufficient to show that there exists a non-zero finite-norm $\widehat{p}$, such that for any finite norm $p$,

$$\widehat{p}^\top \nabla_p \mathcal{L}(\boldsymbol{E}^1, p) \neq 0.$$

Indeed, the above condition means that there is no stationary point for any finite-norm $p$. For gradient flow, we have that

$$\lim_{t \to \infty} \nabla_p \mathcal{L}(\boldsymbol{E}^1, p_t) = 0,$$

which by contradiction implies the desired result.

Now we construct such $\widehat{p}$. Since $d > |\mathcal{S}| + 2$, we have that with high-probability $\boldsymbol{E}^0$ is full row rank. Furthermore, $\boldsymbol{E}^1 = \boldsymbol{E}^0 + \boldsymbol{\Delta}$ and each row of $\boldsymbol{\Delta}$ is in the subspace generated by $v$ and $p_0$. Thus, we can pick $\widehat{p} \perp v, p_0$, so that

$$\boldsymbol{E}^1 \widehat{p} = \boldsymbol{E}^0 \widehat{p}.$$

Furthermore, for any fixed $v, p_0$, let $u_1, u_2$ be the orthogonormal basis of $span(v, p_0)$. Then, with probability 1 we also have $\boldsymbol{E}^0 \left( I - u_1 u_1^\top - u_2 u_2^\top \right)$ also has full row rank.

Without loss of generality, let $x_1$ be an important token in a positive sequence $X_k$, i.e., $\gamma_1(X_k) \geq \frac{\eta_0}{4nT}$. Then, we define $a \in \mathbb{R}^{|\mathcal{S}|}$ such that $a_1 = 1$ and $a_i = 0$ for all $i \neq 1$. Let

$$\boldsymbol{E}^0 \left( I - u_1 u_1^\top - u_2 u_2^\top \right) \bar{p} = a.$$

As $\boldsymbol{E}^0 \left( I - u_1 u_1^\top - u_2 u_2^\top \right)$ has full row rank, there exists a non-zero $\bar{p}$ that solves the above equation. By letting $\hat{p} = \left( I - u_1 u_1^\top - u_2 u_2^\top \right) \bar{p}$, we know that $\hat{p} \perp v, p_0$, and $\boldsymbol{E}^0 \hat{p} = a$. By Lemma A.2, we have that, for any $p$,

$$-\widehat{p}^\top \nabla_p \mathcal{L}(\boldsymbol{E}^1, p) = \widehat{\mathbb{E}} \left[ g(X, y) \left( \sum_{i=1}^{T} \sum_{j > i} (a_i - a_j) q_i q_j (\gamma_i - \gamma_j) \right) \right]$$

$$= g(X_k, y_k) \sum_{j > 1} q_1(X_k) q_j(X_k) \frac{\eta_0}{4nT} > 0,$$

which concludes the proof. $\square$

Next, we show that, if the directional limit exists, then it must select all completely positive/negative tokens.

**Lemma C.2.** *Under Assumption 1, for any $\delta > 0$, by picking*

$$\eta_0 \geq 4n^2 T^2, \quad d \geq \max \left\{ 256, \left( 2 \log \frac{|\mathcal{S}|^2}{\delta} \right)^2, (88\eta_0^2 + 111\eta_0 + 2)^8 \right\},$$

*with probability at least $1 - \delta$, if $p_\infty = \lim_{t \to \infty} \frac{p_t}{\|p_t\|_2}$ exists, then $p_\infty$ satisfies*

$$s_*^X \in \mathcal{S}_X(p_\infty), \qquad \text{for all } X \in \mathcal{X}_n,$$

*where $s_*^X$ denotes the unique completely positive/negative token in the sequence $X$.*

*Proof.* We prove the lemma by contradiction. W.l.o.g., assume by contradiction that there exists $X \in \mathcal{X}_n$ cointaining the important token $x_1$ s.t. $x_1 \notin \mathcal{S}_X(p_\infty)$. We show that there exists $\bar{t}$ such that, for all $t \geq \bar{t}$,

$$\mathcal{L}(\boldsymbol{E}^1, p^t) > \mathcal{L}(\boldsymbol{E}^1, p^1),$$

which contradicts the fact that the gradient flow always decreases the loss.

To see this, we first note that by the definition of $\mathcal{S}_X(p_\infty)$, there exists some $\tau > 0$ independent of $t$ such that

$$\min_{j \neq 1} p_\infty^\top (E_{x_1} - E_{x_j}) = -\tau.$$

W.l.o.g, we assume that $x_2$ is the token that achieves the minimum.

As $\lim_{t \to \infty} \|p_t\|_2 = \infty$ and $\lim_{t \to \infty} \frac{p_t}{\|p_t\|_2} = p_\infty$, we have that, for any $\mu > 0, R > 0$, there exists a large enough $\bar{t}$ such that

$$\|p_t\|_2 \geq 2R, \quad \left\| \frac{p_t}{\|p_t\|_2} - p_\infty \right\|_2 \leq \mu, \qquad \text{for all } t \geq \bar{t}.$$

Thus, we have:

$$\frac{p_t^\top}{\|p_t\|_2}(E_{x_1} - E_{x_2}) = p_\infty^\top (E_{x_1} - E_{x_2}) + \left( \frac{p_t}{\|p_t\|_2} - p_\infty \right)^\top (E_{x_1} - E_{x_2})$$

$$\leq -\tau + 2\mu(4\eta_0 + 2)^2,$$

where we have used the result of Lemma B.1. Thus, by picking $\mu = \frac{\tau}{4(4\eta_0 + 2)^2}$, we have

$$\frac{p_t^\top}{\|p_t\|_2}(E_{x_1} - E_{x_2}) \leq -\frac{\tau}{2},$$

which implies that

$$p_t^\top (E_{x_1} - E_{x_2}) \leq -\tau R.$$

Next, we upper bound $yf(X; p_t, \boldsymbol{E}^1)$. We first note that

$$\frac{q_1}{q_2} = \exp\left( p_t^\top (E_{x_1} - E_{x_2}) \right) \leq \exp(-\tau R),$$

which gives

$$q_1 \leq \exp(-\tau R).$$

Note that

$$yf(X; p_t, \boldsymbol{E}^1) = \sum_{i=1}^{T} q_i \gamma_i$$

$$\leq \exp(-\tau R)\gamma_1 + \max_{j \neq 1} \gamma_j$$

$$\leq \exp(-\tau R) \left( \frac{\eta_0}{2} + (1 + 11\eta_0)d^{-\frac{1}{4}} \right) + (1 + 11\eta_0)d^{-\frac{1}{4}}.$$

Thus, by picking $R \geq \frac{\log d}{4\tau}$, we have

$$yf(X; p_t, \boldsymbol{E}^1) \leq \left( \frac{3}{2} + \frac{23}{2}\eta_0 \right) d^{-\frac{1}{4}} \leq \frac{3}{4}d^{-\frac{1}{8}},$$

which implies a lower bound on the loss:

$$\mathcal{L}(\boldsymbol{E}^1, p_t) \geq \frac{1}{n} \log\left( 1 + \exp\left(-yf(X; p_t, \boldsymbol{E}^1)\right) \right) \geq \frac{1}{n} \log\left( 1 + \exp\left( -\frac{3}{4}d^{-\frac{1}{8}} \right) \right) \geq \frac{1}{2n}, \quad (27)$$

where we used that $d \geq 256$ in the last passage. Under Assumption 1, by Lemma 4.1, we have that $y\alpha_{x_i} \geq 1/(nT)$ if $x_i$ is either the completely positive or the completely negative token in $X$, and otherwise $y\alpha_{x_i} = 0$. Hence, given that each sequence $X$ contains a completely positive or negative token, we have that

$$\frac{1}{T} \sum_{i=1}^{T} y\alpha_{x_i} \geq \frac{1}{nT^2}.$$

As $\eta_0 > 4n^2T^2 > \sqrt{2nT^2/11}$, by applying Lemma B.2, we obtain

$$\mathcal{L}(\boldsymbol{E}^1, p_1) \leq \log\left(1 + \exp\left(-\frac{\eta_0}{4nT^2}\right)\right) \leq \log(1 + \exp(-n)) \leq \exp(-n) < \frac{1}{2n},$$

which gives a contradiction and concludes the proof. $\qquad\square$

Finally, we show that for each possible selection, if $p_t$ converges in direction, it must converge to the max-margin solution. In particular, we first prove the following lemma which gives an approximation to the directional gradient of the locally optimal selection. To do so, we define the secondary selection set and the locally optimal selection as follows:

**Definition C.3.** Given a vector $p$, for each sequence $X$, denote by $\mathcal{S}_X^2(p)$ the secondary selection set given by

$$\mathcal{S}_X^2(p) = \arg\max\{s : p^\top E_s, s \notin \mathcal{S}_X(p)\}. \tag{28}$$

We also denote by $\mathcal{S}_X^<(p)$ the set of tokens that are not chosen in the first and in the second place, i.e.,

$$\mathcal{S}_X^<(p) = X \setminus (\mathcal{S}_X(p) \bigcup \mathcal{S}_X^2(p)). \tag{29}$$

**Definition C.4.** Given a vector $p$, we say that $p$ is locally optimal if for every $(X, y)$ pair, we have

$$\sum_{i \in \mathcal{S}_X(p)} (\gamma_i(X, y) - \gamma_j(X, y)) \geq \mu > 0, \qquad \text{for all } j \in \mathcal{S}_X^2(p),$$

for some constant $\mu$ that does not depends on $p$.

In the definition above and for the rest of this appendix, to help readability, we will abuse notation by letting indices (e.g., $i, j$ above) also denote the corresponding tokens (e.g., $x_i, x_j$ above).

**Lemma C.5.** *Let $\overline{p}$ be a unit-norm vector and $p = R\overline{p}$ for some positive constant $R$. Suppose $\overline{p}$ is a locally optimal direction as defined in Definition C.4 with some $\mu$ that does not depends on $R$. Moreover, suppose there exists a constant $\tau_1$ that may depend on $\overline{p}, \eta_0, n, T, d$ but not on $R$, such that:*

$$\begin{aligned}
&\min_X\{\overline{p}^\top(E_s - E_{s'}), \forall s \in \mathcal{S}_X(p), \forall s' \in \mathcal{S}_X^2(p)\} \geq \tau_1, \\
&\min_X\{\overline{p}^\top(E_s - E_{s'}), \forall s \in \mathcal{S}_X^2(p), \forall s' \in \mathcal{S}_X^<(p)\} \geq \tau_1.
\end{aligned} \tag{30}$$

*Then, for any $\epsilon > 0$, for any $\hat{p} \cong p$ such that $\|\hat{p}\|_2$ does not depend on $R$ and*

$$\min_X\{\hat{p}^\top(E_s - E_{s'}), \forall s \in \mathcal{S}_X(\hat{p}), \forall s' \in X \setminus \mathcal{S}_X(\hat{p})\} \geq \tau_2,$$

*there exists $R$ large enough such that:*

$$-\hat{p}^\top \nabla \mathcal{L}(\boldsymbol{E}^1, p) \leq (1 + \epsilon)\widehat{\mathbb{E}}\left[\sum_{i \in \mathcal{S}_X(p)} \sum_{j \in \mathcal{S}_X^2(p)} (\widehat{a}_i(X) - \widehat{a}_j(X))h_{i,j}(X, y, p)\right],$$

$$-\hat{p}^\top \nabla \mathcal{L}(\boldsymbol{E}^1, p) \geq (1 - \epsilon)\widehat{\mathbb{E}}\left[\sum_{i \in \mathcal{S}_X(p)} \sum_{j \in \mathcal{S}_X^2(p)} (\widehat{a}_i(X) - \widehat{a}_j(X))h_{i,j}(X, y, p)\right],$$

*where $\widehat{a}_i(X) = \hat{p}^\top E_{x_i}, \widehat{a}_j(X) = \hat{p}^\top E_{x_j}$ and*

$$h_{i,j}(X, y, p) = g(X, y)q_i(X)q_j(X)(\gamma_i(X, y) - \gamma_j(X, y)).$$

*Proof.* By Lemma A.2, we can write the directional gradient as follows:

$$-\widehat{p}^\top \nabla_p \mathcal{L}(\boldsymbol{E}^1, p) = \widehat{\mathbb{E}}\left[\sum_{i=1}^{T}\sum_{j>i}(\widehat{a}_i(X) - \widehat{a}_j(X))h_{i,j}(X,y,p))\right]$$

$$= \widehat{\mathbb{E}}\left[\sum_{i\in\mathcal{S}_X(p)}\sum_{j\in\mathcal{S}_X^2(p)}(\widehat{a}_i(X) - \widehat{a}_j(X))h_{i,j}(X,y,p)\right] \tag{B0}$$

$$+ \widehat{\mathbb{E}}\left[\sum_{i\in\mathcal{S}_X(p)}\sum_{j\in\mathcal{S}_X^<(p)}(\widehat{a}_i(X) - \widehat{a}_j(X))h_{i,j}(X,y,p))\right] \tag{B1}$$

$$+ \widehat{\mathbb{E}}\left[\sum_{i\in X\setminus\mathcal{S}_X(p)}\sum_{j>i:j\in X\setminus\mathcal{S}_X(p)}(\widehat{a}_i(X) - \widehat{a}_j(X))h_{i,j}(X,y,p))\right]. \tag{B2}$$

The rest of the proof is to show that

$$-C_1\exp(-\tau_1 R)(\text{B0}) \le (\text{B1}) \le C_1\exp(-\tau_1 R)(\text{B0}),$$
$$-C_2\exp(-\tau_1 R)(\text{B0}) \le (\text{B2}) \le C_2\exp(-\tau_1 R)(\text{B0}),$$

for some $C_1, C_2 > 0$ that do not depend on $R$. Then, by taking $R$ large enough, we obtain the desired result.

First, we simplify (B0). Note that, for all $i, i_0 \in \mathcal{S}_X(p)$, we have that $\widehat{a}_i(X) = \widehat{a}_{i_0}(X)$. Hence, by switching the order of $i, j$, we obtain

$$\sum_{i\in\mathcal{S}_X(p)}\sum_{j\in\mathcal{S}_X^2(p)}(\widehat{a}_i(X) - \widehat{a}_j(X))h_{i,j}(X,y,p) = \sum_{j\in\mathcal{S}_X^2(p)}(\widehat{a}_{i_0}(X) - \widehat{a}_j(X))\sum_{i\in\mathcal{S}_X(p)}h_{i,j}(X,y,p)$$

$$= g(X,y)\sum_{j\in\mathcal{S}_X^2(p)}(\widehat{a}_{i_0}(X) - \widehat{a}_j(X))q_{i_0}(X)q_j(X)\sum_{i\in\mathcal{S}_X(p)}(\gamma_i(X,y) - \gamma_j(X,y)),$$

for any $i_0 \in \mathcal{S}_X(p)$. Since $p$ is a locally optimal direction, we have

$$\sum_{i\in\mathcal{S}_X(p)}(\gamma_i(X,y) - \gamma_j(X,y)) \ge \mu, \qquad \text{for all } j \in \mathcal{S}_X^2(p).$$

Now, we compare (B1) and (B0). By the exact same reason above, we can rewrite

$$\sum_{i\in\mathcal{S}_X(p)}\sum_{j\in\mathcal{S}_X^<(p)}(\widehat{a}_i(X) - \widehat{a}_j(X))h_{i,j}(X,y,p)$$

$$= g(X,y)\sum_{j\in\mathcal{S}_X^<(p)}(\widehat{a}_{i_0}(X) - \widehat{a}_j(X))q_{i_0}(X)q_j(X)\sum_{i\in\mathcal{S}_X(p)}(\gamma_i(X,y) - \gamma_j(X,y)),$$

for any $i_0 \in \mathcal{S}_X(p)$, and we compare to (B0) term-by-term. Namely, for any $X$, $j \in \mathcal{S}_X^2(p)$ and $k \in \mathcal{S}_X^<(p)$, we have:

$$\frac{|\widehat{a}_{i_0}(X) - \widehat{a}_k(X)|}{\widehat{a}_{i_0}(X) - \widehat{a}_j(X)} \le \frac{\|\widehat{p}\|_2 \|E_{x_{i_0}} - E_{x_j}\|_2}{\tau_2} \le \frac{2\|\widehat{p}\|_2 \max_s \|E_s\|_2}{\tau_2} := C_3, \tag{31}$$

$$\frac{q_k(X)}{q_j(X)} = \exp(a_k(X) - a_j(X)) \le \exp(-\tau_1 R), \tag{32}$$

$$\frac{\sum_{i\in\mathcal{S}_X(p)}|\gamma_i(X,y) - \gamma_k(X,y)|}{\sum_{i\in\mathcal{S}_X(p)}(\gamma_i(X,y) - \gamma_j(X,y))} \le \frac{2T\max_s|\gamma_s|}{\mu} \le \frac{2T\max_s\|E_s\|_2}{\mu} := C_4, \tag{33}$$

which implies that, for any $X$, $j \in \mathcal{S}_X^2(p)$ and $k \in \mathcal{S}_X^<(p)$,

$$|\widehat{a}_{i_0}(X) - \widehat{a}_k(X)|q_{i_0}(X)q_k(X)\sum_{i\in\mathcal{S}_X(p)}|\gamma_i(X,y) - \gamma_k(X,y)|$$

$$\le \exp(-\tau_1 R)C_3 C_4(\widehat{a}_{i_0}(X) - \widehat{a}_j(X))q_{i_0}(X)q_j(X)\sum_{i\in\mathcal{S}_X(p)}(\gamma_i(X,y) - \gamma_j(X,y)).$$

Thus, we get that:
$$|(B1)| \leq \exp(-\tau_1 R) T C_3 C_4 |(B0)|.$$

Next, we compare (B2) and (B0). Take any $i' \in X \setminus \mathcal{S}_X(p), k > i' \in X \setminus \mathcal{S}_X(p), i_0 \in \mathcal{S}_X(p), j \in \mathcal{S}_X^2(p)$. We compare
$$(\widehat{a}_{i'}(X) - \widehat{a}_k(X)) h_{i',k}(X, y, p)$$

with each term in (B1). We note that the bounds on $\frac{\widehat{a}_{i'}(X) - \widehat{a}_k(X)}{\widehat{a}_{i_0}(X) - \widehat{a}_j(X)}$ and $\frac{|\gamma_{i'}(X) - \gamma_k(X)|}{\sum_{i \in \mathcal{S}_X(p)} (\gamma_i(X,y) - \gamma_j(X,y))}$ are the same as those in (31) and (33). Furthermore,
$$\frac{q_{i'} q_k}{q_{i_0} q_j} \leq \exp(-\tau_1 R),$$

which gives that
$$|(B2)| \leq T^2 \exp(-\tau_1 R) C_3 C_4 |(B0)|,$$

thus concluding the proof.

$\square$

**Lemma C.6.** *Under Assumption 1, for any $\delta > 0$, by picking*
$$\eta_0 \geq 4n^2 T^2, \quad d \geq \max\left\{ 256, \left(2\log\frac{|\mathcal{S}|^2}{\delta}\right)^2, (88\eta_0^2 + 111\eta_0 + 2)^8 \right\},$$

*with probability $\geq 1 - \delta$ over the initialization, if $p_\infty = \lim_{t \to \infty} \frac{p_t}{\|p_t\|_2}$ exists, then $p_\infty \in \mathcal{P}_*(p_\infty)$.*

*Proof.* We prove the lemma by contradiction. We first assume that there exists $p_\infty$ such that $p_\infty \notin \mathcal{P}_*(p_\infty)$ and $p_\infty = \lim_{t \to \infty} \frac{p_t}{\|p_t\|_2}$. Then, we show that there exists $\widehat{p} \in \mathcal{P}_*(p_\infty)$ such that, for any $\epsilon > 0$, there is $\bar{t}(\epsilon)$ ensuring
$$-\frac{\widehat{p}^\top}{\|\widehat{p}\|_2} \nabla_p \mathcal{L}(\boldsymbol{E}^1, p_t) \geq -(1 - \epsilon) \frac{p_t^\top}{\|p_t\|_2} \nabla_p \mathcal{L}(\boldsymbol{E}^1, p_t), \qquad \text{for all } t \geq \bar{t}(\epsilon).$$

As a consequence, by Lemma A.3, we have that $p_\infty = \frac{\widehat{p}}{\|\widehat{p}\|_2}$, which gives a contradiction.

For the rest of the proof, we fix any $\epsilon > 0$ and denote $R = \|p_t\|_2$. We define $\overline{p_t} = \frac{p_t \|\widehat{p}\|_2}{\|p_t\|_2}$, and we equivalently show that:
$$-\widehat{p}^\top \nabla_p \mathcal{L}(\boldsymbol{E}^1, p_t) \geq -(1 - \epsilon) \overline{p_t} \nabla_p \mathcal{L}(\boldsymbol{E}^1, p_t). \tag{34}$$

To prove this, we first note that since $p_\infty \notin \mathcal{P}_*(p_\infty)$, for all $\frac{\widehat{p}}{\|\widehat{p}\|_2} \in \mathcal{P}_*(p_\infty)$, there exists $\tau_0$ independent of $R$ such that
$$\|\widehat{p} - p_\infty \|\widehat{p}\|_2\|_2 \geq \tau_0.$$

Thus, by the definition of $\mathcal{P}_*(p_\infty)$, there exists $\mathcal{X}_0 \subseteq \mathcal{X}_n$ such that for each sequence $X \in \mathcal{X}_0$, we can find a pair of indices $(i, j)$ with $i \in \mathcal{S}_X(p_\infty), j \in X \setminus \mathcal{S}_X(p_\infty)$ violating the margin, i.e.,
$$(\|\widehat{p}\|_2 p_\infty)^\top (E_{x_i} - E_{x_j}) \leq 1 - 3\tau,$$

for some $\tau < \frac{1}{6}$ that does not depend on $R$. With a slight abuse of notation, we define $\tau$ as
$$\tau = \frac{1}{3} \min\{ \min_{X \in \mathcal{X}_0} \{1 - (\|\widehat{p}\|_2 p_\infty)^\top (E_{x_i} - E_{x_j}), i \in \mathcal{S}_X(p_\infty), j \in \mathcal{S}_X^2(p_\infty)\},$$
$$\min_{X \in \mathcal{X}_n} \{(\|\widehat{p}\|_2 p_\infty)^\top (E_{x_i} - E_{x_j}), i \in \mathcal{S}_X(p_\infty), j \in \mathcal{S}_X^2(p_\infty)\},$$
$$\min_{X \in \mathcal{X}_n} \{(\|\widehat{p}\|_2 p_\infty)^\top (E_{x_i} - E_{x_j}), i \in \mathcal{S}_X^2(p_\infty), j \in \mathcal{S}_X^{\lessgtr}(p_\infty)\}\}.$$

This means that, for all $X \in \mathcal{X}_n$ and for all $(i, j)$ pairs such that $i \in \mathcal{S}_X(p_\infty), j \in \mathcal{S}_X^2(p_\infty)$, we have
$$(\|\widehat{p}\|_2 p_\infty)^\top (E_{x_i} - E_{x_j}) \geq 3\tau;$$

for all pairs $(i, j)$ such that $i \in \mathcal{S}_X^2(p_\infty), j \in \mathcal{S}_X^<(p_\infty)$, we have

$$(\|\widehat{p}\|_2 p_\infty)^\top (E_{x_i} - E_{x_j}) \geq 3\tau;$$

and for all $X \in \mathcal{X}_0, i \in \mathcal{S}_X(p_\infty), j \in \mathcal{S}_X^2(p_\infty)$, we have

$$(\|\widehat{p}\|_2 p_\infty)^\top (E_{x_i} - E_{x_j}) \leq 1 - 3\tau,$$

with some $\tau$ that does not depend on $R$.

Now, we compute the overlap with $\overline{p_t}$. For all $X$ and $(i, j)$, we have

$$\overline{p_t}^\top (E_{x_i} - E_{x_j}) = (\|\widehat{p}\|_2 p_\infty)^\top (E_{x_i} - E_{x_j}) + (\overline{p_t} - \|\widehat{p}\|_2 p_\infty)^\top (E_{x_i} - E_{x_j}).$$

We upper bound

$$|(\overline{p_t} - \|\widehat{p}\|_2 p_\infty)^\top (E_{x_i} - E_{x_j})| \leq \|\widehat{p}\|_2 \left\| \frac{p_t}{\|p_t\|_2} - p_\infty \right\|_2 \|E_{x_1} - E_{x_2}\|_2,$$

and since $\|\widehat{p}\|_2, \|E_{x_1} - E_{x_2}\|_2$ are finite, we have

$$\lim_{t \to \infty} |(\overline{p_t} - \|\widehat{p}\|_2 p_\infty)^\top (E_{x_i} - E_{x_j})| = 0.$$

Thus, we can pick $t_1$, such that for $t \geq t_1$, we have

$$|(\overline{p_t} - \|\widehat{p}\|_2 p_\infty)^\top (E_{x_i} - E_{x_j})| \leq \tau,$$

which implies that, for all $X \in \mathcal{X}_n$ and for all $(i, j)$ pairs such that $i \in \mathcal{S}_X(p_\infty), j \in \mathcal{S}_X^2(p_\infty)$, we have

$$\overline{p_t}^\top (E_{x_i} - E_{x_j}) \geq \tau;$$

for all $(i, j)$ pairs such that $i \in \mathcal{S}_X^2(p_\infty), j \in \mathcal{S}_X^<(p_\infty)$, we have

$$\overline{p_t}^\top (E_{x_i} - E_{x_j}) \geq \tau;$$

and for all $X \in \mathcal{X}_0, i \in \mathcal{S}_X(p_\infty), j \in \mathcal{S}_X^2(p_\infty)$, we have:

$$\overline{p_t}^\top (E_{x_i} - E_{x_j}) \leq 1 - \tau,$$

for some $\tau$ that does not depend on $R$.

Next, we show that $\overline{p_t}$ is a locally optimal solution as per Definition C.4. By Lemma C.1, $p_\infty$ selects all the completely positive/negative tokens. Thus, as $\overline{p_t} \cong p_\infty$, $\overline{p_t}$ also selects such tokens, the rest being irrelevant by Assumption 1. Hence, for any pair $(X, y)$ and for any $j \in X \setminus \mathcal{S}_X(p_t)$, we have:

$$\sum_{i \in \mathcal{S}_X(p_t)} (\gamma_i(X, y) - \gamma_j(X, y)) \geq \frac{\eta_0}{4nT},$$

by picking $d$ large enough (as per the hypothesis of the lemma). By construction, $\hat{p} \cong p_t$, $\|\hat{p}\|_2$ does not depends on $R$ and, moreover, for any $X$,

$$\hat{p}^\top (E_{x_i} - E_{x_j}) \geq 1, \qquad \text{for all } i \in \mathcal{S}_X(\hat{p}), \ j \in X \setminus \mathcal{S}_X(\hat{p}).$$

By applying Lemma C.5 on both $\hat{p}$ and $\overline{p_t}$, we have that for any $\epsilon_1 > 0$ there exist $t_2$ s.t. for all $t \geq \max\{t_1, t_2\}$, we have

$$-\hat{p}^\top \nabla_p \mathcal{L}(\boldsymbol{E}^1, p_t) \geq (1 - \epsilon_1) \widehat{\mathbb{E}} \left[ \sum_{i \in \mathcal{S}_X(p)} \sum_{j \in \mathcal{S}_X^2(p)} (\widehat{a}_i(X) - \widehat{a}_j(X)) h_{i,j}(X, y, p_t) \right],$$

$$-\overline{p_t}^\top \nabla_p \mathcal{L}(\boldsymbol{E}^1, p_t) \leq (1 + \epsilon_1) \widehat{\mathbb{E}} \left[ \sum_{i \in \mathcal{S}_X(p)} \sum_{j \in \mathcal{S}_X^2(p)} (\overline{a_i}(X) - \overline{a_j}(X)) h_{i,j}(X, y, p_t) \right],$$

where $\overline{a_i}(X), \overline{a_j}(X)$ are defined analogously to $\widehat{a}_i(X), \widehat{a}_j(X)$ by replacing $\hat{p}$ with $\overline{p_t}$.

Now, we further show that, for any $\epsilon_2 > 0$, there exist $t_3$ such that for all $t \geq t_3$,

$$
\widehat{\mathbb{E}}\left[\sum_{i \in \mathcal{S}_X(p)} \sum_{j \in \mathcal{S}_X^2(p)} (\overline{a_i}(X) - \overline{a_j}(X)) h_{i,j}(X, y, p_t)\right]
$$

$$
\leq (1 + \epsilon_2)\widehat{\mathbb{E}}_{X \in \mathcal{X}_0}\left[\sum_{i \in \mathcal{S}_X(p)} \sum_{j \in \mathcal{S}_X^2(p)} (\overline{a_i}(X) - \overline{a_j}(X)) h_{i,j}(X, y, p_t)\right].
$$

To see this, we use the same idea as in the proof of Lemma C.5. We can write

$$
\widehat{\mathbb{E}}\left[\sum_{i \in \mathcal{S}_X(p)} \sum_{j \in \mathcal{S}_X^2(p)} (\overline{a_i}(X) - \overline{a_j}(X)) h_{i,j}(X, y, p_t)\right]
$$

$$
= \widehat{\mathbb{E}}_{X \in \mathcal{X}_0}\left[\sum_{i \in \mathcal{S}_X(p)} \sum_{j \in \mathcal{S}_X^2(p)} (\overline{a_i}(X) - \overline{a_j}(X)) h_{i,j}(X, y, p_t)\right] \tag{A0}
$$

$$
+ \widehat{\mathbb{E}}_{X' \in \mathcal{X}_n \setminus \mathcal{X}_0}\left[\sum_{i \in \mathcal{S}_{X'}(p)} \sum_{j \in \mathcal{S}_{X'}^2(p)} (\overline{a_i}(X') - \overline{a_j}(X')) h_{i,j}(X', y', p_t)\right], \tag{A1}
$$

and it is sufficient to show that

$$
\text{(A1)} \leq \epsilon_2 \text{(A0)}.
$$

To prove this, we compare term-by-term. Let $X \in \mathcal{X}_0, X' \in \mathcal{X}_n \setminus \mathcal{X}_0, j \in \mathcal{S}_X^2(p_t), j' \in \mathcal{S}_X^2(p_t)$, and recall that:

$$
\sum_{i \in \mathcal{S}_X(p_t)} (\overline{a_i}(X) - \overline{a_j}(X)) h_{i,j}(X, y, p_t)
$$

$$
= g(X, y)(\overline{a_{i_0}}(X) - \overline{a_j}(X)) q_{i_0}(X) q_j(X) \sum_{i \in \mathcal{S}_X(p_t)} (\gamma_i(X, y) - \gamma_j(X, y)),
$$

$$
\sum_{i \in \mathcal{S}_{X'}(p_t)} (\overline{a_i}(X') - \overline{a_{j'}}(X')) h_{i,j'}(X', y', p_t)
$$

$$
= g(X', y')(\overline{a_{i_1}}(X') - \overline{a_{j'}}(X')) q_{i_1}(X') q_{j'}(X') \sum_{i \in \mathcal{S}_{X'}(p_t)} (\gamma_i(X', y') - \gamma_{j'}(X', y')),
$$

for any $i_0 \in \mathcal{S}_X(p_t), i_1 \in \mathcal{S}_{X'}(p_t)$. Note that

$$
\frac{g(X', y')}{g(X, y)} \leq \frac{\max_{X,y} g(X, y)}{\min_{X,y} g(X, y)} \leq \max_{X,y}(1 + \exp(yf(X))) \leq (1 + \exp(\eta_0)) := C_5. \tag{35}
$$

By using the same argument as in (31) and (33), we have

$$
\frac{\overline{a_{i_1}}(X') - \overline{a_{j'}}(X')}{\overline{a_{i_0}}(X) - \overline{a_j}(X)} \leq C_3,
$$

$$
\frac{\sum_{i \in \mathcal{S}_{X'}(p_t)} (\gamma_i(X', y') - \gamma_{j'}(X', y'))}{\sum_{i \in \mathcal{S}_X(p_t)} (\gamma_i(X, y) - \gamma_j(X, y))} \leq C_4.
$$

Finally, we need to upper bound:

$$
\frac{q_{i_1}(X') q_{j'}(X')}{q_{i_0}(X) q_j(X)}.
$$

We note that

$$
a_{i_1}(X') - a_{j'}(X') \geq R/\|\hat{p}\|_2,
$$

$$
a_{i_0}(X) - a_j(X) \leq (1 - \tau)R/\|\hat{p}\|_2,
$$

where $a_i(X) = p_t^\top E_{x_i}$. Thus by Lemma A.4, we have:

$$q_{i_0}(X) \geq \frac{1}{T}, \quad q_j(X) \geq \frac{1}{T \exp((1-\tau)R/\|\hat{p}\|_2)}, \quad q_{i_1}(X') \leq 1, \quad q_{j'}(X') \leq \frac{1}{\exp(R/\|\hat{p}\|_2)},$$

which implies that

$$\frac{q_{i_1}(X')q_{j'}(X')}{q_{i_0}(X)q_j(X)} \leq T^2 \exp(-\tau R/\|\hat{p}\|_2).$$

Thus, for each $X \in \mathcal{X}_0, X' \in \mathcal{X}_n \setminus \mathcal{X}_0, j \in \mathcal{S}_X^2(p_t), j' \in \mathcal{S}_X^2(p_t)$, we have

$$\sum_{i \in \mathcal{S}_{X'}(p)} (\overline{a_i}(X') - \overline{a_{j'}}(X'))h_{i,j'}(X', y', p_t) \leq C_6 \exp(-\tau R/\|\hat{p}\|_2) \sum_{i \in \mathcal{S}_X(p)} (\overline{a_i}(X) - \overline{a_j}(X))h_{i,j}(X, y, p_t).$$

Thus by picking large enough $t_3$ which gives large enough $R$, we have:

$$(A1) \leq \epsilon_2(A0).$$

This allows us to conclude that

$$-\hat{p}^\top \nabla_p \mathcal{L}(\boldsymbol{E}^1, p_t) \geq (1 - \epsilon_1)\widehat{\mathbb{E}} \left[ \sum_{i \in \mathcal{S}_X(p)} \sum_{j \in \mathcal{S}_X^2(p)} (\widehat{a}_i(X) - \widehat{a}_j(X))h_{i,j}(X, y, p_t) \right]$$

$$\geq (1 - \epsilon_1)\widehat{\mathbb{E}}_{X \in \mathcal{X}_0} \left[ \sum_{i \in \mathcal{S}_X(p)} \sum_{j \in \mathcal{S}_X^2(p)} (\widehat{a}_i(X) - \widehat{a}_j(X))h_{i,j}(X, y, p_t) \right],$$

$$-\overline{p_t}^\top \nabla_p \mathcal{L}(\boldsymbol{E}^1, p_t) \leq (1 + \epsilon_1)(1 + \epsilon_2)\widehat{\mathbb{E}}_{X \in \mathcal{X}_0} \left[ \sum_{i \in \mathcal{S}_X(p)} \sum_{j \in \mathcal{S}_X^2(p)} (\overline{a_i}(X) - \overline{a_j}(X))h_{i,j}(X, y, p_t) \right].$$

Note that, for each $X \in \mathcal{X}_0$,

$$\widehat{a}_i(X) - \widehat{a}_j(X) \geq 1, \qquad \overline{a_i}(X) - \overline{a_j}(X) \leq 1 - \tau,$$

which gives that

$$-\hat{p}^\top \nabla_p \mathcal{L}(\boldsymbol{E}^1, p_t) \geq -\frac{1 - \epsilon_1}{(1 + \epsilon_1)(1 + \epsilon_2)(1 - \tau)}\overline{p_t}^\top \nabla_p \mathcal{L}(\boldsymbol{E}^1, p_t).$$

Since $\epsilon_1, \epsilon_2$ can be arbitrarily small, the proof is complete. $\qquad \square$

## C.4 Proof of Lemma 4.5

We first prove two auxiliary lemmas showing that for any max-margin solution, at least one constraint must be tight.

**Lemma C.7.** *Consider the max-margin problem in* (12)*, and let $\hat{p}$ be the solution. Then, there exist $X \in \mathcal{X}_n, s \in \mathcal{S}_X(p), s' \in X \setminus \mathcal{S}_X(p)$ such that:*

$$\hat{p}^\top(E_s - E_{s'}) = 1.$$

*Proof.* Assume by contradiction that for all $X \in \mathcal{X}_n, s \in \mathcal{S}_X(p), s' \in X \setminus \mathcal{S}_X(p)$,

$$\hat{p}^\top(E_s - E_{s'}) > 1.$$

Define

$$\tau = \min_{X \in \mathcal{X}_n, s \in \mathcal{S}_X(p), s' \in X \setminus \mathcal{S}_X(p)} \hat{p}^\top(E_s - E_{s'}),$$

and we know that $\tau > 1$. Then, $\hat{p}' = \frac{\hat{p}}{\tau}$ is also a feasible solution to (12) with $\|\hat{p}'\|_2 < \|\hat{p}\|_2$, which contradicts to the fact that $\hat{p}$ is the solution to the max-margin problem. $\qquad \square$

**Lemma C.8.** *Under the same condition on $d$ in Lemma 4.5, let $\hat{p}$ be the unique solution of the max-margin problem in* (12) *that only selects the completely positive and negative tokens. Then we have:*

$$\|\hat{p}\|_2 \leq 4n.$$

*Proof.* By the definition of the max-margin problem, it is sufficient to construct a feasible $p$ with $\|p\|_2 \leq 4n$, which implies that:
$$\|\hat{p}\|_2 \leq \|p\|_2 \leq 4n.$$

Since $\hat{p}$ only selects completely positive and negative tokens, we know that $(\bigcup_X \mathcal{S}_X(\hat{p})) \bigcap (\bigcup_X \overline{\mathcal{S}_X(\hat{p})}) = \emptyset$. We let

$$p = 2 \sum_{s \in \bigcup_X \mathcal{S}_X(\hat{p})} E_s^0,$$

and by (23) we know that $\|p\|_2 \leq 4n$, since there are at most $n$ completely positive and negative tokens.

It remains to show that $p$ is feasible with high probability. To see this, for $s \in \bigcup_X \mathcal{S}_X(\hat{p})$, we have

$$p^\top E_s^1 = 2 + \sum_{s' \in \mathcal{S}_X(\hat{p}):s' \neq s} (E_s^0)^\top E_{s'}^0 + \frac{\eta_0}{2} \alpha_s p^\top v + p^\top \mathrm{err}_s$$

$$\geq 2 - \frac{\eta_0}{2} \alpha_s \frac{n}{\sqrt{d}} \sqrt{2 \log \frac{|\mathcal{S}|^2}{\delta}} - 22\sqrt{2n}\eta_0 d^{-\frac{1}{4}},$$

$$\geq 2 - \frac{\eta_0}{2} \frac{n}{\sqrt{d}} \sqrt{2 \log \frac{|\mathcal{S}|^2}{\delta}} - 22\sqrt{2n}\eta_0 d^{-\frac{1}{4}}$$

where the first inequality follows from (23) and Lemma 4.1. Similarly, for $s \in \bigcup_X \overline{\mathcal{S}_X(\hat{p})}$,

$$p^\top E_s^1 \leq \frac{\eta_0}{2} \frac{n}{\sqrt{d}} \sqrt{2 \log \frac{|\mathcal{S}|^2}{\delta}} + 22\sqrt{2n}\eta_0 d^{-\frac{1}{4}}$$

Thus, by picking

$$d \geq \left( 2\eta_0 \left( n \sqrt{2 \log \frac{|\mathcal{S}|^2}{\delta}} + 44\sqrt{2n} \right) \right)^4,$$

we have $p^\top (E_s^1 - E_{s'}^1) \geq 1$, with $s \in \bigcup_X \mathcal{S}_X(\hat{p}), s' \in \bigcup_X \overline{\mathcal{S}_X(\hat{p})}$, which indicates the feasibility of $p$ and finishes the proof. $\square$

*Proof of Lemma 4.5.* Let $\hat{p}'$ be the max-margin solution of (12) with a different selection. By Theorem 4.3, we have that, for all $X, s_*^X \in \mathcal{S}_X(\hat{p}')$. We denote by $i_*^X$ the index of $s_*^X$. Assume by contradiction $p_\infty = \frac{\hat{p}'}{\|\hat{p}'\|_2}$. We will now show that this implies the following statement: for any $\epsilon > 0$, there is a $t(\epsilon)$ ensuring

$$-\frac{\hat{p}^\top}{\|\hat{p}\|_2} \nabla_p \mathcal{L}(\boldsymbol{E}, p_t) \geq -(1 - \epsilon) \frac{p_t^\top}{\|p_t\|_2} \nabla_p \mathcal{L}(\boldsymbol{E}, p_t), \qquad \text{for all } t \geq t(\epsilon). \tag{36}$$

Then, by Lemma A.3, we have that $p_\infty = \frac{\hat{p}}{\|\hat{p}\|_2}$, which gives a contradiction.

As in the proof of Lemma C.6, we define $\overline{p_t} = \frac{p_t}{\|p_t\|_2} \|\hat{p}\|_2$. Thus, (36) is equivalent to

$$-\hat{p}^\top \nabla_p \mathcal{L}(\boldsymbol{E}, p_t) \geq -(1 - \epsilon) \overline{p_t}^\top \nabla_p \mathcal{L}(\boldsymbol{E}, p_t).$$

First of all, since $\hat{p}, \hat{p}'$ are two max-margin solutions, by the definition of max-margin solution and Lemma C.7, we have:

$$\hat{p}^\top (E_{s_*^X} - E_s) \geq 1, \qquad \forall s \in X \setminus s_*^X, \forall X \in \mathcal{X}_n$$
$$(\hat{p}')^\top (E_s - E_{s'}) = 1, \qquad \exists X \in \mathcal{X}_n, s \in \mathcal{S}_X(\hat{p}'), s' \in X \setminus \mathcal{S}_X(\hat{p}'),$$

which implies that

$$\frac{\hat{p}'^\top \|\hat{p}\|_2}{\|\hat{p}'\|_2} (E_s - E_{s'}) = \frac{\|\hat{p}\|_2}{\|\hat{p}'\|_2} = 1 - \mu < 1, \qquad \exists X \in \mathcal{X}_n, s' \in X \setminus \mathcal{S}_X(\hat{p}'), \forall s \in \mathcal{S}_X(\hat{p}').$$

For simplicity, we define $\overline{\mathcal{S}^\circ_X(\hat{p}')} \subseteq X \setminus \mathcal{S}_X(\hat{p}')$ such that

$$(\hat{p}')^\top (E_s - E_{s'}) = 1, \qquad \forall X \in \mathcal{X}_n, s' \in \overline{\mathcal{S}^\circ_X(\hat{p}')}, s \in \mathcal{S}_X(\hat{p}'),$$

and note that $\overline{\mathcal{S}^\circ_X(\hat{p}')}$ can be empty for some $X$. Also define

$$\tau_0 = \min_{X \in \mathcal{X}_n, s \in \mathcal{S}_X(\hat{p}'), s' \in \overline{\mathcal{S}_X(\hat{p}')} \setminus \overline{\mathcal{S}^\circ_X(\hat{p}')}} (\hat{p}')^\top (E_s - E_{s'}) - 1.$$

As $\lim_{t \to \infty} \overline{p_t} = \frac{\hat{p}' \|\hat{p}\|_2}{\|\hat{p}'\|_2}$, for any $\epsilon_1 \in (0, \mu)$ small enough, there exists a $t_1$ ensuring the following for all $t \geq t_1$:

$$\overline{p_t}^\top (E_s - E_{s'}) \leq 1 - \mu + \epsilon_1 < (1-\mu)(1+\tau_0) - \epsilon_1, \qquad \forall X \in \mathcal{X}_n, s \in \mathcal{S}_X(\hat{p}'), s' \in \overline{\mathcal{S}^\circ_X(\hat{p}')},$$

$$\overline{p_t}^\top (E_s - E_{s'}) \geq (1-\mu)(1+\tau_0) - \epsilon_1, \qquad \forall X \in \mathcal{X}_n, s \in \mathcal{S}_X(\hat{p}'), s' \in \overline{\mathcal{S}_X(\hat{p}')} \setminus \overline{\mathcal{S}^\circ_X(\hat{p}')}, \tag{37}$$

which implies that $\mathcal{S}^2_X(p_t) \subseteq \overline{\mathcal{S}^\circ_X(\hat{p}')}$, if $\overline{\mathcal{S}^\circ_X(\hat{p}')} \neq \emptyset$.

By applying Lemma C.5 to $\overline{p_t}$, we obtain that, for any $\epsilon_2 > 0$, there exists a $t_2$ ensuring that, for all $t \geq t_2$,

$$-\overline{p_t}^\top \nabla_p \mathcal{L}(\boldsymbol{E}^1, p_t) \leq (1+\epsilon_2)\widehat{\mathbb{E}}\left[ g(X,y) \sum_{i \in \mathcal{S}_X(p_t)} \sum_{j \in \mathcal{S}^2_X(p_t)} (\overline{a_i}(X) - \overline{a_j}(X)q_i(X)q_j(X)(\gamma_i(X) - \gamma_j(X))) \right]$$

$$\leq (1+\epsilon_2)\widehat{\mathbb{E}}\left[ g(X,y) \sum_{i \in \mathcal{S}_X(p_t)} \sum_{j \in \mathcal{S}^2_X(p_t)} (\overline{a_i}(X) - \overline{a_j}(X))q_i(X)q_j(X)|\gamma_i(X) - \gamma_j(X)| \right] \tag{C1}$$

Now we further approximate (C1). For simplicity, we define $\mathcal{X}_\circ \in \mathcal{X}_n$ such that, for all $X \in \mathcal{X}_\circ$, $\overline{\mathcal{S}^\circ_X(\hat{p}')} \neq \emptyset$. We show that for any $\epsilon_3 > 0$, there exists $t_3$ ensuring that, for all $t \geq \max\{t_1, t_2, t_3\}$,

$$(\text{C1}) \leq (1+\epsilon_2)(1+\epsilon_3)\frac{1}{n} \sum_{X \in \mathcal{X}_\circ} g(X,y) \sum_{i \in \mathcal{S}_X(p_t)} \sum_{j \in \mathcal{S}^2_X(p_t)} (\overline{a_i}(X) - \overline{a_j}(X))q_i(X)q_j(X)|\gamma_i(X) - \gamma_j(X)|$$

To see this, for any $\epsilon_3 > 0$, we show that for $t \geq \max\{t_1, t_2, t_3\}$ and any $X \in \mathcal{X}_\circ, X' \in \mathcal{X}_n \setminus \mathcal{X}_\circ$,

$$g(X', y) \sum_{i \in \mathcal{S}_X(p_t)} \sum_{j \in \mathcal{S}^2_X(p_t)} (\overline{a_{i^{X'}_*}}(X') - \overline{a_j}(X'))q_{i^{X'}_*}(X')q_j(X')(\gamma_{i^{X'}_*}(X') - \gamma_j(X')))$$

$$\leq \frac{1}{n}\epsilon_3 g(X,y) \sum_{i \in \mathcal{S}_X(p_t)} \sum_{j \in \mathcal{S}^2_X(p_t)} (\overline{a_{i^X_*}}(X) - \overline{a_j}(X))q_{i^X_*}(X)q_j(X)(\gamma_{i^X_*}(X) - \gamma_j(X))). \tag{38}$$

Indeed, using the same methods as in (31), (33), (35), we have

$$\frac{\overline{a_i}(X') - \overline{a_j}(X')}{\overline{a_i}(X) - \overline{a_j}(X)} \leq C_3, \quad \forall i \in \mathcal{S}_X(p_t)$$

$$\frac{\gamma_i(X') - \gamma_j(X'))}{\gamma_i(X) - \gamma_j(X))} \leq C_4, \quad \forall i \in \mathcal{S}_X(p_t)$$

$$\frac{g(X', y')}{g(X, y)} \leq C_5,$$

where $C_3, C_4, C_5$ are constants that do not depend on $R$. It remains to upper bound $\frac{q_i(X')q_j(X')}{q_i(X)q_j(X)}$, which is equivalent to $\frac{q_{i^{X'}_*}(X')q_j(X')}{q_{i^X_*}(X)q_j(X)}$ as all $i \in \mathcal{S}_X(p_t)$ has the same $q_i$.

By Lemma A.4 together with (37), we have:

$$q_{i^{X'}_*}(X')q_j(X') \leq \exp(-((1-\mu)(1+\tau_0) - \epsilon_1)R),$$

$$q_{i^X_*}(X)q_j(X) \geq \frac{1}{T^2}\exp(-(1-\mu+\epsilon_1)R),$$

where for simplicity we denote $R = \|p_t\|_2$. This implies

$$\frac{q_{i_*^{X'}}(X')q_j(X')}{q_{i_*^X}(X)q_j(X)} \leq T^2 \exp(-((1-\mu)\tau_0 - 2\epsilon_1)R).$$

Since $\|p_t\|_2 \to \infty$, and $C_3, C_4, C_5$ do not depend on $R$, we can pick $t_3$ large enough such that:

$$nT^2 C_3 C_4 C_5 \exp(-((1-\mu)\tau_0 - 2\epsilon_1)R) \leq \epsilon_3,$$

which implies (38). Thus, we have

$$-\overline{p_t}^\top \nabla_p \mathcal{L}(\boldsymbol{E}^1, p_t)$$

$$\leq (1+\epsilon_2)(1+\epsilon_3)\frac{1}{n} \sum_{X \in \mathcal{X}_\circ} g(X,y) \sum_{i \in \mathcal{S}_X(p_t)} \sum_{j \in \mathcal{S}_X^2(p_t)} (\overline{a_i}(X) - \overline{a_j}(X))q_i(X)q_j(X)|\gamma_i(X) - \gamma_j(X)|$$

$$= (1+\epsilon_2)(1+\epsilon_3)\frac{1}{n} \sum_{X \in \mathcal{X}_\circ} g(X,y) \sum_{j \in \mathcal{S}_X^2(p_t)} (\overline{a_{i_*^X}}(X) - \overline{a_j}(X))q_{i_*^X}(X)q_j(X)|\gamma_{i_*^X}(X) - \gamma_j(X)|$$

$$+ (1+\epsilon_2)(1+\epsilon_3)\frac{1}{n} \sum_{X \in \mathcal{X}_\circ} g(X,y) \sum_{i \in \mathcal{S}_X(p_t):i \neq i_*^X} \sum_{j \in \mathcal{S}_X^2(p_t)} (\overline{a_i}(X) - \overline{a_j}(X))q_i(X)q_j(X)|\gamma_i(X) - \gamma_j(X)|$$

$$\tag{39}$$

We then compute by Lemma A.2 that

$$-\hat{p}^\top \nabla_p \mathcal{L}(\boldsymbol{E}^1, p_t) = \widehat{\mathbb{E}}\left[g(X,y) \sum_{j \in \overline{\mathcal{S}_X(p_t)}} (\widehat{a_{i_*^X}}(X) - \widehat{a_j}(X))q_{i_*^X}(X)q_j(X)(\gamma_{i_*^X}(X) - \gamma_j(X)))\right]$$

$$\tag{D1}$$

$$+ \widehat{\mathbb{E}}\left[g(X,y) \sum_{i \in \mathcal{S}_X(p_t),i \neq i_*^X} \sum_{j \in \overline{\mathcal{S}_X(p_t)}} (\widehat{a_i}(X) - \widehat{a_j}(X))q_i(X)q_j(X)(\gamma_i(X) - \gamma_j(X)))\right]$$

$$\tag{D2}$$

$$+ \widehat{\mathbb{E}}\left[g(X,y) \sum_{i \in \overline{\mathcal{S}_X(p_t)}} \sum_{j \in \overline{\mathcal{S}_X(p_t)}:j>i} (\widehat{a_i}(X) - \widehat{a_j}(X))q_i(X)q_j(X)(\gamma_i(X) - \gamma_j(X)))\right].$$

$$\tag{D3}$$

We show that by picking

$$d \geq \left(\frac{2816Tn^2(1+2\eta_0)}{\mu}\right)^4,$$

we have

$$|(\text{D2})| \leq \frac{\mu}{2}(\text{D1}),$$

and for any $\epsilon_4 > 0$, there exists $t_4$ such that for all $t \geq t_4$,

$$|(\text{D3})| \leq \epsilon_4(\text{D1}).$$

To show $|(\text{D2})| \leq \frac{\mu}{2}(\text{D1})$, it suffices to note that, for any $j \in \overline{\mathcal{S}_X(p_t)}$,

$$\widehat{a_{i_*^X}}(X) - \widehat{a_j}(X) \geq 1,$$
$$|\widehat{a_i}(X) - \widehat{a_j}(X)| \leq 16n(1+2\eta_0), \qquad \forall i,$$
$$|\gamma_i(X) - \gamma_j(X)| \leq 22\eta_0 d^{-1/4}, \qquad \forall i \neq i_*^X,$$
$$\gamma_{i_*^X}(X) - \gamma_j(X) \geq \frac{\eta_0}{4nT},$$

where the second inequality follows from Lemma C.8 and Lemma B.1. To show $|(D3)| \leq \epsilon_4 (D1)$, we use again the inequalities above and the same strategy as in Lemma C.5.

Thus we obtain that:

$$-\hat{p}^\top \nabla_p \mathcal{L}(\boldsymbol{E}^1, p_t)$$

$$\geq \left(1 - \frac{\mu}{2} - \epsilon_4\right) \widehat{\mathbb{E}} \left[ g(X,y) \sum_{j \in \overline{\mathcal{S}_X(p_t)}} (\widehat{a_{i_*^X}}(X) - \widehat{a}_j(X)) q_{i_*^X}(X) q_j(X) (\gamma_{i_*^X}(X) - \gamma_j(X))) \right]$$

$$\geq \left(1 - \frac{\mu}{2} - \epsilon_4\right) \frac{1}{n} \sum_{X \in \mathcal{X}_0} g(X,y) \sum_{j \in \mathcal{S}_X^2(p_t)} (\widehat{a_{i_*^X}}(X) - \widehat{a}_j(X)) q_{i_*^X}(X) q_j(X) (\gamma_{i_*^X}(X) - \gamma_j(X))),$$

where in the last inequality we use that all the summand are non-negative.

Note that

$$\widehat{a_{i_*^X}}(X) - \widehat{a}_j(X) = 1, \qquad \overline{a_{i_*^X}}(X) - \overline{a_j}(X) \leq 1 - \mu + \epsilon_1, \qquad \forall X \in \mathcal{X}_0.$$

Thus, it remains to show that

$$\frac{\mu}{4n} \sum_{X \in \mathcal{X}_0} g(X,y) \sum_{j \in \mathcal{S}_X^2(p_t)} (\widehat{a_{i_*^X}}(X) - \widehat{a}_j(X)) q_{i_*^X}(X) q_j(X) (\gamma_{i_*^X}(X) - \gamma_j(X))$$

$$\geq (1 + \epsilon_2)(1 + \epsilon_3) \frac{1}{n} \sum_{X \in \mathcal{X}_\circ} g(X,y) \sum_{i \in \mathcal{S}_X(p_t): i \neq i_*^X} \sum_{j \in \mathcal{S}_X^2(p_t)} (\overline{a_i}(X) - \overline{a_j}(X)) q_i(X) q_j(X) |\gamma_i(X) - \gamma_j(X)|$$

$$\tag{40}$$

We have that

$$\widehat{a_{i_*^X}}(X) - \widehat{a}_j(X) \geq 1, \qquad \overline{a_i}(X) - \overline{a_j}(X) \leq 1 - \mu + \epsilon_1, \qquad \forall i \in \mathcal{S}_X(p_t), \quad \forall X \in \mathcal{X}_0.$$

Furthermore,

$$|\gamma_i(X) - \gamma_j(X)| \leq 22 \eta_0 d^{-1/4}, \quad \forall i \neq i_*^X, \qquad \gamma_{i_*^X}(X) - \gamma_j(X) \geq \frac{\eta_0}{4nT}.$$

As $d \geq \left(\frac{2816 T n^2 (1 + 2\eta_0)}{\mu}\right)^4 \geq \left(\frac{176 n^2 T (2 - \mu)}{\mu}\right)^4$, (40) holds and the proof is complete. $\qquad \square$

# D  Details of numerical experiments

For all numerical simulations, we use the AdamW optimizer from `torch.optim`, and we reduce the learning rate in a multiplicative fashion by a factor $\gamma = 0.1$ at epochs 100 and 200, i.e.,

$$\text{LR}_{\text{new}} = \text{LR}_{\text{old}} \cdot \gamma.$$

We adhere to the batch size of 128 and fix the embedding dimension to 2048.

**IMDB and Yelp datasets.** The hyperparameters *do not* differ between the two-layer model and the one-layer model. We set the number of training epochs to 500, the learning rate to 0.01, and the weight decay to $10^{-8}$.

**Synthetic data.** We set the number of training epochs to 196, the learning rate to $10^{-4}$, and the weight decay to $10^{-4}$.

# E  One-step analysis for multi-class classification

We consider there are $K$ classes. The sequence $X$ is defined as usual, and the label $y \in \mathbb{R}^k$ is the one-hot vector indicating the corresponding class. The new model becomes

$$f(X) = VX\texttt{Softmax}(X^\top p) \in \mathbb{R}^K,$$

where $V = [v_1, \ldots, v_K]^\top \in \mathbb{R}^{K \times d}$.

We consider the standard cross-entropy loss:

$$\mathcal{L}_n(p, V) = \widehat{\mathbb{E}}\left[-\log\frac{\exp(y^\top f(X))}{\sum_{k=1}^{K}\exp(e_k^\top f(X))}\right],$$

and compute the gradient as:

$$\partial_{f(X)}\mathcal{L}_n(p, \boldsymbol{E}) = \frac{-\sum_{k=1}^{K}\exp(e_k^\top f(X))(y - e_k)}{\sum_{k=1}^{K}\exp(e_k^\top f(X))},$$

$$\nabla_{E_s}[f(X)]_k = \sum_{i=1}^{T}\left(\sum_{j\neq i}(\mathbb{1}_{x_i=s} - \mathbb{1}_{x_j=s})q_i q_j\right)E_{x_i}^\top v_k p + \sum_{i=1}^{T}\mathbb{1}_{x_i=s}q_i v_k,$$

$$\nabla_{E_s}\mathcal{L}_n(p, \boldsymbol{E}) = \widehat{\mathbb{E}}\left[\sum_{k=1}^{K}\partial_{[f(X)]_k}\nabla_{E_s}[f(X)]_k\right]$$

$$= \widehat{\mathbb{E}}\left[\sum_{k=1}^{K}\left(\frac{-\sum_{\ell\neq k}\exp(e_k^\top f(X))}{\sum_{k=1}^{K}\exp(e_\ell^\top f(X))}\mathbb{1}_{y=e_k} + \frac{\exp(e_k^\top f(X))}{\sum_{k=1}^{K}\exp(e_k^\top f(X))}\mathbb{1}_{y\neq e_k}\right)\right.$$

$$\left.\cdot\left(\sum_{i=1}^{T}\left(\sum_{j\neq i}(\mathbb{1}_{x_i=s} - \mathbb{1}_{x_j=s})q_i q_j\right)E_{x_i}^\top v_k p + \sum_{i=1}^{T}\mathbb{1}_{x_i=s}q_i v_k\right)\right].$$

Thus, at the initial step, we have:

$$\nabla_{E_s}\mathcal{L}_n(p, \boldsymbol{E}) \approx \frac{1}{T}\widehat{\mathbb{E}}\left[\sum_{k=1}^{K}\left(\frac{-(K-1)}{K}\mathbb{1}_{y=e_k} + \frac{1}{K}\mathbb{1}_{y\neq e_k}\right)\sum_{i=1}^{T}\mathbb{1}_{x_i=s}\right]v_k.$$

Consequently, after one gradient step, we obtain the correlation

$$E_s^\top v_k \approx \frac{\eta_0}{T}\widehat{\mathbb{E}}\left[\sum_{i=1}^{T}\frac{K-1}{K}\mathbb{1}_{y=e_k, x_i=s} - \frac{1}{K}\mathbb{1}_{y\neq e_k, x_i=s}\right].$$

Here, by using $\approx$, we hide quantities that have a vanishing order in $d$, and these passages could be made rigorous with a more involved technical analysis.

# F  Additional numerical experiments

## F.1  Variance of one-step gradient descent

In this section, we numerically validate the optimization procedure, which is more in line with the theoretical model of "one-step" analysis presented in the main body. More precisely, we perform one full-gradient step on the embeddings (as in our theoretical analysis), and we then fix the context embeddings, only training the $\langle\text{cls}\rangle$ embedding as well as the output vector $v$ till convergence. The results for the one-layer model (1) on IMDB data are shown in Figure 4. In this case, both overlap statistics display similar behavior to the one in Figures 1 and 3. However, the average standard deviation over the bins is reduced by roughly 8 times for the inner product with $v$ and by roughly 6 times for the inner product with $p$.

We attribute the larger variance of full training dynamics (i.e., with all parameters optimized simultaneously) to both the longer training time for context embeddings and the dataset itself. The dependence on the longer training of the embeddings is indicated in the experiments in Figure 4, and the dependence on the dataset is apparent by comparing Figure 2 (synthetic datasets, small variance) with Figures 1 and 3 (real datasets, large variance).

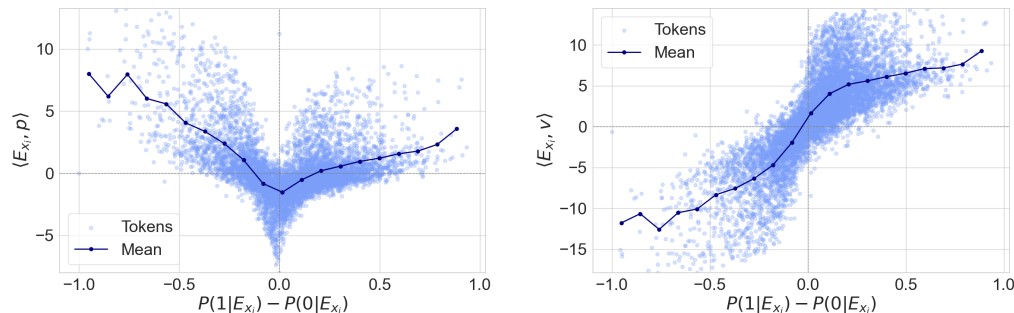

Figure 4: Dot-product of token embeddings with $\langle \mathrm{cls} \rangle$ embedding $p$ (left) and regression coefficients $v$ (right), as a function of the token-wise difference in posterior probabilities, for IMDB dataset under the "one-step" procedure.

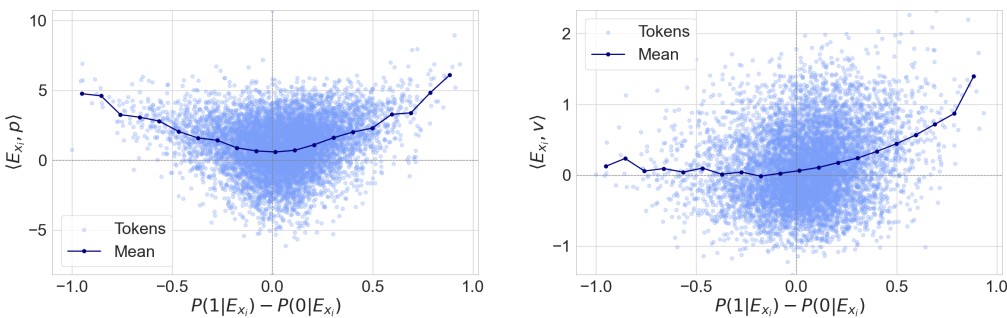

Figure 5: Dot-product of token embeddings with $\langle \mathrm{cls} \rangle$ embedding $p$ (left) and regression coefficients $v$ (right), as a function of the token-wise difference in posterior probabilities, for IMDB dataset for one-layer attention with MLP head.

## F.2 The effect of MLP layer

In this section, we identify how the presence of the extra MLP head after the attention layer in (1) affects the phenomenology described in the main body. Namely, we consider the MLP block which processes input $x \in \mathbb{R}^d$ as follows:

$$x_1 = \mathrm{ReLU}(W_1 x + b_1), \quad W_1 \in \mathbb{R}^{d \times d}, \quad b_1 \in \mathbb{R}^d,$$
$$x_2 = \mathrm{ReLU}(W_2 x + b_2), \quad W_2 \in \mathbb{R}^{d \times d}, \quad b_2 \in \mathbb{R}^d.$$

Then, the output of the self-attention in (1), i.e.,

$$\mathtt{Softmax}(p^\top \boldsymbol{E}_X^\top) \boldsymbol{E}_X \in \mathbb{R}^d,$$

is treated as an input for the MLP block. Afterwards, the MLP output is multiplied with the final regression vector $v$. The correlation analysis for the architecture above is displayed in Figure 5. It is clear that the behavior of $E_s^\top p$ is mostly unchanged, while the monotonicity of $E_s^\top v$ is less evident. This indicates that the model will still select important tokens using $E_s^\top p$, but the interaction between the embeddings and the output vector $v$ is more complicated due to the presence of the MLP layer.

## F.3 Frozen orthogonal embeddings

In this section, we address if training embeddings in model (1) is indeed necessary. Namely, we consider fixing token embeddings to their (almost) orthogonal configuration at initialization and training only $\langle \mathrm{cls} \rangle$ embedding and the output regression vector $v$. The resulting test accuracy is around 72% with test error around 0.6. This is a significant downgrade from the original baseline: the test accuracy is around 95% with test error around 0.1. This clearly indicates the importance of training the embeddings in the self-attention model (1).

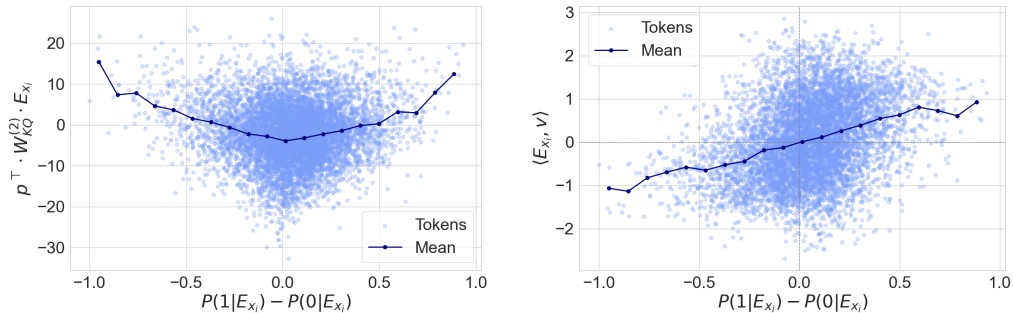

Figure 6: Dot-product of token embeddings with $\langle \text{cls} \rangle$ embedding $p$ and key-query matrix of the second attention layer $W_{KQ}^{(2)}$ (left) and regression coefficients $v$ (right), as a function of the token-wise difference in posterior probabilities, for IMDB dataset and two-layer attention model (18) with key-query matrices.

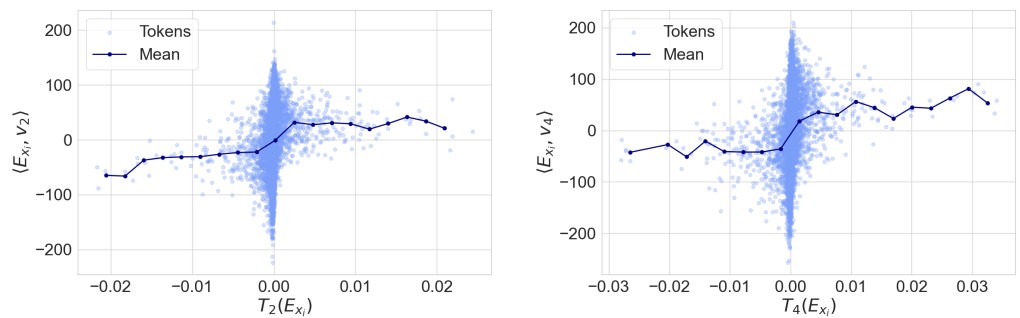

Figure 7: Dot-product of token embeddings with the corresponding regression coefficients $v_k$ (left: $k = 2$; right: $k = 4$), as a function of the token-wise multi-class statistic for Yelp dataset. The token statistic $T_k$ on the $x$-axis is defined as per (41).

### F.4 Influence of key-query matrix

In this section, we explore if the presence of key-query matrices in the self-attention mechanism changes the qualitative implications of our results. For our experimental setup, we consider a two-layer model (18) with key-query matrices in both instances of softmax operator. We present our findings in Figure 6. First, observe that the behavior of the overlap with the regression vector $v$ remains unchanged. Second, we note that adding $W_{KQ}^{(2)}$ in the second layer should not make a qualitative difference. In particular, $W_{KQ}^{(2)} \cdot p$ plays the same role as $p$, which is corroborated by the results in Figure 6. Besides, one can recover the dynamics of key-query matrices given the dynamics of $p$ (cf., Lemma 1 in [37]).

### F.5 Multi-class simulations

Given the analysis in Section E, we consider a modified version for the token-wise statistic. Namely, we focus on the following "one-versus-all" score:

$$T_k(E_s) = \frac{1}{T} \cdot \mathbb{E}\left[\sum_{j=1}^{T} \frac{K-1}{K} \cdot \mathbb{1}_{y=k, x_i=s} - \frac{1}{K} \cdot \mathbb{1}_{y=k, x_i \neq s}\right]. \tag{41}$$

The RHS of (41) can be viewed as a generalization of the average signed frequency (3) that indicates the importance of a token $s$ with respect to class $k$. In particular, if we divide the RHS of (3) by a factor of $K - 1$ the first term becomes the co-occurrence of token $s$ with label $k$, and the second term is the average co-occurrence of token $s$ with other labels. Thus, intuitively, if a token $s$ occurs much

more frequently with label $k$ than other labels, this token is more important with respect to label $k$ and the RHS of (41) will be larger.

For the multi-class dataset, we consider Yelp reviews. We use the 5-star rating directly and disregard the middle-ground rating of 3 (4 classes in total). We train the one-layer model (1) on this data. The results for two classes are presented in Figure 7. Namely, it is clear that $E_s^\top v_k$ is still increasing with respect to the importance of the token $s$ for each class $k$, which aligns with the previous observations for binary classification.

