# OpenReview forum: "Attention with Trained Embeddings Provably Selects Important Tokens"
_NeurIPS.cc/2025/Conference — NeurIPS 2025 poster_

### Official Review · Reviewer_XscA · 2025-06-02

**Clarity:** 3
**Significance:** 3
**Originality:** 3
**Rating:** 5
**Confidence:** 2

**Summary:**

The paper studies the effect of gradient training on the encoding of text tokens in a simplified attention mechanism. They show that even a single gradient step aligns the feature weights such that high frequency words get up weighted.

**Questions:**

- How can the proposed method be extended to more relatistc attention scenarios?
- Can the authors speculate about the practical relevance of their findings, e.g. can this be used for model diagnostics?
- Can other tasks then sentiment analysis be studied with the same framework, like contrastive learning?

**Ethical Concerns:**

["NO or VERY MINOR ethics concerns only"]

**Final Justification:**

After considering the reviews by the other reviewers, the rebuttal and the careful explanations and additional experiments, this paper still seems a worthwhile study, and the authors did a good job of adressing the other reviewer's comments. Of course, theoretical analysis requires simplification, so I do think the approach taken by the authors is warranted.

**Limitations:**

Brief, but ok

**Quality:**

3

**Strengths And Weaknesses:**

Strengths:
- Paper provides theoretical analysis of a simplified scenario for how attention mechanisms learn to adapt to token frequency.
- Theoretical claims are backed-up by experimental evidence.
- The mathemical analysis seems solid, although I could not follow all steps.

Weaknesses:
- Only a simplified model of attention is studied and only a single layer.
- Importance of features is equalized (at least in writing) with importance - but there maybe frequent yet irrelevant features. Also, there may be other semantic concepts that interact, e.g. negating concepts.
- Only limited set of synthetic experiments is performed.
- Only sentiment analysis with a simple classification setting is studied, what about other tasks like contrastive learning.

---

> ### Author Rebuttal · Authors · 2025-07-30
>
> We thank the reviewer for the comments and for the positive evaluation of our work. We address all weaknesses and questions below.
>
> ---
>
> **W1. Simplified model.**
>
> As for the simplified attention model, we remark that **dropping the attention weights does not qualitatively change the problem**. In fact, the product of the attention weights and $p$ plays the same role as $p$ alone without the attention weights, which is to control the attention probability of each token after softmax. Thus, even if we add attention weights $W_{KQ}$, we still expect all the results for $p$ proved in our work to hold for $W_{KQ} p$. Besides, as discussed in [37, Lemma 1], one can recover the dynamics of $W_{KQ}$ given the dynamics of $p$.
>
> We also note that we have conducted experiments on two-layer networks which show a similar phenomenology as the single-layer model, see Figure 1 of our submission.
>
> ---
>
> **W2. Frequent yet irrelevant features; semantic concepts that interact.**
>
> The metric we choose to capture the importance of features is equal to the frequency of occurrence with the +1 label **minus** the frequency of occurrence with the -1 label. This is different from just taking the frequency of the token. In fact, frequent yet irrelevant features will likely appear a similar number of times associated with the +1 label and with the -1 label, and our metric will be small in such cases.
>
> The reviewer makes a good point in mentioning interactions between semantic concepts and we leave a detailed investigation of those to future work. Let us just mention that, while we agree that labels should depend on the interaction of semantic concepts, our results indicate that the first-order statistics already give important information about the labels.
>
> ---
>
> **W3. Limited set of synthetic experiments.**
>
> During the rebuttal, we have performed **additional experiments and ablations**:
>
> (1) We have shown that the variance of token embeddings in the real data experiments is reduced by training the context embeddings just for one step (as opposed to training them until convergence). See our response to **Q3** or *reviewer vwUg* for details.
>
> (2) We have shown that a qualitatively similar picture emerges for multi-class classification. See our response to **Q2** of *reviewer E1M4* for details.
>
> (3) We have analyzed the impact of MLP layers. See our response to **Q4** of *reviewer E1M4* for details.
>
> (4) We have provided evidence that the trained embeddings improve performance on the classification tasks (as opposed to keeping the embeddings fixed to their initialization). See our response to **W1** of *reviewer q6jg* for details.
>
> (5) We have shown that a qualitatively similar picture emerges for a two-layer model with trained attention weight matrices $W_{KQ}^{(1)}$ and $W_{KQ}^{(2)}$. See our response to **Q2** of *reviewer q6jg* for details.
>
> ---
>
> **W4. Only sentiment analysis with simple classification.**
>
> As mentioned above, during the rebuttal, we have considered a **variety of other settings, including multi-class classification** for which we provide both additional experiments and theoretical results. See our response to **Q2** of *reviewer E1M4* for more details.
>
> As for contrastive learning, we discuss the connection in our response to **Q3** below.
>
> ---
>
> **Q1. More realistic attention scenarios?**
>
> First, we note that **adding attention weights $W_{KQ}$ in the second layer does not make qualitative difference**, since now $W_{KQ} p$ has the same role as $p,$ and as discussed in [37, Lemma 1], we can recover the dynamics of $W_{KQ}$ given the dynamics of $p.$
>
> To support this view, we have conducted an **additional experiment** on a two-layer model having attention weights both in the first ($W_{KQ}^{(1)}$) and in the second ($W_{KQ}^{(2)}$) layer. We have found that **$E_s^\top v$ and $E_s^\top W_{KQ}^{(2)} p$ have the same behaviour as in Figure 1**. See our response to **Q2** of *reviewer q6jg* for the exact data in a tabular form.
>
> Thus, the experiment confirms that **dropping the attention weight matrix is a reasonable simplification of the model that does not make a qualitative difference**. In the revision, we will add plots for this experiment, as well as a discussion (due to this year’s NeurIPS policy, we are not allowed to post links to plots in our response).
>
> ---
>
> **Q2. Practical relevance, e.g. for model diagnostics?**
>
> This is an interesting point. The correlations **$E_s^\top v, E_s^\top p$ could provide a useful criterion to detect whether the model is performing well**. In fact, if such correlations are always small, regardless of the importance of the token, then we expect the model to perform poorly. This is in fact the case if we leave the embeddings untrained, as in an ablation we performed to respond to **W1** raised by *reviewer q6jg*.
>
> Specifically, we have performed an **additional experiment providing evidence that the trained embeddings indeed improve performance on the classification tasks**. In particular, we have fixed context embeddings (randomly initialized) and we have only trained the CLS token embedding and the classifier weights. The resulting test accuracy is around 72% with test error around 0.6, while for the original baseline the test accuracy is around 95% with test error around 0.1. This clearly indicates the importance of training the embeddings. We will report the result of this experiment in the revision.
>
> ---
>
> **Q3. Other tasks, e.g. contrastive learning?**
>
> We have provided extra theoretical and experimental results on multi-class classification problems. Please refer to our response to **Q2** of *reviewer E1M4* for details.
>
> Regarding contrastive learning, while there is some high-level connection, our framework is unlikely to be directly applicable. As an example, consider CLIP. Then, intuitively we expect each image to be relevant only to certain words (tokens) in the text. However, we highlight two technical difficulties.
>
> (1) We need a proper definition of the importance of each token to each image from the dataset.
>
> (2) Due to the unsupervised nature of contrastive learning and the difference in architectures used compared to our case, we need to understand what structure of the embeddings would be crucial to learning the datasets.
>
> Resolving such technical difficulties provides an exciting avenue for future research.

---

> > ### Comment · Reviewer_XscA · 2025-08-01
> >
> > I thank the reviewers for their additional explanations. I found them helpful, and think that also the responses to the other reviewers explained their approach well. Unfortunatley, I am not an expert, but I would still advocate accepting the paper, including the promised additions.

---

> > > ### Author Response · Authors · 2025-08-04
> > >
> > > Thank you for the positive feedback on our paper! We would be happy to have further discussions in case you have any additional questions or comments.

---

### Official Review · Reviewer_q6jg · 2025-07-01

**Clarity:** 4
**Significance:** 2
**Originality:** 3
**Rating:** 4
**Confidence:** 4

**Summary:**

This paper shows the role of token embeddings. It studies how the trained token embeddings capture the importance of tokens and selects the relevant tokens. The authors characterize the token embeddings after one gradient step and the implicit bias of \<cls\> embedding. In addition, they conduct synthetic and realistic experiments to support the theoretical findings.

**Questions:**

1. What may happen if the output vector $v$ and the token embedding $E_X$ are trained simultaneously?
2. In line 275, the two-layer model has three parameter matrices $p,v,W_X$, with only $p$ being trained. What would happen if there are two trained matrices $W_{KQ}^{(1)}$ and $W_{KQ}^{(2)}$ in the two layers?

**Ethical Concerns:**

["NO or VERY MINOR ethics concerns only"]

**Final Justification:**

I thank the authors for addressing my concerns. Although it still lacks a rigorous theoretical analysis for training all parameters together, some experiments help verify the results. I have accordingly increased my score.

**Limitations:**

Yes.

**Paper Formatting Concerns:**

I do not find any formatting issues.

**Quality:**

3

**Strengths And Weaknesses:**

Strengths:

1. The topic is interesting and may be the first theoretical study of the embedding training dynamics.
2. The paper is well-written and easy to follow.
3. The mathematical analysis and proofs are solid and clear.

Weaknesses:

1. The paper demonstrates that the trained token embeddings help select the relevant tokens by capturing the importance of different tokens. However, it does not provide evidence that the trained embeddings improve performance on the classification tasks. It is possible that even untrained orthogonal token embeddings could achieve similar classification accuracy, suggesting that the training may not be essential for the task.
2. Assumption 1 appears quite strong, as it assumes that each sequence contains only one relevant token, which must be completely positive or completely negative. What if a sequence contains multiple relevant tokens, or if some relevant tokens are only partially positive or negative? Would the model still be able to select all relevant tokens under these conditions?
3. The output vector $v$ is fixed, and the token embeddings are trained for only one step. It appears somewhat simplistic. It involves fixing all other parameters and only training the embedding $p$, which does not seem to show much novelty compared to [1].

[1] Davoud Ataee Tarzanagh, Yingcong Li, Xuechen Zhang, and Samet Oymak. Max-margin token selection in attention mechanism. NeurIPS, 2023.

---

> ### Author Rebuttal · Authors · 2025-07-30
>
> We thank the reviewer for the comments and for pointing out the strengths of our work. The reviewer raises **valid points** (e.g., on untrained orthogonal token embeddings and on adding $W_{KQ}$ matrices in the model) that we **addressed via additional experiments**. We now elaborate on these points, addressing all questions and concerns below.
>
>
> ---
>
>
> **W1. Untrained embeddings.**
>
> We have performed an additional experiment providing **evidence that the trained embeddings indeed improve performance on the classification tasks**. In particular, we have fixed context embeddings (randomly initialized) and we have only trained the CLS token embedding and the classifier weights. The resulting test accuracy is around 72% with test error around 0.6, while for the original baseline the test accuracy is around 95% with test error around 0.1. This clearly indicates the importance of training the embeddings. We will report the result of this experiment in the revision.
>
> ---
>
>
> **W2. Assumption 1.**
>
> If the sequence contains multiple positive or negative tokens, the behaviour is expected to depend on the specific structure of the dataset. The difficulty is that the model may not necessarily select all relevant tokens. In fact, only selecting one relevant token from each sequence is in principle sufficient, and the negative directional gradient of $p$ along the max-margin direction that selects all the relevant tokens may not be larger than that of other selections. Nevertheless, **two results follow from our analysis**:
>
> (1) If $p$ converges in direction, it **must select at least one relevant token for each sequence**, simply because the loss is non-increasing.
>
> (2) If $p$ converges in direction and selects all relevant tokens, $p$ **must be the max-margin direction of such selection**. This can be obtained by following the approach obtained to derive our Theorem 4.3, which compares the directional gradient of $p$ along the max-margin direction and non-max-margin direction of the same selection.
>
> The case with multiple partially positive or negative tokens is even harder, and it is likely to require additional assumptions on the training set, as well as completely different techniques.
>
> We finally note that assuming only one relevant token is common in the literature [A, B].
>
> [A] Attention layers provably solve single-location regression. ICLR, 2025.
>
> [B] Max-margin token selection in attention mechanism. NeurIPS, 2023.
>
> ---
>
> **W3. Novelty compared to [1].**
>
> Let us elaborate on the technical novelty compared to [1]:
>
> (1) Our results work for **more general settings** as compared to [1]. The directional convergence results in [1] require either only one sequence in the dataset [1, Theorem 2], or an initialization that is close enough to certain directions [1, Theorem 3]. In contrast, **Theorem 4.3 in our work requires none of those restrictions** to characterize all possible directions the gradient flow converges to.
>
> (2) Our results require **fundamentally different techniques** as compared to [1]. Specifically, [1, Theorem 3] suggests that, for any locally optimal max-margin direction, we can find a regime such that the direction of $p$ will converge to that direction. Note that a locally optimal max-margin direction in [1, Definition 2] does **not** necessarily select all the relevant tokens. Our results, on the contrary, show that **$p$ must necessarily converge to a max-margin direction that selects all relevant tokens**. Thus, our focus is on proving the impossibility of converging to “bad” directions under random initialization and one-step of gradient descent on the embeddings, while [1] focuses on proving the convergence under a close-enough initialization to the target direction.
>
> (3) [1] does not consider any connections between the directional convergence and the importance of the tokens, which, in contrast, is our main focus.
>
> ---
>
> **Q1. $v$ and $E_X$ trained simultaneously?**
>
> If we train $v$ but fix the context embeddings, the implicit bias of $p$ is not affected in the limit. To see this, we compute
>
> $$-\nabla_v \mathcal L_n(p,v) =  \hat{\mathbb E}[ y g(X,y) \sum_{i=1}^T q_i(X) E_{x_i} ] \approx \hat{\mathbb E}[ y g(X,y) \sum_{x_i \in \mathcal S_X(p)} q_i(X) E_{x_i} ].$$
>
> Note that, if $p$ converges in direction, it must select all relevant tokens. This means that $y E_s^\top v$ will be increasing if $s$ is relevant, and $E_s^\top v$ will not move a lot if $s$ is irrelevant, since it appears exactly the same number of times in positive and negative sequences.
>
> While we do not have a rigorous theoretical characterization for training all parameters $E,p,v$ together, in our experiments of Figures 1-3 the token embeddings $E$ are trained simultaneously with $p.$ Thus, we expect the token selection to be the same as if we only trained $p$.
>
> ---
>
> **Q2. Only $p$ trained; adding $W_{KQ}^{(1)}$ and $W_{KQ}^{(2)}$.**
>
> First, we note that in the two-layer model experiments, $p,v, E_x$ are trained simultaneously.
>
> Second, we note that **adding $W_{KQ}^{(2)}$ in the second layer does not make a qualitative difference**, since now $W_{KQ}^{(2)} p$ plays the same role as $p$ and, as discussed in [37, Lemma 1], we can recover the dynamics of $W_{KQ}^{(2)}$ given the dynamics of $p.$
>
> Finally, we have conducted an **additional experiment** on a two-layer model with attention weights both in the first ($W_{KQ}^{(1)}$) and the second ($W_{KQ}^{(2)}$) layer. We have found that **$E_s^\top v$ and $E_s^\top W_{KQ}^{(2)} p$ have the same behaviour as in Figure 1**. The results are reported in the table below.
>
> |   |   |   |  |  |  |  |  |  |  |  |  |  |  | | | | | | | |
> |-------|-------|-------|-------|-------|-------|-------|-------|-------|-------|-------|-------|-------|-------|-------|-------|------|------|------|------|-------|
> | **$P(1\|E_s) - P(0\|E_s)$** | -0.95 | -0.85 | -0.76 | -0.66 | -0.56 | -0.47 | -0.37 | -0.28 | -0.18 | -0.08 | 0.01  | 0.11  | 0.21  | 0.30  | 0.40  | 0.50 | 0.59 | 0.69 | 0.79 | 0.89  |
> |**Average  $p^\top  W_{KQ}^{(2)} E_s$** | 15.47 | 7.37  | 7.79  | 4.64  | 3.68  | 1.60  | 0.68  | -0.61 | -2.22 | -2.80 | -3.92 | -3.18 | -2.23 | -1.43 | -0.15 | 0.29 | 3.21 | 2.98 | 7.90 | 12.44 |
>
>
>
> | | | | | | | | | | | | | | | | | | | | | |
> |---------------------------|-------|-------|-------|-------|-------|-------|-------|-------|-------|-------| ------|------|------|------|------|------|------|------|------|------|
> | **$P(1\|E_s) - P(0\|E_s)$** | -0.95 | -0.85 | -0.76 | -0.66 | -0.56 | -0.47 | -0.37 | -0.28 | -0.18 | -0.08 | 0.01 | 0.11 | 0.21 | 0.30 | 0.40 | 0.50 | 0.59 | 0.69 | 0.79 | 0.89 |
> | **Average  $E_s^\top v$** | -1.07 | -1.14 | -0.82 | -0.69 | -0.58 | -0.65 | -0.53 | -0.44 | -0.18 | -0.13 | 0.01 | 0.11 | 0.26 | 0.39 | 0.54 | 0.63 | 0.81 | 0.73 | 0.61 | 0.93 |
>
>
> The experiment confirms that **dropping the attention weight matrix is a reasonable simplification of the model that does not make a qualitative difference**. In the revision, we will add plots for this experiment, as well as a discussion (due to this year’s NeurIPS policy, we are not allowed to post links to plots in our response).

---

> > ### Comment · Reviewer_q6jg · 2025-08-05
> >
> > I thank the authors for addressing my concerns. Although it still lacks a rigorous theoretical analysis for training all parameters together, some experiments help verify the results. I have accordingly increased my score.

---

> > > ### Author Response · Authors · 2025-08-06
> > >
> > > Thank you for the valuable comments and suggestions that have helped us improve the paper, and for raising the evaluation of our paper! We are happy to have further discussion in case the reviewer has additional questions or comments.

---

### Official Review · Reviewer_E1M4 · 2025-07-02

**Clarity:** 3
**Significance:** 2
**Originality:** 3
**Rating:** 4
**Confidence:** 3

**Summary:**

This paper theoretically and empirically demonstrates how token embeddings capture the correlations between tokens and labels during training, using a simple one-layer attention model and a binary classification task. Specifically, the authors first prove that after a single gradient step with the standard logistic loss, the token embeddings begin to encode the empirical importance of each token by aligning with the output vector in proportion to the corresponding empirical frequencies. They then show that, under the assumption that each sequence contains exactly one completely positive or completely negative token and all remaining tokens are irrelevant, gradient flow converges in direction to a classifier embedding that corresponds to the max-margin separator, which maximally distinguishes relevant tokens from irrelevant ones. Finally, the paper verifies these theoretical results with numerical experiments on synthetic and IMDB/Yelp datasets, demonstrating that the learned embeddings indeed reflect the empirical statistics of token-label associations for both one-layer and two-layer attention models.

**Questions:**

(1) Why do the authors choose to use binary classification tasks as the illustrative tasks? Is it because it is simple for analysis and the results align with the sentiment dataset?

(2) If yes to Q1, how to make sure that the conclusions drawn form the binary-classification can extend to other scenarios like multi-classification tasks? Is there any empirical or theoretical thinkings about it?

(3) The theoretical results require the embedding dimension $d$ to grow at least logarithmically in vocabulary size, which can lead to very large $d$ for realistic datasets. I am wondering how sensitive are your conclusions to smaller embedding dimensions commonly used in practice?

(4) The analysis omits the influence of MLP on token embedding training. Could the author please explain what will the attention with MLP impact the conclusions brought by the attention-only structures?

(5) The study mainly focus on the language tasks; however, as transformer or attention mechanism is also widely in vision tasks, could the authors please elaborate how the conclusions brought by this paper can be extended to the vision tasks?

**Ethical Concerns:**

["NO or VERY MINOR ethics concerns only"]

**Final Justification:**

After the rebuttal and discussion with authors, the results are summarized as follows:

**Weaknesses:**

**Weak1 Strong Assumption and Weak3 Generalization concerns:** The authors' reply is convincing to me, which solves my concerns;

**Weak 2 Only focus on first-order statistics:** I acknowledge that the connection between token embeddings and first-order statistics were not revealed in previous works, but the higher-order relations cannot be neglected.

**Questions:**

The authors solve all my questions especially for the multi-classification case and the role of MLP.

**Summary:**

Overall, I appreciate the contributions of this work on revealing the connections of token embeddings and labels in the training process even though there are some limitations like the only focus on first-order statistics. However, because this theory cannot be trivially proved by the real transformer structure, I will give it 4 for borderline acceptance.

**Limitations:**

Please see above weaknesses and questions.

**Paper Formatting Concerns:**

No paper formatting concerns.

**Quality:**

3

**Strengths And Weaknesses:**

**Strengthes:**

**(1):** This work offers a new perspective on understanding the function and properties of token embeddings by analyzing their behavior during gradient descent through both clear and solid theoretical analysis and numerical experiments. It reveals that token embeddings capture the empirical importance of tokens in the dataset, while the classifier embedding converges to the direction corresponding to max-margin solutions that optimally separate relevant tokens from irrelevant ones. These findings help bridge our understanding of how token embeddings in large language models evolve during training and how they contribute to the model’s predictions.

**(2):** The numerical analyses strongly support the theoretical claims. In both the synthetic data experiments and the real-world IMDB/Yelp datasets, it is clearly demonstrated that training a simple attention classifier leads to embeddings that effectively capture how frequently and strongly each token is associated with the label, consistent with the theoretical predictions.

**(3):** The writing is overall clear and easy to follow. The paper is well-organized, and the theoretical claims and proofs presented in the main text are sound and do not exhibit major issues.

**Weaknesses:**

**(1) Strong Assumptions:** The paper assumes that each sequence in the dataset contains exactly one completely positive token or one completely negative token, with all remaining tokens treated as irrelevant. However, in reality, classification labels typically depend on the relationships among combinations of tokens. Under these conditions, the proposed theoretical results do not straightforwardly generalize to a wide range of tasks or more realistic language models. Even for the binary classification tasks studied in this paper, Figures 1 and 3 reveal many outlier points where token embeddings deviate from the predicted trends. Therefore, additional analysis is needed to understand and bridge the gap caused by such outliers.

**(2): Only focus on first-order statistics** As indicates by the paper, only the marginal frequency with which each token appears in positive versus negative examples is considered. However, in reality, higher-order relationships are common in natural language processing for the exiting language models and a purely frequency-based perspective cannot fully explain how modern language models learn rich representations from data.

**(3): Generalization concern:** The paper focuses primarily on a one-layer model and a simple two-layer extension, while modern language models employ deep architectures with multiple layers and attention heads. Bridging the gap between the theory developed here and the behavior of large-scale models remains non-trivial, as suggested by the more variable empirical results. Additionally, this work does not explore the role of feedforward networks (MLPs) in shaping token embeddings, which may further limit the conclusions drawn solely from attention mechanisms.

---

> ### Author Rebuttal · Authors · 2025-07-30
>
> We thank the reviewer for the comments and for appreciating the perspective brought forward by our work. The reviewer raises **valid points** (e.g., on multi-class classification and the addition of MLP layers) that we **addressed via additional experiments**. We also clarified our assumption on the embedding dimension $d$. We now elaborate on these points, addressing all questions and concerns below.
>
> ---
>
> **W1. Strong Assumptions.**
>
> While we agree that labels depend on token combinations, we argue that our setting, albeit simplified, is a first step towards a full understanding of the structure of trained embeddings. We bring the following evidence in favor of this argument:
>
> (1) The setting with a single relevant token was also considered in previous work [1,2].
>
> (2) Generalization to multi-class classification is rather direct and presents a similar phenomenology. See our response to **Q2** below for details.
>
> (3) Fixing the context embeddings and only training the CLS embedding as well as the output vector significantly reduces the variance in our scatterplots, indicating that the outlier points mentioned by the reviewer are likely due to the longer training of the embeddings. See our response to **Q3** of *Reviewer vwUg* for details.
>
> ---
>
> **W2. Only focus on first-order statistics.**
>
> While we agree that labels should depend on higher-order relations, our results indicate that the first-order statistics still give important information about the labels. In particular, in the experiments, we see that tokens with different average signed frequency play a different role in the classification (tokens with large $|E_s^\top v|$ and $E_s^\top p$ play a more important role). In addition, the connections between token embeddings and first-order statistics were not revealed in previous works to our best knowledge.
>
> ---
>
> **W3. Generalization concern.**
>
> We will discuss the role of MLPs, see our response to **Q4** below.
>
> ---
>
> **Q1. Why binary classification?**
>
> In binary classification, it is more intuitive to define the importance of the token w.r.t the labels (i.e., the average signed frequency we defined in the paper); in contrast, in other settings, e.g. generative tasks, quantifying such importance is less obvious.
>
> As mentioned in the introduction, we first conducted experiments on the sentiment dataset, and then we provided a theoretical analysis to understand the experimental phenomenon, rather than picking a-posteriori a setting that aligns with the theoretical analysis.
>
> Finally, we note that binary classification problems have been widely considered in prior work, see e.g. [2, 3].
>
> ---
>
> **Q2. Multi-class classification?**
>
> We provide below both **theoretical and experimental evidence that our conclusions for binary classification extend to the multi-class setting**.
>
> *Theoretical evidence.* Consider $K$-class classification, let $v_k$ be the $k$-th classifier and perform one step of gradient descent on the token embeddings. Then, after some manipulations, we obtain that
> $$ E_s^\top v_k \approx \frac{\eta_0}{T} \hat{\mathbb E}\left[ \sum_{i=1}^T \frac{K-1}{K} 1_{y = e_k, x_i = s } - \frac{1}{K} 1_{y \neq e_k, x_i = s }\right].$$
>
> The RHS of the equation above can be viewed as a generalization of the average signed frequency that indicates the importance of a token with respect to class $k.$ In particular, if we divide the RHS by a factor of $K-1,$ the first term becomes the co-occurrence of token $s$ with label $k,$ and the second term is the average co-occurrence of token $s$ with other labels. Thus, intuitively, **if a token occurs much more frequently with label $k$ than other labels, this token is more important w.r.t. label $k,$ and the correlation with $v_k$ is larger**.
>
> *Experimental evidence.* We conduct experiments on the Yelp datasets with **4 classes**, and we find that **$E_s^\top v_k$ is still increasing with respect to the importance of the token for each class $k$, as for binary classification**. We display the results for class 2 and 4 in the tables below. For class 1 and 3, the results are similar.
>
>
>
> |       |        |        |        |        |        |        |        |        |       |       |       |       |       |       |       |       |       |       |       |
> |-------|--------|--------|--------|--------|--------|--------|--------|--------|-------|-------|-------|-------|-------|-------|-------|-------|-------|-------|-------|
> | **$P(1\|E_s) - P(0\|E_s)$** | -0.026 | -0.020 | -0.017 | -0.014 | -0.011 | -0.008 | -0.005 | -0.002 | 0.001 | 0.005 | 0.008 | 0.011 | 0.014 | 0.017 | 0.020 | 0.023 | 0.026 | 0.029 | 0.032 |
> | **Average $E_s^\top v_2$** | -42.375| -27.306| -51.164| -20.713| -40.923| -41.925| -41.884| -35.691| 18.255| 35.633| 30.524| 56.211| 44.184| 23.280| 45.341| 43.210| 62.935| 81.274| 53.548|
>
>
> |       |        |        |        |        |        |        |        |        |        |       |       |       |       |       |       |       |       |       |       |
> |-------|--------|--------|--------|--------|--------|--------|--------|--------|--------|-------|-------|-------|-------|-------|-------|-------|-------|-------|-------|
> | **$P(1\|E_s) - P(0\|E_s)$** | -0.021 | -0.018 | -0.016 | -0.014 | -0.011 | -0.009 | -0.007 | -0.004 | -0.002 | 0.000 | 0.002 | 0.005 | 0.007 | 0.009 | 0.012 | 0.014 | 0.016 | 0.019 | 0.021 |
> | **Average $E_s^\top v_4$** | -64.776| -66.074| -37.040| -32.499| -31.109| -30.686| -26.341| -23.153| -22.310| -0.648| 31.915| 27.601| 30.951| 29.300| 19.455| 29.693| 41.506| 34.151| 21.060|
>
>
> In the revision, we will add the derivation of the formula above, plots for the experiments, as well as a discussion (due to this year’s NeurIPS policy, we are not allowed to update the PDF or post links to plots in our response).
>
> ---
>
> **Q3. Embedding dimension.**
>
> We highlight that **the vocabulary in our paper only includes the tokens that appear in the training set**, rather than the whole vocabulary of the tokenizer. We note that **the embedding dimension $d$ grows roughly linearly in the vocabulary size in the numerical simulations**: we pick embedding dimension 2048 (as common in practical models), and the vocabulary size is 7766 for IMDB and 5424 for Yelp.
>
> ---
>
> **Q4. Role of MLP.**
>
> We have performed additional experiments to study the impact of MLP layers. In particular, we have added one MLP layer after the softmax, with the same hidden dimension as the embedding. We find that $E_s^\top p$ is still increasing with $\alpha_s$, while the monotonicity of $E_s^\top v$ is less evident. This indicates that **the model will still select important tokens using $E_s^\top p$**, but the interaction between the embeddings and the output vectors is more complicated due to the presence of the MLP layer.
>
>
> |       |       |       |       |       |       |       |       |       |       |       |      |      |      |      |      |      |      |      |      |      |
> |-------|-------|-------|-------|-------|-------|-------|-------|-------|-------|-------|------|------|------|------|------|------|------|------|------|------|
> | **$P(1\|E_s) - P(0\|E_s)$** | -0.95 | -0.85 | -0.76 | -0.66 | -0.56 | -0.47 | -0.37 | -0.28 | -0.18 | -0.08 | 0.01 | 0.11 | 0.21 | 0.30 | 0.40 | 0.50 | 0.59 | 0.69 | 0.79 | 0.89 |
> | **Average $E_s^\top v$** | 0.13  | 0.24  | 0.06  | 0.09  | 0.04  | 0.10  | 0.01  | 0.04  | -0.01 | 0.03  | 0.06 | 0.11 | 0.18 | 0.24 | 0.34 | 0.44 | 0.57 | 0.72 | 0.87 | 1.40 |
>
>
> |       |       |       |       |       |       |       |       |       |       |       |      |      |      |      |      |      |      |      |      |      |
> |-------|-------|-------|-------|-------|-------|-------|-------|-------|-------|-------|------|------|------|------|------|------|------|------|------|------|
> | **$P(1\|E_s) - P(0\|E_s)$** | -0.95 | -0.85 | -0.76 | -0.66 | -0.56 | -0.47 | -0.37 | -0.28 | -0.18 | -0.08 | 0.01 | 0.11 | 0.21 | 0.30 | 0.40 | 0.50 | 0.59 | 0.69 | 0.79 | 0.89 |
> | **Average $E_s^\top p$** | 4.76  | 4.61  | 3.26  | 3.08  | 2.80  | 2.06  | 1.60  | 1.43  | 0.89  | 0.67  | 0.59 | 0.71 | 1.11 | 1.61 | 2.02 | 2.30 | 3.28 | 3.39 | 4.84 | 6.10 |
>
> In the revision, we will add plots for the experiments, as well as a discussion (due to this year’s NeurIPS policy, we are not allowed to post links to plots in our response).
>
> ---
>
> **Q5. Vision tasks.**
>
> The embedding layer of a typical vision transformer (ViT) is a linear transformation over the patches, and we regard the frequency of patches as not very meaningful due to the continuous nature of the vision data. Since our results require a finite vocabulary, the study of vision tasks would require a different definition of the importance of patches, which we leave to future work.
>
> ---
>
> **References**
>
> [1] Attention layers provably solve single-location regression. ICLR, 2025.
>
> [2] Max-margin token selection in attention mechanism. NeurIPS, 2023.
>
> [3] Implicit bias and fast convergence rates for self-attention. TMLR, 2025.

---

> > ### Comment · Reviewer_E1M4 · 2025-08-06
> >
> > Thanks for the detailed reply from the authors, which solves my major concerns including the validity of assumptions, generalization to multi-class classification, and the analysis of considering MLP layer. After thoroughly reading the paper, I believe the strengths of the work outweigh its limitations. I will raise my rating to 4.

---

> > > ### Author Response · Authors · 2025-08-06
> > >
> > > Thank you for the valuable comments and suggestions that have helped us improve the paper, and for raising the evaluation of our paper! We are happy to have further discussion in case the reviewer has additional questions or comments.

---

### Official Review · Reviewer_vwUg · 2025-07-22

**Clarity:** 3
**Significance:** 2
**Originality:** 2
**Rating:** 4
**Confidence:** 3

**Summary:**

This paper studies a simplified one-layer attention model in a binary sequence classification setting, where the only trainable parameters are the input token embeddings. The authors analyze the training dynamics and show that, after one gradient step, the embeddings begin to align with the fixed output vector, with their magnitudes scaled by the *signed* frequency of each token. This effectively encodes first-order statistics of the input into the embeddings.

Next, with the input embeddings frozen after the first step, the authors characterize the implicit bias of the `<cls>` token embeddings, showing it converges to the max-margin solution that separates positive/negative tokens (tokens that appear more frequently in sequences from one of the classes) from the irrelevant ones (those equally likely to appear in the sequences of any class). This result relies on the assumption that each sequence contains exactly one relevant token that determines the label.

The authors support their theoretical findings by experiments on both synthetic datasets and sentiment classification tasks (IMDB and Yelp). In both settings, they observe that, on average, tokens with higher signed frequency receive higher attention (measured by inner product of their embeddings with the `<cls>` token) and align more strongly with the output vector.

**Questions:**

1. What’s the motivation for fixing the token embeddings $E$ and only training the `<cls>` token? Is this just for simplification, or is there an argument that further training on $E$ does not provide more insights?

2. In the synthetic data generation, are tokens sampled independently as in equation (15)? If so, does the generated data actually satisfy Assumption 1? Or are the results in the synthetic experiments for a more general setting?

3. I understand that the IMDB and Yelp datasets differ substantially from the synthetic setup, but the variance in token embeddings seems quite high. Since the one-step result holds for arbitrary data, would training for just one step lead to lower variance and better alignment with the theoretical predictions? In other words, is the large variance an artifact of longer training or more of other dataset-specific factors?

**Ethical Concerns:**

["NO or VERY MINOR ethics concerns only"]

**Final Justification:**

The rebuttal clarified the connections to some closely related works and added useful discussion and new empirical results, which made the contribution, scope, and limitations much clearer. The core message and setup design are well chosen, and the experiments support the claims; the new results also reveal the limitations, giving a more complete picture. Based on this, I raised my score to borderline accept. I’m keeping it at borderline as the analysis still relies on several simplifying assumptions about the model.

**Quality:**

3

**Strengths And Weaknesses:**

I found the setup for analyzing how first-order statistics get encoded into the input embeddings clean, and the experiments support this intuition to some extent.

The model, however, is simplified. The attention weights are dropped entirely, and attention effectively reduces to computing directly the inner products between the token embeddings and the `<cls>` token.

The max-margin convergence result overlaps with prior work, as also cited in the paper, e.g., [32, 37, 40], which also considers similar synthetic setups. The comparison with [37] is helpful, especially in clarifying the distinction between local and global convergence. However, [40] also presents global convergence analysis in a related setting. If there are meaningful differences in results, techniques, or assumptions compared to [40], it would be helpful to include a discussion.

---

> ### Author Rebuttal · Authors · 2025-07-30
>
> We thank the reviewer for the comments. We appreciate that the reviewer found our setup to be clean and, while we acknowledge the simplifications it brings, we highlight that **it still displays the same phenomenology as more realistic setups**, with the important advantage of being mathematically tractable. We have also discussed in detail differences with [40] and performed the **additional experiment suggested in the last question** finding a conclusion aligned with the intuition of the reviewer. We now elaborate on these points, addressing all questions and concerns below.
>
> ---
>
> **W1. Simplified model.**
>
> The problem we address in our work is to characterize the structure learnt by embeddings during gradient descent training. **Dropping the attention weights does not qualitatively change this problem**. In fact, the product of attention weights and $p$ plays the same role as $p$ alone without attention weights, which is to control the attention probability of each token after softmax. Thus, even if we add attention weights $W_{KQ}$, we still expect all the results for $p$ proved in our work to hold for $W_{KQ} p$. Besides, as discussed in [37, Lemma 1], one can recover the dynamics of $W_{KQ}$ given the dynamics of $p$.
>
> ---
>
> **W2. Comparison with [40].**
>
> There are **two key differences** between our setup and that of [40]:
>
> (1) **[40, Assumption 1] requires that all the tokens in non-relevant positions are nearly orthogonal to each other**, which is crucial to their proof. In the notation of our paper, they require that, for any $X, X’,$ $E_{x_i}, E_{x’_j}$ are almost orthogonal to each other, for any $x_i, x_j’$ that are not relevant.  This is **not true in our case**, since all the irrelevant tokens appear at least in two sequences (otherwise, the token would have non-zero average signed frequency). Specifically, for any irrelevant token $s,$ there exist $X,X’$ such that $x_i=x_j’=s.$ Since $s$ has non-zero embedding, [40, Assumption 1] cannot be true in our case, and the global convergence result cannot be applied.
>
> (2) [40, Assumption 2] requires all the irrelevant tokens in the sequence to have **exactly the same score** $\gamma_s = y E_s^\top v.$ This assumption leads to the fact that for each sequence, only the term that involves the relevant tokens contributes to the directional gradient, see [40, Equation (5)]. **Our setting does not satisfy this extra assumption**. Even though the score differences between different irrelevant tokens are of order $O(1/\sqrt{d}),$ these errors matter when the gradient norm is diverging to infinity, and the analysis in [40] cannot be applied.
>
> Thus, our focus is on proving the impossibility of converging to “bad” directions under random initialization and one-step of gradient descent on the embeddings. In contrast, [40] focuses on proving the convergence under extra assumptions on the orthogonality of embeddings and scores, which is a fundamentally different problem.
>
> ---
>
> **Q1. Why fix token embeddings?**
>
> Fixing the context token embeddings is indeed for technical simplification. However, we note that the behavior described by the theory is well aligned with our numerical experiments in which all token embeddings are trained until convergence, see Figures 1-3. Thus, **we do expect that further training on the embeddings does not significantly change the structure learnt via gradient descent**.
>
> ---
>
>
> **Q2. Clarification on synthetic data generation.**
>
> The synthetic data is generated exactly as in Equation (15), and the datasets in general do not satisfy Assumption 1. Thus, our results in the synthetic experiments hold for a more general setting. To recover a setting that satisfies Assumption 1, we can simply set $\tilde \delta = (|\mathcal S|-1)/|\mathcal S|,$ then with high probability the dataset satisfies Assumption 1. Overall, the goal of the synthetic experiments is to illustrate the connections between the importance of the tokens and the structure of the embeddings, which motivates our work. We will clarify these points in the revision.
>
> ---
>
> **Q3. High variance in IMDB and Yelp datasets.**
>
> This is an interesting question, and the reviewer has the right intuition. We have performed **additional experiments**, where we train one full-gradient step on the embeddings (as in the theory) and we then fix the context embeddings only training the CLS embedding as well as the output vector. As a result, the average standard deviation over the bins is reduced by roughly 8 times for the inner product with $v$ and by roughly 6 times for the inner product with $p$, compared to training all context embeddings until convergence. This gives evidence that, **by training the context embeddings only for one step, the variance is indeed significantly lower**.
>
> We expect the large variance to be due to both the longer training of embeddings and the dataset itself. The dependence on the longer training of the embeddings is indicated in the experiments mentioned above, and the dependence on the dataset is apparent by comparing Figure 2 (synthetic datasets, small variance) with Figures 1 and 3 (real datasets, large variance).
>
> In the revision, we will add plots for the experiments mentioned above (due to this year’s NeurIPS policy, we are not allowed to post a link to a plot in our response), as well as a discussion.

---

> > ### Comment · Reviewer_vwUg · 2025-08-06
> >
> > Thank you for the clarifications.
> >
> > Including the discussion of [40] in the related-work section is a good addition, given its close connection to your synthetic setup.
> >
> > To restate what I see in the experiments discussed in Q3: a single gradient-descent step aligns the model with the first-order statistics of the tokens. With more training, that alignment weakens (as we see in the plots already in the paper) because the dataset also contains higher-order statistics that an optimal classifier must capture; these emerge in the parameters only after additional training. The current data model cannot probe this phenomenon, since knowing the first-order statistics already yields the optimal classifier and the task does not depend on higher-order token features.
> >
> > Not that this limitation diminishes the present contribution, but I think it's helpful to note such limitations and clarify which parts of the theory would still hold in a more complex setting where higher-order statistics matter, as an example, whether Lemma 4.1 assumes only first-order statistics are relevant to the task.

---

> > > ### Author Response · Authors · 2025-08-06
> > >
> > > Thank you for your valuable suggestions and comments!
> > >
> > > We will indeed add the detailed comparison with [40] in the revision as suggested by the reviewer.
> > >
> > > We will also add the experiments with one-step GD on the embeddings on real datasets (i.e. the settings in our response to **Q3**), and we will add the discussion on the potential limitation of one-step GD in learning the higher-order statistics.
> > >
> > > Regarding the scope of the theoretical results, we note that Lemma 4.1 **does not**  require any assumptions on the datasets, which means that it can be applied to **datasets with arbitrary complicated higher-order relations,** including real datasets. In fact, the more important assumption here is the large dimension of the embeddings, and Lemma 4.1 follows from the structure of the first gradient in high-dimensional settings.
> > >
> > > We also summarize the scope of each lemma and theorem below:
> > >
> > > *Lemma 4.2:* Only require high-dimensional assumption, same as Lemma 4.1, thus would still hold for more complicated datasets.
> > >
> > > *Theorem 4.3, Lemma 4.4, Lemma 4.5:* Require specific assumptions on the structure of datasets (Assumption 1), and in general, it is hard to generalize to complicated datasets. Nevertheless, we expect those results can be generalized to datasets that are slightly more complicated than in Assumption 1. Please refer to our response to **W2** of *Reviewer q6jg* for more details.
> > >
> > > We will also add the discussions of the scope of each theoretical result in the revision to avoid confusion.
> > >
> > > Finally, we would like to thank the reviewer again for the valuable comments and suggestions that have helped us improve the paper, and we are happy to have further discussion in case the reviewer has additional questions or comments.

---

> > > > ### Comment · Reviewer_vwUg · 2025-08-08
> > > >
> > > > Thanks for the discussion and clarifications.
> > > >
> > > > While the analysis uses a simplified setup with some limitations, the addition of the results and discussions in the rebuttal clarifies the scope and makes the contribution clearer and more useful. I’ll raise my score to 4.

---

### Note · Authors · 2025-08-12

We would like to thank all reviewers for their valuable feedback that has helped us improve the paper! We summarize the major changes in our revision below:

- We add discussions that dropping the attention weights does not qualitatively change the problem and experiments on two-layer networks with KQ matrix which further justify this, which address  **W1** of *Reviewer vwUg*, **W1** of *Reviewer XscA* and **Q2** of *Reviewer q6jg*.

- We add a detailed comparison with [37] ([1] in the review of *Reviewer E1M4*) and [40], specifically a discussion on the fundamental technical differences and why their results can not be directly applied, which addresses  **W2** of *Reviewer vwUg* and **W3** of *Reviewer E1M4*.

- We add the experiments with one-step GD on the embeddings and orthogonal embeddings on real datasets. The experiments show that one-step of GD on embeddings significantly reduces the variance of $E_s^\top v$ and $E_s^\top p$ compared to fully-trained embeddings, and improves the performance compared to orthogonal embeddings, which indicates that training one-step GD is still meaningful. The experiments address **Q1, Q3** of *Reviewer vwUg*, **Q2** of *Reviewer XscA* and **W1** of *Reviewer q6jg*. We also add discussions on the potential limitation of one-step GD in learning the higher-order statistics and the scope of each theoretical result, as suggested by *Reviewer vwUg* in the discussions. We add the potential application to model diagnostics as proposed in **Q2** of *Reviewer XscA*.


- We add experiments that show the similar phenomena under our settings are also expected on two different settings: *(1)* multi-class classification problems ($E_s^\top v_k$ increasing w.r.t the importance of the token for each class $k$), and *(2)* one-layer model with MLP layer after softmax ($E_s^\top p$ still selects important tokens), which indicates the potential generalization of our theoretical results and addresses **W3, Q1, Q2 ,Q4** of *Reviewer E1M4* and **W4** of *Reviewer XscA*.

- We add  discussions on the non-rigorous theoretical results of having multiple relevant tokens in each sequence, and training $v, E_X$ simultaneously, as a response to **W2, Q1** of *Reviewer E1M4*. We also discuss the technical difficulties of obtaining fully rigorous theoretical results under these settings.

- We add clarification on synthetic data generation that addresses **Q2** of *Reviewer vwUg*.

---

### Decision · Program_Chairs · 2025-09-17

**Decision:**

Accept (poster)

**Comment:**

The paper gives a theoretical account of how trained token embeddings in a simplified attention classifier encode first-order token-label statistics after one gradient step and how training the CLS vector drives max-margin token selection under a single-relevant-token data model. The main claims are supported by synthetic and IMDB/Yelp analyses, including a two‑layer variant that shows similar trends. The authors’ rebuttal strengthened the work with clarifications against closely related results, a multi‑class extension (derivation and experiments), MLP and two‑layer WKQ ablations, and an accuracy ablation showing trained embeddings are important addressing several reviewer concerns. The chief limitations remain the strong data/model assumptions and the focus on first‑order statistics.